# The mechanics of landslide mobility with erosion

Shiva P. Pudasaini [1,2 ✉] & Michael Krautblatter [1]

Erosion can significantly increase the destructive power of a landslide by amplifying its volume, mobility and impact force. The threat posed by an erosive landslide is linked to its mobility. No mechanical condition has yet been presented for when, how and how much energy erosive landslides gain or lose. Here, we pioneer a mechanical model for the energy budget of erosive landslides that controls enhanced or reduced mobility. Inertia is related to an entrainment velocity, is a fundamentally new understanding. This ascertains the true inertia of erosive landslides, making a breakthrough in correctly determining the landslide mobility. Erosion velocity, which regulates the energy budget, determines the enhanced or reduced mobility. Newly developed energy generator offers the first-ever mechanical quantification of erosional energy and a precise description of mobility. This addresses the long-standing question of why many erosive landslides generate higher mobility, while others reduce mobility. We demonstrate that erosion and entrainment are different processes. Landslides gain energy and enhance mobility if the erosion velocity exceeds the entrainment velocity. Energy velocity delineates distinct excess energy regimes. Newly introduced mobility scaling and erosion number deliver the explicit measure of mobility. Presented dynamical equations correctly include erosion induced net momentum production.

[1] Technical University of Munich, Chair of Landslide Research, Arcisstrasse 21, 80333 Munich, Germany. [2] University of Bonn, Institute of Geosciences, Geophysics Section, Meckenheimer Allee 176, 53115 Bonn, Germany. ✉email: shiva.pudasaini@tum.de

Erosion, entrainment, and deposition are dominant and complex mechanical processes in geophysical mass flows including landslides, avalanches, and debris flows. Such events can disproportionately increase their volume and destructive potential, and become exceptionally mobile by entraining sediment from the bed as they rush down mountain slopes[1–6]. Landslide mobility is associated with erosion-induced excessive volume and material properties, and is characterized by an enormous impact force, exceptional travel distance, and inundation area. Mobility is among the most important features of the landslide as it directly measures the threat posed by the landslide. Mobility is governed by the state of energy of the landslide and entrainment can increase the landslide volume by several orders of magnitude[4,7–10]. The breach of the moraine dam of Lake Palcacocha by the 1941 glacial lake outburst flood event (Cordillera Blanca, Peru) lowered the valley bottom by as much as 50 m in some parts[11]. Thus, erosion-induced excessive volume is a key control on the flow dynamics including velocity, travel distance, flow depth, and impact area, in turn affecting the number of fatalities[1,7,8,12,13]. Erosion, entrainment, and associated flow bulking in landslide-prone areas and debris-flow torrents are a major concern for civil and environmental engineers and landuse planners, and require a cost-intensive remediation. Estimations of flow volume, velocity, and the travel distance are key for assessment of mass flow hazard, design of protective structures, and mitigation measures[12–14]. Different field and laboratory studies on bed sediment entrainment[2,15–20] have suggested that the spatially varying erosion rates and entrainment processes are dependent on the geomorphological, lithological, and mechanical conditions. These processes are of theoretical and practical interests both for scientists and engineers[21]. A proper understanding of landslide erosion, entrainment, and resulting increase in mass (or, volume) is a basic requirement for an appropriate modeling of landslide motion and its impact because the associated risk is directly related to the landslide mass and its velocity. However, as mechanical controls of erosion and entrainment are not well understood yet, evolving volume, run-out, and impact force of landslides and debris flows are often largely underestimated[21].

Physical experiments[4,18,22–24] and theoretical modeling[8,25–27] demonstrate the importance of erosion phenomena in landslides and debris flows. In recent years, there has been rapid increase in the studies of erosion and entrainment in both laboratory[4,18,25] and field scales[13,28–31]. Hereby, continuum mechanical[27,32] and kinetic theory[33] approaches have been applied to investigate erosion phenomena. Empirical[8,16,34–36] and mechanical[25,37] erosion models have been developed. Most erosion models consider effectively single-phase, or at most quasi two-phase flows[8,25,35,37–40]. Erosion may depend on the flow depth, flow velocity, solid concentration, density ratio, bed slope or, the effective stresses at the interface, and initial and boundary conditions[17,41–44]. Recently, an increasing number of numerical models incorporating erosion have been proposed[8,26,35,45–47]. However, the erosion rates presented and utilized in these works are either not based on physical principles or, these are physically inconsistent[13,27]. Specific physical shortcomings will be presented in the "Results" and "Discussion" sections.

Although erosion and deposition play an important role in mass transport and shaping the landscape[1,6,7,13,21,48] our understanding of these processes is not sufficient to apply or extrapolate them beyond empirical experience. Despite the importance of entrainment to hazard assessment and landscape evolution[28–31], a clear understanding of the basic process still remains elusive owing to a lack of high-resolution field-scale data, and also limited flow parameters in laboratory experiments[4]. However, due to the complex terrain, infrequent occurrence, and high time and cost demands of field measurements, the available field data[9,17,20,21,49] are insufficient[13]. This is because a proper understanding and interpretation of the data obtained from the field measurements are often challenging because of the very limited knowledge of the material properties, flow dynamics, and boundary conditions. Measurements are locally or discretely based on points in time and space[9,17,20,21,49]. Physics-based models and numerical simulations may overcome these limitations and facilitate a more complete understanding by investigating much wider aspects of the flow parameters, erosion, mobility, and deposition. Similarly, exact, analytical solutions[50–53] can provide important insights into the complex flow behaviors and their consequences.

By extending the general debris-flow model[54], Pudasaini and Fischer[27] proposed a process-based two-phase erosion–deposition model, which, to a large extent, is capable of adequately describing these complex phenomena commonly observed in landslides, avalanches, debris flows, and bedload transports. These mechanical erosion-rate models proved that the effectively reduced friction (force) in erosion is equivalent to the momentum production. This solves the long-standing dilemma of mass mobility, and shows that erosion can enhance the mass flow mobility. The importance of the Pudasaini and Fischer[27] mechanical erosion model for two-phase mass flows consisting of viscous fluid and solid particles are increasingly realized in recent simulations of mixture mass flows considering the real catastrophic events[6,48,55–60]. These modeling approaches have clearly indicated how the mechanical erosion model could appropriately simulate the actual flow dynamics, surge development, run-out or mobility, and deposition morphology based on the mechanical erosion rates and the erosion-induced momentum productions.

It becomes increasingly evident that bed entrainment, as an interplay of different complex phenomena, plays a major role in determining the landslide and debris-flow propagation pattern[27]. Combining various physical processes, Cascini et al.[28] observed from the 1998 Sarno-Quindici events that, increasing entrainment rate inside the channel may diminish the final run-out of channelized landslides of the flow type. Cuomo et al.[30] deduced from the 1999 Cervinara and the 2005 Nocera Inferiore debris avalanches that bed entrainment can be a dissipative mechanism to reduce mobility of unchannelized flow-like landslides. By analyzing several channelized and unchannelized flows interacting during the propagation, Cuomo et al.[31] suggested that bed entrainment in the central-lower part of the propagation path could reduce the run-out. These findings can be explained[27]. Physically, entraining material from the ground into the moving mass may cause a reduction in kinetic energy (or, the momentum) potentially exceeding the increase in its potential energy. This may occur for flows in moderate to low slope angles. Erosion-related mobility is site and material-specific, and depends on erosional and momentum exchange mechanisms, involved flow rheologies, and how net mass and momentum productions are considered in the dynamical model equations. The Pudasaini and Fischer[27] model built a foundation by mechanically including the momentum production. Nevertheless, their model appears incomplete as they did not deal with the inertia of the entrained mass and could not present a clear mechanical condition for when and how the mobility of an erosive landslide will be enhanced or reduced and how to quantify it. Thus, whether erosion will result in the enhanced mass flow mobility and the quantification of its influence remains an unsolved problem.

Extending the Pudasaini and Fischer[27] model, here, we address this important issue by explicitly deriving mechanical conditions for the mobility of erosive landslides. By introducing a simple landslide mobility equation, we mechanically explain how and when erosive landslides enhance or reduce their mobility. This has been made possible by physically correctly considering the inertia of the erosive landslide. The model offers the first-ever opportunity to distinctly

quantify the mobility of an erosive landslide. We also present some analytical results with plausible parameters and reveal several major novel dynamical aspects associated with erosion-induced landslide mobility. We reveal that the erosion velocity plays a key role in appropriately determining the energy budget of an erosive landslide, providing a precise description of mobility in terms of energy velocity and energy generator. Importantly, a novel mechanism of landslide-propulsion has been identified that emerges from the net momentum production, providing the erosion-thrust to the land-slide. We construct two dimensionless numbers, the mobility scaling and the erosion number, the first of this kind in mass flow modeling, delivering the explicit measure of mobility. The mobility scaling (precisely) quantifies the contribution of erosion in landslide mobility. We analytically obtain the landslide velocity, where there is an increase in landslide velocity due to the entrainment of eroded material by solving the landslide mobility equation that quantifies the effect of erosion in landslide mobility. The explicit form of velocity is technically important in solving engineering problems as it provides the practitioners with key information in quickly estimating the impact force for erosive landslide. Obtained velocities indicate the fact that erosion can have the major control on landslide dynamics. We also derive a full set of dynamical equations in conservative form, in which the momentum balance correctly includes the erosion-induced change in inertia and the momentum production. This constitutes a foundation for legitimate and physically plausible simulation of landslide motion with erosion.

## Results

**Basic balance equations for erosive landslide**. First, we define variables and parameters. The landslide mass is denoted by $m$. The superscripts $^m$ and $^b$ represent the parameters or the quantities in the landslide (mixture), and the erodible basal substrate, respectively. For simplicity, we consider a geometrically two-dimensional flow. Let $t$ be the time, $(x, z)$ be the coordinates and $(g^x, g^z)$ the gravity accelerations along and perpendicular to the slope, respectively. Let, $h$ and $u$ be the flow depth and the mean flow velocity along the slope. Similarly, $\gamma^m, \alpha^m, \mu^m$ be the density ratio between the fluid and the particles $\left( \gamma^m = \rho_f^m / \rho_s^m \right)$, volume fraction of the solid particles (coarse solid and fine solid particles is represented by a single solid-phase: $\alpha^m := \alpha_s + \alpha_{fs}$, and $\rho^m := \rho_s^m$ is the effective density of the solid in the mixture), and the basal friction coefficient $\left( \mu^m = \tan \delta^m \right)$ related to the basal friction angle $\delta^m$, in the mixture. Analogously, $\rho^b = \rho_s^b$ is the density of the solid, $\gamma^b = \rho_f^b / \rho_s^b$ is the density ratio between the fluid and the particles, and $\alpha^b$ is the solid volume fraction in the bed material. $\mu^b = \tan \delta^b$ is the Coulomb friction parameter associated with the erodible bed with the friction angle $\delta^b$. Note that differences in $\mu^m$ and $\mu^b$ can result due to the difference in the landslide material and the bottom material. In simple situation, $\mu^m$ and $\mu^b$ can have similar values[27]. Furthermore, $b = b(x; t)$ is the erodible bed surface that evolves in space and time, $E$ is the basal erosion rate, $u^b$ is the erosion velocity, $K$ is the earth pressure coefficient for the solid (composed of coarse solid and fine solid) particles in the mixture as a function of the internal ($\phi^m$) and the basal ($\delta^m$) friction angles, and $C_{DV}$ is the viscous drag coefficient.

Next, we define some (mostly) fundamentally new mechanical concepts and terminologies as follows. This will help to comprehensively streamline the paper. Erosion: a mechanical process by which the bed material is mobilized by the flow. Entrainment: a mechanical process by which the eroded material is incorporated (entrained) and taken along with by the flow. Erosion-velocity: $u^e := u^b$, the velocity of the eroded material from the basal substrate. Entrainment-velocity: $u^{ev} := u - u^b$, the velocity of the entrained mass, or the velocity of the landslide minus the velocity of the eroded mass. Energy-velocity:

$u^{env} := u^b - (u - u^b) = u^e - u^{ev}$, the velocity associated with the net momentum production due to erosion that generates the excess (kinetic) energy, or the erosion velocity in excess to the entrainment velocity. Energy: associated with the erosion-induced net momentum production that results in the excess (kinetic) energy. Energy-generator (or, the mobility generator): the parameter that generates the excess energy (or the mass flow mobility) due to erosion. Energy-budget: the state of available excess energy generated by the net momentum production due to erosion. Erosion-rate: $E = -\partial b / \partial t$, the negative time rate of change of the basal topography, or the time rate of change of mass resulting in the mass production. The mechanical significance and the mathematical structures of these concepts and terminologies will be clearer in the due places when they appear in the text.

The Eulerian and Lagrangian descriptions are adopted here by purpose that will be clear in due places. First, we write the existing balance equations in standard and usual Eulerian form. Then, we utilize the most simple and convenient discrete procedure for the correct handling of the true inertia of the landslide with erosion. Classically, similar approaches have been used in deriving rocket equations. The Lagrangian descriptions are the most simple form that clearly demonstrate how erosion can result in substantially increased landslide velocity leading to enhanced mobility. Finally, the full set of erosional landslide model equations are written in the Eulerian description in the more sophisticated form that was largely facilitated by the Lagrangian descriptions.

We begin with the mass and momentum balance equations. Consider the multi-phase mass flow model[61] and include the viscous drag and erosion[27]. For simplicity, we assume that the relative velocity between the coarse and fine solid particles ($u_s, u_{fs}$) and the fluid phase ($u_f$) in the debris material is negligible, that is, $u_s \approx u_{fs} \approx u_f = u$, and so is the viscous deformation of the fluid. Furthermore, the coarse solid and the fine solid particles are assumed to have similar physical properties and constitute the solid volume fraction in the mixture. Thus, the mixture is composed of the solid $\left( \alpha_s + \alpha_{fs} = \alpha^m \right)$ and the fluid fraction $\left( \alpha_f \right)$ such that $\alpha_s + \alpha_{fs} + \alpha_f = \alpha^m + \alpha_f = 1$. This reduces the situation to the effectively two-phase model[27] representing the motion of the mixture consisting of the solid ($\alpha^m$) and the fluid ($\alpha_f$) phases. To facilitate for the derivation of a simple model, with these assumptions, we are considering an effectively single-phase mixture flow. Because of the effectively single-phase nature of the model being developed here, either we can consider an extra closure relation for $\alpha^m$, or parameterize it, or consider it as a constant. To keep the present basic model simple, we have assumed locally uniform mixture. Then, by summing up the mass and momentum balance equations[27,61], we obtain a single mass and momentum balance equation describing the motion of an erosive landslide and the evolution of the erodible bed surface, $b$ as:

$$\frac{\partial h}{\partial t} + \frac{\partial}{\partial x}(hu) = E, \tag{1}$$

$$\begin{aligned}
\frac{\partial}{\partial t}(hu) &+ \frac{\partial}{\partial x}\left[ hu^2 + (1 - \gamma^m)\alpha^m g^z K \frac{h^2}{2} \right] \\
&= h\left[ g^x - \frac{u}{|u|}(1 - \gamma^m)\alpha^m g^z \mu^m - g^z\{1 - (1 - \gamma^m)\alpha^m\}\frac{\partial h}{\partial x} - g^z \frac{\partial b}{\partial x} - C_{DV} u^2 \right] \\
&+ u^b E,
\end{aligned} \tag{2}$$

$$\frac{\partial b}{\partial t} = -E, \tag{3}$$

where $E$ and $u^b E$ are the mass and momentum productions, respectively, and $-(1 - \alpha^m) g^z \partial h / \partial x$ emerges from the hydraulic pressure gradient associated with the possible interstitial fluid in the landslide. Moreover, the term containing $K$ on the left hand side, and the right hand side in the momentum equation (2) represent all the involved forces. The first term in the square bracket on the left hand side of (2) describes the advection, while the second term describes the extent of the local deformation that stems from the hydraulic pressure gradient of the free-surface of the flow. The first, second, third, fourth, fifth and the sixth terms on the right hand side of (2) are forces due to the gravity acceleration; effective Coulomb friction which includes lubrication due to the buoyancy $(1 - \gamma^m)$, liquefaction due to the solid volume fraction $(\alpha^m)$, and the factor $\frac{u}{|u|}$ indicates the Coulomb friction force acting against the motion; the term associated with buoyancy and the fluid-related hydraulic pressure gradient; the force due to the spatial variation of the basal surface; viscous drag and the erosion-induced momentum production, respectively. Equation (3) makes a direct connection between the erosion process and the evolution of the bed by updating the basal topography with erosion, which in turn, explains how the changes in the bottom topography affect the mobility of the landslide. By setting $\gamma^m = 0$ and $\alpha^m = 1$, we obtain a pure dry granular flow or an avalanche motion. For this choice, the third term on the right hand side of (2) vanishes. However, we keep $\gamma^m$ and $\alpha^m$ also to include different aspects of possible fluid effects in the landslide.

In simple situation, the bottom is defined with the locally inclined $x$ axis. This is a standard procedure involving erosion along the slope. To make the model more general, following Pudasaini and Mergili[61], we have also included the term $-g^z \partial b / \partial x$ in the momentum balance (2) that incorporates the effect of the local variation of the topography (on top of the $x$ axis) on the flow dynamics. This superposition accounts for the detailed topographic information on top of the mean slope. Moreover, the time evolution of the basal surface due to erosion is given by $\partial b / \partial t = -E$ in (3).

For the purpose of developing a simple landslide mobility equation, momentarily we consider the motion down an inclined slope. In this situation $\frac{u}{|u|} = 1$ and $-g^z \frac{\partial b}{\partial x} = 0$. Later, we will again restore these terms while presenting the full model equations. Due to the uniform mixture assumption, the local spatial variation of $\alpha^m$ can be ignored. However, it has been restored while constructing the full model at the end of this section. Then, the momentum balance equation (2) can be re-written as:

$$h \left[ \frac{\partial u}{\partial t} + u \frac{\partial u}{\partial x} \right] + u \left[ \frac{\partial h}{\partial t} + \frac{\partial}{\partial x} (hu) \right]$$
$$= h \left[ g^x - (1 - \gamma^m) \alpha^m g^z \mu^m - g^z \left\{ ((1 - \gamma^m) K + \gamma^m) \alpha^m + (1 - \alpha^m) \right\} \frac{\partial h}{\partial x} - C_{DV} u^2 \right]$$
$$+ u^b E.$$
(4)

Note that for $K = 1$ (which mostly prevails for extensional flows[62]), the third term on the right hand side associated with $\partial h / \partial x$ simplifies drastically, because $\left\{ ((1 - \gamma^m) K + \gamma^m) \alpha^m + (1 - \alpha^m) \right\}$ becomes unity. This also indicates that by assuming the isotropic mixture $(K = 1)$ one loses some important information about the solid content and the buoyancy effect.

We eliminate the existing erroneous perception on erosive landslide. As entrainment introduces new mass into the system the inertia is increased. One might simply think that the expression $u \left[ \frac{\partial h}{\partial t} + \frac{\partial}{\partial x} (hu) \right]$ on the left hand side in the momentum equation (4) can be replaced by $uE$, where the flow velocity $u$ is multiplied by the erosion rate (the time rate of change of mass resulting in the mass production), $E$ from the mass balance (1). However, here, one must be very cautious. The velocity associated

with this increased mass must be handled carefully. In reality, the erosion-induced produced mass (with the rate $E$) is not transported by the flow velocity $u$ itself but, it is transported by a fundamentally different velocity that can be substantially lower than $u$. As we will see later, this newly revealed fact becomes a game-changer and makes a breakthrough in correctly determining the state of energy (or, momentum) and thus the mobility associated with an erosive landslide. Below, we derive a physically correct momentum balance equation for an erosional landslide and prove that the direct substitution $u \left[ \frac{\partial h}{\partial t} + \frac{\partial}{\partial x} (hu) \right] = uE$ in the inertial part of (4) results in a physically wrong momentum balance equation. This appears from an erroneous understanding of the erosional landslide, but prevails in many existing models, see, for example[35,37,63].

**Correct derivation of the relevant momentum balance equation.** We derive a simple basic erosional landslide equation. However, the most elegant derivations that do not require any conditions have been presented in the Supplementary Information. The situation of an erosive landslide is as follows. Let at time $t_1 = t$ the landslide of mass $m$ moves with the velocity $u$ and the eroded mass $\Delta m$ that has just been mobilized a moment ago moves with the erosion velocity $u^b$ in which $u^b < u$. At the later time $t_2 = t + \Delta t$, the landslide with mass $m$ strikes the little mass $\Delta m$ with a perfectly inelastic collision, which is natural to happen. Consequently, the mass $\Delta m$ is embedded in the landslide resulting in the entrainment. At this time, the total of the landslide mass and the entrained mass $(m + \Delta m)$ moves together with a single velocity $(u + \Delta u)$, where $\Delta u$ is the increment in the landslide velocity $u$. So, in the frame of reference of a stationary observer, the momentum $P_1$ at time $t_1$ and the momentum $P_2$ at time $t_2$, respectively, are:

$$P_1 = m u + \Delta m u^b, \qquad (5)$$

and

$$P_2 = (m + \Delta m)(u + \Delta u). \qquad (6)$$

Conservation of linear momentum states the following relation incorporating all the forces $F$ including the forces applied to the landslide and the entrained mass:

$$F = \lim_{\Delta t \to 0} \frac{P_2 - P_1}{\Delta t}. \qquad (7)$$

Since

$$P_2 - P_1 = mu + m\Delta u + u\Delta m + \Delta u \Delta m - mu - u^b \Delta m$$
$$= m\Delta u + \Delta u \Delta m + (u - u^b) \Delta m \approx m\Delta u + (u - u^b) \Delta m, \qquad (8)$$

we have now the formally and correctly derived momentum equation for an erosional landslide:

$$F = m \frac{du}{dt} + u^{ev} \frac{dm}{dt}, \qquad (9)$$

in which, the higher order term $\Delta u \Delta m \ll 1$ is ignored and $u^{ev} = u - u^b$ is the entrainment-velocity. Moreover, $dm/dt$ is positive. We call (9) the (basic) erosional landslide equation. One may derive a similar equation for the depositional landslide.

Equation (9) can be obtained in many different ways (see the Supplementary Information). All three derivations, one presented above, and the two in the Supplementary Information, lead to the same result. This proves that we are physically and mathematically fully consistent.

It is important to note that $P_2 - P_1 = m\Delta u + (u - u^b) \Delta m$ is the main structure that any physically correct derivation must produce for the erosional landslide. This is clear from the derivation presented above and the two alternative derivations presented in the Supplementary Information. Moreover, at this

point, it is crucial to realize, that the momentum equation (9) for the erosional landslide must be derived rigorously as done here by following the first-principles, but cannot just be speculated arbitrarily.

The fundamental understanding here, as revealed by (9) is, that the increased inertia due to the increase in the mass of the landslide is not related to the velocity of the landslide $u$, but it is associated with the entrainment velocity, $u^{ev}$. Depending on the erosion situation, that we will discuss later, $u^{ev}$ can be substantially less than the landslide velocity $u$. Thus, the true increased inertia $u^{ev} dm/dt$ can be much less than incorrectly proposed previously, $u\, dm/dt$. Hence, the classical, or the direct representation of $F = \frac{d}{dt}(mu)$ as $F = m\frac{du}{dt} + u\frac{dm}{dt}$ is fundamentally wrong for erosional situation. This led many to the erroneous conclusion: that either erosion results only in reduced mass flow mobility, because the landslide consumes more energy resulting in the reduced mobility of the erosive landslide, or that erosion does not change the mass flow mobility as the energy loss in entrainment is balanced by the produced momentum. Later, we prove that, in general, both conclusions are mechanically incorrect.

In special situation, when the eroded mass enters the landslide with almost the velocity of the landslide itself, then $u^{ev} \approx 0$, and there is (almost) no increase (change) of inertia. This can happen if the basal substrate is very weak. Examples include a fully saturated or, liquefied bed material[37] such that with almost no consumption of energy, the basal substrate can be eroded. However, in a particular situation, if the substrate is so strong mechanically that the erosion hardly takes place, and even if it takes place, the erosion velocity $\left(u^b\right)$ can be as low as zero, only then, the classical approach might seem to be applicable. Which, effectively means, that the classical approach works only for non-erosional situation, but not for landslide with erosion. So, those models, which are based on the unphysical formulation of the momentum balance, for example, refs. [35,37,63], are not appropriate in simulating landslide motion with erosion.

This implies, that the correct consideration of inertia is crucial for the precise derivation of the dynamical landslide model with erosion. However, it is evident, that the law (9) cannot be obtained directly by rearranging the inertial terms in the Newton's second law of motion, but rather must be derived carefully by correctly considering the conservation of momentum for an erosional landslide as done above. Those erosion models that are based on the direct use of the Newton's second law of motion with regard to the inertial part of the momentum balance equation cannot represent the true mechanism of erosion and the subsequent dynamics.

We call (9) the landslide-rocket-equation. In the form, (9) is similar to the famous Tsiolkovsky Rocket-Equation[64]. However, there are fundamental differences. First, the way we derive the model is different. Second, the mass of the rocket is decreasing (since it consumes fuel), so $dm/dt$ is negative. But, for erosional landslide $dm/dt$ is positive as the mass of landslide is increasing. Third, although the multiplier of $dm/dt$ is positive for both the erosional landslide and the rocket, they have quite different perspectives and mechanisms. For the rocket, it is the velocity of the exhaust, say $u^{ex}$. But, for the erosional landslide, it is the velocity of the landslide minus the velocity of the eroded mass that is entrained by the landslide. Thus, depending on the magnitude of the erosion velocity, the entrainment velocity $u^{ev}$ can be substantially less than the landslide velocity, as the velocity of the eroded particle, that is entrained by the landslide, is a positive quantity that, depending on the situation (the flow and the bed morphology), can be as high as the velocity of the landslide itself. Further detail on it can be found in Supplementary Information.

**The landslide mobility equation: a novel model formulation.** Since, in general, the erosion velocity cannot exceed the landslide velocity[27], the entrainment-velocity $u^{ev} = u - u^b$ is a nonnegative quantity. For convenience, we write (9) in terms of $u - u^b$:

$$m\frac{du}{dt} + \left(u - u^b\right)\frac{dm}{dt} = F, \tag{10}$$

where $m$ is the landslide mass. Now, we can compare (10) with (4), which is written in the depth-averaged form and for a constant mass density. So, without loss of generality, we can carefully, and consistently set mass (per unit channel length) $m = h$ (because the material density cancels out in the momentum balance (2) or (4) under consideration, so we should take $m = h$ instead of $m = \rho^m h$) resulting in $\frac{dm}{dt} = \frac{dh}{dt} = \frac{\partial h}{\partial t} + \frac{\partial}{\partial x}(hu) = E$, which is (1), and the material derivative $\partial u/\partial t + u\partial u/\partial x = du/dt$, yielding

$$\frac{du}{dt} = g^x - (1 - \gamma^m)\alpha^m g^z \mu^m - g^z\left[\left((1 - \gamma^m)K + \gamma^m\right)\alpha^m + (1 - \alpha^m)\right]\frac{\partial h}{\partial x}$$
$$- C_{DV}u^2 + (2u^b - u)E\frac{1}{h}, \tag{11}$$

where, out of $2u^b E$, one $u^b E$ already exists in the force terms in $F$ that entered as momentum production[27], as seen in (2); however, the other $u^b E$ emerges from the correct handling of the erosion-induced changed inertia. We can draw an important conclusion from (11): since for erosion $E > 0$, whether the erosion-related mass flow mobility will be enhanced, reduced or neutralized (remains unaltered) depends exclusively on whether $\left(2u^b - u\right) > 0$, $\left(2u^b - u\right) < 0$, or $\left(2u^b - u\right) = 0$. This has been exclusively elaborated in the following sections.

Equation (11) can be cast in different forms. Following Pudasaini and Fischer[27], we can write $u^b = \lambda^b u$, where $\lambda^b$ is the erosion drift (associated with the erosion velocity). So, (11) reduces to

$$\frac{du}{dt} = g^x - (1 - \gamma^m)\alpha^m g^z \mu^m - g^z\left[\left((1 - \gamma^m)K + \gamma^m\right)\alpha^m + (1 - \alpha^m)\right]\frac{\partial h}{\partial x}$$
$$- C_{DV}u^2 + (2\lambda^b - 1)E\frac{u}{h}. \tag{12}$$

The closure for the erosion drift and its influence in landslide mobility has been presented later.

For the full and better simulation of the erosive landslide, we must numerically integrate (12) together with (1) that includes evolution of both the flow velocity and the flow depth. This will be discussed later. Here, we are mainly interested in developing a simple model that can be solved analytically to highlight the main essence of erosion-induced energy (momentum) and the associated mobility of the landslide in terms of its velocity.

Further simplification of (12) is possible. For simplicity, we can parameterize (mainly in space, see later (18), the main model) the landslide (or, the flow) depth $h$, and write (12) as

$$\frac{du}{dt} = \mathcal{A} - \mathcal{C}u^2 + (2\lambda^b - 1)E\frac{u}{h}, \tag{13}$$

where, $\mathcal{A} = g^x - (1 - \gamma^m)\alpha^m g^z \mu^m - g^z\left[\left((1 - \gamma^m)K + \gamma^m\right)\alpha^m + (1 - \alpha^m)\right]\frac{\partial h}{\partial x}$ takes into account the topography induced downslope component of gravity, the first term; effective basal friction including the buoyancy reduced normal load and lubrication, the second term; and the force due to the free-surface pressure gradient of the landslide

(including the possible presence of the interstitial fluid), the third term, which, depending on the negative or positive slope of the landslide, will enhance or reduce the motion[62]. As mentioned earlier, for extensional flows, $K \approx 1$, so $\left[\left((1-\gamma^m)K + \gamma^m\right)\alpha^m + (1-\alpha^m)\right]$ reduces to unity. Moreover, $\mathcal{C} = C_{DV}$ is the viscous drag coefficient. Equation (13) can be written in the simple form

$$\frac{du}{dt} = \mathcal{A} + \mathcal{B}u - \mathcal{C}u^2, \qquad (14)$$

where $\mathcal{B} = (2\lambda^b - 1)E/h$. Equation (14) can be solved exactly.

One can apply any erosion rate $E$ in the above equations. As in Pudasaini and Fischer[27], we consider the drift factor $\lambda^m$ that is associated with the velocity of the particle in the debris mixture at the lowest level, $u^m$, with the mean velocity of the flow, $u$; the relation $u^m = \lambda^m u$. Following the mechanical erosion-rate model[27]:

$$E = \frac{g \cos\zeta \left[(1-\gamma^m)\rho^m\mu^m\alpha^m - (1-\gamma^b)\rho^b\mu^b\alpha^b\right]}{(\rho^m\lambda^m\alpha^m - \rho^b\lambda^b\alpha^b)}\left(\frac{h}{u}\right), \quad (15)$$

(13) can be written as:

$$\frac{du}{dt} = \mathcal{A} - \mathcal{C}u^2 + (2\lambda^b - 1)E^P, \qquad (16)$$

with

$$E^P = \frac{g \cos\zeta \left[(1-\gamma^m)\rho^m\mu^m\alpha^m - (1-\gamma^b)\rho^b\mu^b\alpha^b\right]}{(\rho^m\lambda^m\alpha^m - \rho^b\lambda^b\alpha^b)}, \qquad (17)$$

where for dry flows and substrate, $\alpha^m$ and $\alpha^b$ are unity, otherwise these must be parameterized or closed. Furthermore, as for the sliding mass, the parameters are considered analogously for the erodible basal substrate as indicated by the superscript $b$. Differences in the material parameters across the erosion interface (between the landslide and the bed material) results in erosion[27]. Furthermore, with entrainment, values of $\rho^m, \gamma^m, \mu^m$, and $\alpha^m$ should be appropriately updated in proportion to the newly entrained material to the sliding material. However, in the present modeling frame, such updating can be achieved only through their parameterizations. For example, $\rho^m$ cannot be considered as a full variable as it adds complications in the simplification of the mass and the momentum balance equations (1) and (2). So, consideration of $\rho^m$ as a state variable, or its time and spatial variation, is out of scope here. We call $E^P$ the erosion parameter, which as given by (17), incorporates many essential physical and mechanical aspects involved in erosion, and explicitly determines the erosion intensity. The great advantage of (16) is that the erosion-enhanced flow mobility can now be explicitly evaluated in terms of velocity, as all the quantities (except $u$) on the right hand side of (16) are measurable, or given. This is the first-ever physics-based model to do so. Thus, it has enormous application potential.

It is now so convenient that (16) can be simply written as

$$\frac{du}{dt} = (\mathcal{A} + \mathcal{P}_M) - \mathcal{C}u^2, \qquad (18)$$

where $\mathcal{P}_M = (2\lambda^b - 1)E^P$ is the overall mobility parameter (the erosion-induced net momentum production per unit depth or the force per unit mass) that quantifies the total erosion-related enhanced mass flow mobility by amplifying the landslide acceleration. We call (18) the landslide mobility equation, which can be solved analytically to obtain the landslide velocity with erosion.

Since $u^b = \lambda^b u$, the erosion velocity is associated with the parameter $\lambda^b$. The form of $E^P$ in (17) contains no odds. First, in reality, $\lambda^b$ lies in a close or broader neighborhood of 1/2 that is contained in (0, 1). So, the legitimate values of $\lambda^b$ is around 1/2.

This has been proven and explicitly explained in Pudasaini and Fischer[27]. Second, mechanically, the erosion velocity is controlled by the net shear stress (applied by the flow minus resisted by the bed material). This means, the manner by which $\lambda^b$ changes is controlled by the numerator or the net shear stress. In other words, in connection to the erosion drift equation (see below), in total, the higher value of $\lambda^b$ usually corresponds to the higher mobility parameter $\mathcal{P}_M$.

**The state of energy and mobility of an erosive landslide.** Mobility, perhaps, is the most important aspect in landslide modeling as it is the direct measure of the threat posed by the landslide, and is simply associated with its excessive volume (or, mass), enormous impact force, the exceptional travel distance, and velocity, and the widespread inundation area. Mobility is governed by the state of energy of the landslide and is expressed in terms of the landslide velocity together with the erosion velocity. So, here, we focus on landslide energy budget. The state of mobility is associated with the sign of $(2\lambda^b - 1)$ and is amplified by the factor $E^P$ in $\mathcal{P}_M = (2\lambda^b - 1)E^P$ in (18). We call $(2\lambda^b - 1)$ the energy generator (or the mobility generator), and write as $\mathcal{P}_{M_{eg}} = (2\lambda^b - 1)$, the parameter that generates the excess mass flow mobility due to erosion. Note that the (excess) energy refers to the net momentum production due to erosion. This will be explained later. Mass flow mobility (or the velocity) will be enhanced, reduced or remains unchanged depending on whether $(2\lambda^b - 1) > 0$, $(2\lambda^b - 1) < 0$, or $(2\lambda^b - 1) = 0$. As $E^P$ determines the erosion magnitude, it is of utmost importance to systematically analyze $(2\lambda^b - 1)$, because this will tell us the state of mobility (associated with the sign), and how the erosion is amplified (its magnitude) that ultimately regulates the strength and consequence of erosion as measured by the landslide velocity. This is how the energy-generator changes the game and fully controls the mobility of the erosive landslide.

In general, $\lambda^b$ may take any value in the domain (0, 1). However, in solving some engineering and applied problems, we need to physically constrain $\lambda^b$. There can be different possibilities for this, but Pudasaini and Fischer[27] provide a physical model for $\lambda^b$ by presenting an analytical erosion drift equation:

$$\lambda^m = \left(1 + \frac{\rho^b}{\rho^m}\frac{\alpha^b}{\alpha^m}\right)\lambda^b. \qquad (19)$$

As for $\lambda^b$, $\lambda^m$ also takes the values in the domain (0, 1). However, in general, as proven by (19), the velocity of the eroded particle cannot be larger than the velocity of the particle at the flow bottom, we have the constrain $0 < \lambda^b < \lambda^m < 1$. As discussed in Pudasaini and Fischer[27], the drift equation (19) is mechanically rich. Following the exclusive consideration of the shallow flow models in the literature[8,62], we may simplify the situation by assuming the plug flow which implies that $\lambda^m \approx 1$. Now, it becomes mechanically very interesting to analyze the landslide mobility with (19). Below, we consider three special situations, with respect to the inertial number, $N_i = \rho^b\alpha^b/\rho^m\alpha^m$.

Inertially neutral erodible bed substrate results in no change in energy and mobility. For this, the inertia of the bed material is equal to the inertia of the material in the flow: $\rho^b\alpha^b = \rho^m\alpha^m$. Thus, we obtain $\lambda^b = 1/2\lambda^m$, which implies that $\lambda^b = 1/2$, and $2\lambda^b - 1 = 0$. In this situation, there is no gain or loss of momentum or energy, and thus, the landslide mobility remains unchanged even for the erosive landslide. This is a very special situation, however, less likely to occur in nature. Moreover, note that the value $\lambda^b = 1/2$ that distinguishes between increased or

reduced mobility arises from the term inside the brackets of (19) when $\rho^b \alpha^b = \rho^m \alpha^m$.

Inertially weaker erodible bed substrate results in enhanced energy and mobility. The inertia of the material in the bed can be lower than the inertia of the material in the flow: $\rho^b \alpha^b < \rho^m \alpha^m$. Then, we obtain $\lambda^b > 1/2\lambda^m$, which implies that $\lambda^b \in (1/2, 1)$, and $(2\lambda^b - 1) \in (0, 1)$. In other words, this is the situation in which the change in inertia of the landslide in incorporating the inertially (mechanically) weaker material is less than its change in inertia if it would have incorporated inertially equally stronger material. This suggests that the erosion-induced gained momentum or energy results in the enhanced-mobility of the erosive landslide.

Inertially stronger erodible bed substrate results in reduced energy and mobility. The inertia of the bed material can be higher than inertia of the material in the flow: $\rho^b \alpha^b > \rho^m \alpha^m$. This implies $\lambda^b < 1/2\lambda^m$, and $\lambda^b \in (0, 1/2)$. Thus, $(2\lambda^b - 1) \in (-1, 0)$. So, for this, the change in inertia of the landslide in incorporating the inertially stronger material is higher than its change in inertia if it would have incorporated inertially equally stronger material. This implies the erosion-induced momentum loss or energy loss, and results in the reduced mobility even for the erosive landslide.

The above conditions can be unified into a single frame: $(2\lambda^b - 1) \in (-1, 1) = (-1, 0) \cup \{0\} \cup (0, 1)$ for $\lambda^b \in (0, 1) = (0, 1/2) \cup \{1/2\} \cup (1/2, 1)$, covering the whole spectrum of momentum or energy loss, or equilibrium, or gain, that result in reduced, neutral, or enhanced landslide mobility. With the knowledge of the energy generator $\mathcal{P}_{M_{eg}} = (2\lambda^b - 1)$, the involved net energy in landslide erosion can now be quantified from the mobility parameter $\mathcal{P}_M$. Such an explicit and fully mechanical description of the state of energy (or, momentum) and the associated mobility of an erosive landslide is seminal.

As $\lambda^b$ is related to $1/2\lambda^m$ and $\lambda^m \approx 1$, following the analysis in Pudasaini and Fischer[27], and the above discussion, technically suitable natural domain of $\lambda^b$ is: $(-\Delta\lambda + 1/2, 1/2 + \Delta\lambda)$, where $\Delta\lambda$ is a small positive number, say, typically 1/4, such that the value of $\lambda^b$ is always contained in (0, 1). The drift factor $\lambda^b$ is more likely to approach 1 rather than to 0 indicating the energy gain than the energy loss in erosion. The range of values of $\lambda^b$ and $\Delta\lambda$ may depend on the erosion situation. This is a technical aspect that needs to be properly handled during the model application to laboratory and/or field data. The drift equation (19) provides a practically useful mechanical closure for $\lambda^b$, and thus for $\Delta\lambda$.

**The role of erosion velocity in mass flow mobility.** In many of the previous erosion models, the velocity of the eroded mass has been set to zero, or it does not appear at all. For the first time, Pudasaini and Fischer[27] rigorously proved with a mechanical erosion model that setting the erosion velocity to zero is physically incorrect. In this line, the above analysis clearly expands our understanding of erosion related phenomena and shows that, whether the erosive landslide will gain or lose (or, remain unchanged) energy, or in other words, whether it will have enhanced or reduced (or, neutral) mobility as compared to the non-erosive one, primarily depends on the velocity of the eroded mass $u^b$. In technical terms, it depends on the value of the drift factor $\lambda^b$ explaining how big is the erosion velocity with respect to the flow velocity. Erosion velocity closer to the flow velocity results in the most mobile flow. Because, in this situation, the momentum production $(u^b E)$ due to the reduced friction in erosional situation overtakes the momentum loss due to the increased inertia $(u - u^b)E$. This is how most probably it happens in nature for erosive landslides. As $u^b \to u$, the increase in

inertia associated with the entrainment tends to vanish. Then, the flow attains the highest gain in energy resulting in the highest mobility, as measured by the gained or produced momentum $(u^b E)$ of the flow due to erosion. This analysis clearly reveals the fact, that paired with the momentum production and correct handling of the change of inertia in describing the erosion-related energy, the erosion velocity plays a key and outstanding role in appropriately determining the energy budget and, thus the mobility of an erosive landslide.

Moreover, if the erosion velocity is less than one-half of the flow velocity then, the landslide loses energy. This results in reduced mobility. The highest energy is consumed in erosion if the erosion velocity is much smaller than the flow velocity, this means when the erosion velocity is almost zero. In this situation, $(-uE)$ is the reduced momentum, which is produced by the increased inertia due to entrainment when the entrained mass enters the landslide with zero velocity. Physically, this is impossible as proved by Pudasaini and Fischer[27], as $\lambda^b \neq 0$, however, this refers to the situation in many existing and influential erosion models[25,35,37,46,47,63].

Interestingly, the erosion will not change the energy status, and thus the mobility, of the landslide if the erosion velocity is one-half of the flow velocity. Such a special situation has also been mentioned in refs. [8,27], which, however, is very restricted, and less likely to happen in nature. So, the present paradigm further enhances the mechanical erosion model by Pudasaini and Fischer[27] and offers a complete and legitimate model for erosive landslide.

**Erosion-, entrainment-, energy-velocity: new mechanical concepts.** Here, we formally introduce three important concepts with their underlying mechanics. These are: (i) the erosion-velocity, $u^e = u^b$, (ii) the entrainment-velocity, $u^{ev} = u - u^b$, and (iii) the energy-velocity, $u^{env} = u^b - (u - u^b)$. We call these, the-three-$E$ mechanical concepts. While the erosion velocity was first introduced by Pudasaini and Fischer[27], the entrainment velocity and the energy velocity are completely new concepts. In fact, $u^e$, $u^{ev}$, and $u^{env}$ already appear in previous sections. However, these concepts are systematically collected here for the better logical sequence and structural reasons. This comprehensively helps to distinguish between erosion and entrainment, and to formally delineate different energy or mobility regimes that will be explained later.

Understanding the difference between erosion and entrainment is important. The existing literature could not distinguish between the erosion and entrainment as these terms are used interchangeably. However, here, we have made it very clear with the mechanical expressions, that the erosion and entrainment are essentially different phenomena. Erosion is a process by which the bed material is mobilized by the flow with the velocity $u^e = u^b$, while entrainment is intrinsically another process by which the eroded material is incorporated (entrained) and taken along with by the flow with the velocity $u^{ev} = u - u^b$. This fundamentally enhances our understanding of basic, but different processes in erosion-related phenomena in landslide by clearly defining, and distinguishing the mechanisms of erosion and entrainment. These are important novel aspects.

Erosion velocity and entrainment velocity systematically appear in the fundamentally derived momentum equation (9). The erosion velocity $(u^b)$ enters the momentum equation (2) through the boundary conditions[27,62] applied to the erodible interface, that combined with the erosion rate ($E$), produces the erosion-induced momentum production, $u^b E$ in (2). This process is induced due to the mobilized bed material. Whereas the entrainment velocity $(u - u^b)$ appears fundamentally differently

due to the correct derivation of the relevant momentum balance equation as clearly revealed by (9) that entrains the newly eroded and added material associated with $\frac{dm}{dt}$ with the erosion velocity $u - u^b$ producing the term $\left(u - u^b\right) \frac{dm}{dt} = u^{ev} \frac{dm}{dt}$. So, in (9), $F$ in the left contains $u^b E$ that is produced by one process, but $u^{ev} \frac{dm}{dt}$ emerges on the right that is generated by completely another process. Hence, it is structurally and mechanically clear, that the entrainment velocity, as given by the relative velocity, $u - u^b$, in fact, represents the velocity of the entrained mass associated with $\frac{dm}{dt}$.

The perception and structure of the energy-velocity is important. With the erosion rate $E$, for the landslide mass $m(= \rho^m h)$ (per unit flow length), the excess kinetic energy generated by $u^{env}$ can be realized as $\mathcal{E}^{env} = \frac{1}{2} m u^{env} E$ which has the dimension of energy, because both $u^{env}$ and $E$ have the dimension of velocity. An expression similar to this can be obtained from the erosion-induced net momentum production, the last term on the right hand side of (11), which, when multiplied by the landslide mass $\left(m = \rho^m h\right)$ results in $\mathcal{E}_h^{env} = \left(\rho^m h\right)\left(2u^b - u\right)E \frac{1}{h} = m\left(2u^b - u\right)E \frac{1}{h} = m u^{env} E \frac{1}{h}$. So, we can write, $\mathcal{E}_h^{env} = 2\mathcal{E}^{env} \frac{1}{h}$, which is twice the erosion-induced (excess) kinetic energy (resulting from $u^{env}$) normalized by the flow depth. This physically justifies the use of the term energy-velocity for the expression $u^{env}$, because it generates the erosion-induced excess energy and has the dimension of velocity. Note that depending on the sign of $u^{env}$, $\mathcal{E}_h^{env}$ can be positive (for $u^{env} > 0$, or $\lambda^b > 1/2$) or negative (for $u^{env} < 0$, or $\lambda^b < 1/2$) resulting in the enhanced or the reduced mobility of the landslide. This has further been explained below. The energy-velocity $u^{env} = u^b - \left(u - u^b\right)$ is associated with the total (net) momentum production, with contribution $u^b$ emerging from the reduced friction and $-\left(u - u^b\right)$ from the changed (reduced) inertia. It plays an exclusively unique and dominant role in formulating the mobility equation and in determining the state of energy. Thus, the energy-velocity provides the universal picture of the erosion-induced mobility.

To deal with the more sophisticated situation we might also include the energy balance equation to complement the mass and the momentum balances. However, for now, we begin with the first-ever simple mechanical model capable of describing the excess energy of the landslide associated with erosion, where erosion induces the net momentum production giving rise to the excess kinetic energy.

Now, we can delineate different energy regimes. The energy velocity, $u^{env} = u^b - \left(u - u^b\right) = u^e - u^{ev}$ constitutes the state of energy or the energy budget and clearly delineates the three energy regimes associated with the erosive landslide: The landslide energy remains unchanged if the energy-velocity is zero, $u^{env} = 0$. The landslide gains energy in erosion if the energy-velocity is positive, $u^{env} > 0$. The landslide loses energy even in erosion if the energy-velocity is negative, $u^{env} < 0$. In terms of $\lambda^b$, these regimes correspond to $\lambda^b = 1/2$, $\lambda^b > 1/2$, and $\lambda^b < 1/2$, respectively. So, the energy, and thus the mobility, of an erosive landslide is fully controlled by the erosion velocity. This signifies the prime role of erosion velocity in correctly determining the state of erosive landslide.

We can now explain the novel mechanism of landslide-propulsion and erosion-thrust. With the definitions, their mechanics and the discussions in the previous sections, we draw a central conclusion: the landslide gains energy, and thus enhances its mobility if the energy-velocity is positive, specifically, if the erosion velocity is greater than the entrainment velocity, i.e., $u^e > u^{ev}$. We call this phenomenon the landslide-propulsion,

emerging from the net momentum production, that provides the erosion-thrust to the landslide. This means, if the erosion velocity is greater than one-half of the flow velocity, i.e., $u^b > u/2$, the mobility is enhanced. This is equivalent to the condition $\lambda^b > 1/2$. In other words, the landslide gains energy to enhance its mobility if the eroded material is easily entrainable with the velocity lower than the erosion velocity. These are quite natural phenomena but revealed here for the first time.

**Analytical solution to the landslide mobility.** Now, we construct an exact analytical solution. The landslide mobility equation (18) can be solved analytically to explicitly obtain the landslide velocity. Exact analytical solutions to simplified cases of non-linear debris avalanche model equations are necessary to calibrate numerical simulations[65]. These problem-specific solutions provide important insights into the full behavior of the system. A physically meaningful exact solution explains the true and entire nature of the problem associated with the model equation[51–53,66]. However, numerical solutions are always subject to questions as such solutions are based on some assumptions and applied approximations that may not follow the laws of nature. So, in general, the physically relevant exact solutions are superior over the numerical simulations[67]. Nevertheless, the numerical solutions can cover the broad spectrum of the complex flow dynamics, and once tested and validated against the analytical solutions, may provide even more accurate results than the simplified analytical solutions.

The model (18) can be solved either in Eulerian form with the left hand side written as $\partial u / \partial t + u \partial u / \partial x$, or in the Lagrangian form written as $du/dt$. Since here we aim to explicitly quantify the effect of erosion in landslide velocity, for simplicity, we consider (18) in Lagrangian form and obtain the exact landslide velocity. However, we mention, that the exact solution of (18) can also be obtained in Eulerian form, but is very demanding mathematically. Pudasaini and Krautblatter[67] have presented various exact analytical solutions for landslide velocity, however, without considering the erosion effects. Here, we focus on the velocity of an erosive landslide considering the case where there is an increase in the landslide velocity due to the erosion-induced entrainment, i.e., for $\mathcal{P}_M > 0$ corresponding to $\lambda^b > 1/2$. However, analytical results for $\mathcal{P}_M < 0$ corresponding to $\lambda^b < 1/2$ resulting in reduced mobility even for an erosive landslide can be obtained and discussed similarly.

Now, we present the dynamics of the landslide mobility model. The model (18) is a first-order non-linear ordinary differential equation that possesses an exact analytical solution for the time evolution of the landslide velocity in the form of the tangent hyperbolic function:

$$u(t) = \sqrt{\frac{\mathcal{A} + \mathcal{P}_M}{\mathcal{C}}} \tanh\left[\sqrt{(\mathcal{A} + \mathcal{P}_M)\mathcal{C}}\ (t - t_i) + \tanh^{-1}\left(\sqrt{\frac{\mathcal{C}}{\mathcal{A} + \mathcal{P}_M}}\ u_i\right)\right],$$
(20)

where, $u_i$ is the initial (or, boundary) condition at a given time $t = t_i$.

Steady-state velocity plays an important role in practical application. For sufficiently large time (equivalently, sufficiently long distance), (20) can be represented as

$$\lim_{t \to \infty} u(t) = \sqrt{\frac{\mathcal{A} + \mathcal{P}_M}{\mathcal{C}}},$$
(21)

which is the steady-state velocity of the landslide, that is determined by the applied forces $\mathcal{A}$ and $\mathcal{C}$, and the erosion-induced mobility parameter $\mathcal{P}_M$, also a force. This particular solution could already be obtained from (18) by assuming the

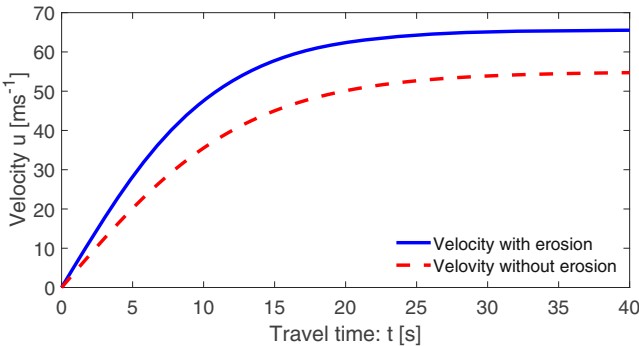

**Fig. 1 Time evolution of the landslide velocity with and without erosion given by (20).** Erosion enhances the landslide velocity and thus its mobility. With erosion, the steady-state velocity is higher and is reached earlier than the same without the erosion. This is due to the erosion-induced gain in momentum that increases the instantaneous velocity for which the drag takes shorter time to bring the motion to the steady-state, but with higher value.

steady-state condition, $0 = \mathcal{A} + \mathcal{P}_M - \mathcal{C}u^2$. The explicit time-independent form of the velocity in (21) is important in quickly solving technical and engineering problems. We call it the representative (steady-state) landslide mobility velocity, $u_s^{lm}$, and write

$$u_s^{lm} = \sqrt{\frac{\mathcal{A} + \mathcal{P}_M}{\mathcal{C}}}. \qquad (22)$$

Although it is simple as it appears, (22) includes many of the dominant and essential physical aspects of the material and the flow as carried by the definitions of $\mathcal{A}, \mathcal{P}_M$ and $\mathcal{C}$. The involved parameters can be estimated from their definitions, and depending on the situation, can have wide range of values.

To quantify $u$ in (20) and (22), we consider the often used and physically plausible parameter values with appropriate dimensions[6,27,48] as follows: for sliding mass: $\delta^m = 40°, \gamma^m = \rho_f^m/\rho_s^m = 1100/2900, \alpha^m = 0.75$; for the erodible basal substrate: $\delta^b = 10°, \gamma^b = \rho_f^b/\rho_s^b = 1000/2000, \alpha^b = 0.5, \lambda^b = 0.69$; where $\lambda^b$ is computed from these parameters. Furthermore, we consider a slope inclined at an angle $\zeta = 45°$. With these, we obtain the typical values of the model parameters as: $\mathcal{A} = 4.2271$, $\mathcal{P}_M = 1.7988$; and utilize $\mathcal{C} = 0.0014$. This results in some representative velocities of fast moving landslides: $u_s^{lm} = 55$ ms$^{-1}$ without erosion, and $u_s^{lm} = 65.6$ ms$^{-1}$ with erosion, which already shows significant difference between these velocities. However, based on the parameter values, the relative difference in velocities, with and without erosion, can be even higher as $\mathcal{P}_M$ might possibly be higher than $\mathcal{A}$. Here, we have just presented a possible scenario. These velocities are quite reasonable for fast to rapid landslides and debris avalanches and correspond to several natural events[68]. Simulation results show that the front of the 2017 Piz-Chengalo Bondo landslide (Switzerland) moved with more than 25 ms$^{-1}$ already after 20 s of the rock avalanche release[6], and later it moved at about 50 ms$^{-1}$, as mentioned in ref. [69]. The 1970 rock-ice avalanche event in Nevado Huascaran (Peru) reached a mean velocity of 50–85 ms$^{-1}$ at about 20 s, but the maximum velocity in the initial stage of the movement reached as high as 125 ms$^{-1}$, see[5,7,70]. The 2002 Kolka glacier rock-ice avalanche in the Russian Kaucasus accelerated with the velocity of about 60–80 ms$^{-1}$, but also attained the velocity as high as 100 ms$^{-1}$, mainly after the incipient motion[1,7]. All these events were substantially to highly erosive. By properly

selecting the model parameters, such exceptionally high velocities as inferred from the field (as mentioned above) can be obtained from the new model. Yet, the model must further be scrutinized with carefully calibrated parameters by reproducing laboratory experiments and back analyzing the natural events, which however, is not within the scope here.

It is crucial to acquire an in-depth picture of the time evolution of landslide velocity with erosion. The full time evolution of the landslide velocity with erosion given by (20) has been shown in Fig. 1 with $u_i = 0$ at $t_i = 0$. The flow dynamics is controlled by the competition (interaction) between the overall (net) driving and the resisting forces, $\mathcal{A} + \mathcal{P}_M$ and $\mathcal{C}u^2$, respectively. Importantly, if the initial velocity is less than the steady-state velocity, i.e., $u_i < u_s^{lm}$, then after its inception, the landslide accelerates (rapidly or slowly, depends on the magnitude of $u_s^{lm} - u_i$) because $\mathcal{A} + \mathcal{P}_M$ dominates $\mathcal{C}u^2$. Example includes the situation when the landslide is initially triggered with zero velocity, e.g., due to the slope failure from its static condition. However, in long time, as $\mathcal{C}u^2$ balances $\mathcal{A} + \mathcal{P}_M$, $u(t)$ asymptotically approaches, from below, the steady-state velocity, $u_s^{lm}$. This is the situation presented in Fig. 1. On the other hand, if the initial velocity is higher than the steady-state velocity, i.e., $u_i > u_s^{lm}$, then, after its triggering, the landslide decelerates (rapidly or slowly, depends on the magnitude of $u_i - u_s^{lm}$) because $\mathcal{C}u^2$ dominates $\mathcal{A} + \mathcal{P}_M$. The landslide triggered by strong seismic shacking is an example for this. Nevertheless, in long time, as $\mathcal{A} + \mathcal{P}_M$ tends to neutralize $\mathcal{C}u^2$, $u(t)$ asymptotically approaches, from above, the steady-state velocity, $u_s^{lm}$. Technically, $u_s^{lm}$ provides an important information of landslide velocity with erosion for landslide engineers and practitioners. Equations (20) and (22) clearly indicate that the higher the value of the mobility parameter $\mathcal{P}_M$ the earlier the landslide reaches its steady-state with substantially higher velocity. This is quite natural, because as erosion enhances the velocity, it takes relatively shorter time for the drag to control the acceleration of the landslide. In other words, this also proves that erosion enhances mobility for the positive values of the mobility parameter $\mathcal{P}_M$.

We can now quantify the importance of erosion. Figure 1 shows that, around $t = 15$ s, the velocities with and without erosion take values of about 57 and 44 ms$^{-1}$, respectively, with the maximum difference of 13 ms$^{-1}$. And, in long time, the corresponding steady-state velocities are 65.6 and 55 ms$^{-1}$. As the dynamic pressure is proportional to the square of the velocity, with respect to the steady-state velocities, the dynamic pressure with erosion is about 42% higher than the same without erosion. However, with respect to the maximum difference in the velocities at $t = 15$ s, the dynamic pressure with erosion is even 68% higher than the same without erosion. Crucially, these contrasts in velocities result in completely different run-out and deposition scenarios. This clearly manifests the importance of the correct inclusion of the erosion in modeling the landslide dynamics and run-out.

If we consider both the landslide and the basal substrate consisting of only solid particles and neglect all the fluid-related parameters (forces), we need to set $\alpha^m = 1, \alpha^b = 1, \gamma^m = 0, \gamma^b = 0$. Then, the velocities with and without erosion would be much smaller, and attain the steady-state values of 43.56 and 28.23 ms$^{-1}$, respectively. So, the steady-state is reached much later in time. However, the relative difference is 15.33 ms$^{-1}$, which is higher than before. This is because of the strongly reduced value of $\mathcal{A}$, but $\mathcal{P}_M$ decreases only slightly (to 1.1 and 1.5, respectively). The results are presented in Fig. 2. Yet, the maximum difference in velocities with and without erosion is about 18.30 ms$^{-1}$ (=39.4–21.1 ms$^{-1}$) at around $t = 25$ s. So, at this point, the dynamic pressure with erosion

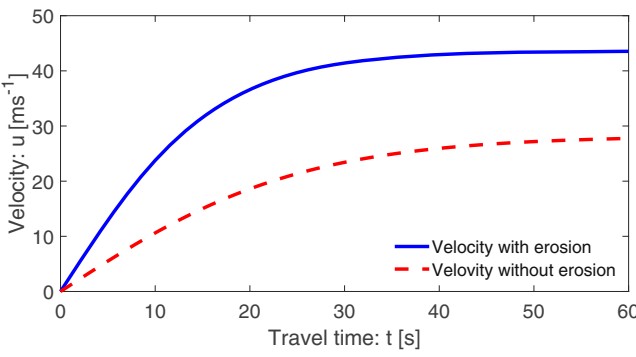

**Fig. 2 Time evolution of velocity with and without erosion for dry landslide and erodible bed substrate given by (20).** The landslide velocity, and thus its mobility, is largely enhanced by erosion.

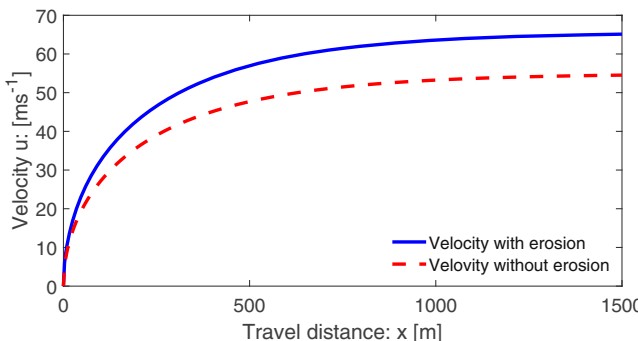

**Fig. 3 Evolution of the landslide velocity as a function of travel distance with and without erosion given by (25).** Erosion enhances the landslide velocity and thus its mobility.

is about 2.5 times higher than the same without erosion, which is a huge contrast.

Alternatively, the velocity can be expressed as a function of travel distance. For a mass point motion, we may write:

$$\frac{du}{dt} = \frac{du}{dx}\frac{dx}{dt} = u\frac{du}{dx}. \tag{23}$$

Then, (18) takes the form

$$u\frac{du}{dx} = (\mathcal{A} + \mathcal{P}_M) - \mathcal{C}u^2, \tag{24}$$

which can be solved analytically to obtain exact solution for the landslide velocity as a function of travel distance:

$$u(x) = \sqrt{\frac{\mathcal{A} + \mathcal{P}_M}{\mathcal{C}}\left[1 - \left(1 - \frac{\mathcal{C}}{\mathcal{A} + \mathcal{P}_M}u_i^2\right)\frac{1}{\exp(2\mathcal{C}(x - x_i))}\right]}, \tag{25}$$

where, $u_i$ is the initial velocity at $x_i$. The results have been presented in Fig. 3, where, both velocities have the same limiting values as in Fig. 1, otherwise their behaviors are quite different. In space, the velocity shows a hyper increase after the incipient motion. However, the time evolution of velocity is first slow (almost linear) then fast, and finally attains the steady-state, the common limiting value for both the solutions (20) and (25). These results indicate that, in any situations (Figs. 1–3), the differences in the landslide velocities with and without erosion are huge. This demonstrates the control of erosion over the landslide mobility.

We introduce the mobility scaling and erosion number. By considering the simple initial condition $u_i = 0$ at $x_i = 0$, the

structure of solution (25) clearly indicates that, there exists a unique number $\mathcal{S}_M$:

$$\mathcal{S}_M = \sqrt{1 + \frac{\mathcal{P}_M}{\mathcal{A}}}, \tag{26}$$

such that

$$u(x) = \mathcal{S}_M\, u_{n_{er}}(x), \quad u_{n_{er}}(x) = \sqrt{\frac{\mathcal{A}}{\mathcal{C}}\left[1 - \frac{1}{\exp(2\mathcal{C}x)}\right]}, \tag{27}$$

where, $u_{n_{er}}$ is the landslide velocity without erosion. We call $\mathcal{S}_M$ the mobility scaling. Both mechanically and technically, $\mathcal{S}_M$ has a great significance. First, it is simple, and exclusively depends on all the measurable physical and mechanical parameters of the landslide, the net driving force $\mathcal{A}$ and the mobility parameter $\mathcal{P}_M$. Second, it is a novel dimensionless number that scales the landslide mobility through velocity. Third, with the knowledge of the mobility parameter $\mathcal{P}_M$, the practitioners can recover the velocity of an erosive landslide from (27), even previously not knowing the velocity with erosion. This is a special property of the solution (25). This idea can equally be applied for general simulation results. Fourth, $\mathcal{S}_M$ depends non-linearly on $\mathcal{P}_M$. As discussed previously, $\mathcal{P}_M > 0$, $\mathcal{P}_M = 0$, or $\mathcal{P}_M < 0$ delineate the enhanced, neutralized, or reduced mobility regimes, so the range of $\mathcal{S}_M$ should be understood accordingly. Hence, for $\mathcal{P}_M = 0$, $\mathcal{S}_M = 1$ degenerates to the landslide without erosion, while $\mathcal{S}_M > 1$ for positive value of $\mathcal{P}_M$ corresponds to the erosion-enhanced mobility. However, $\mathcal{S}_M < 1$ in the negative $\mathcal{P}_M$ domain is that for reduced mobility. While $\mathcal{P}_M$ delivers the overall mobility as the additional force induced by erosion in the dynamical system (25), the mobility scaling $\mathcal{S}_M$ provides us with the direct and explicit measure of mobility by contrasting the landslide dynamics without erosion from that with erosion. As $\mathcal{S}_M$ exactly quantifies the contribution of erosion in landslide mobility, technically, this is the most attractive and pleasant feature of the mobility scaling.

In the definition of $\mathcal{S}_M$ in (26), the ratio

$$\mathcal{E}_N = \frac{\mathcal{P}_M}{\mathcal{A}}, \tag{28}$$

plays a central role. We call $\mathcal{E}_N$ the erosion number. The erosion number $\mathcal{E}_N$ is the second novel dimensionless number presented here as a ratio between the erosion-induced force $\mathcal{P}_M$ (also called the mobility parameter) and the net driving force $\mathcal{A}$ (per unit mass). Depending on whether $\mathcal{P}_M$ is positive, zero or, negative, $\mathcal{E}_N$ can be positive, zero or, negative implying the enhanced, unaltered or, reduced mobility. In connection to the definition of the mobility scaling $\mathcal{S}_M$, the possible negative value of $\mathcal{E}_N$ is the structural requirement, however, is not an odd. We may also call $\mathcal{E}_N$ the erosion-mobility number. The dependency of the mobility scaling as a function of the erosion number has been shown in Fig. 4. Typical parameter values for the results presented in Fig. 4 are: $\mathcal{A} = 4.2271$, $\mathcal{P}_M = 1.7988$, $\mathcal{E}_N = 0.4258$, $\mathcal{S}_M = 1.1941$, respectively.

The overall net driving force can be negative. We note that in situations when $\frac{\mathcal{P}_M}{\mathcal{A}} < -1$ or $\mathcal{A} + \mathcal{P}_M < 0$, we must re-derive the solutions to replace (20) and (25), as the new solutions would be structurally and dynamically different because of the changed interactions between the associated system forces. However, we do not elaborate in this aspect here.

The new model can be reduced to the classical Voellmy mass point model. In the structure, the model (18) or, (24), and its solution (20) or, (25) exist in literature[62] and is classically called Voellmy's mass point model[71,72] or Voellmy–Salm model[73,74]. Perla et al.[63] also called (18) the governing equation for the center of mass velocity. However, $(1 - \gamma^m)$, $\alpha^m$, and the term associated

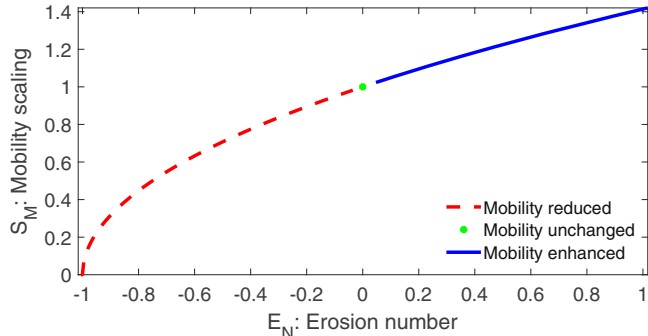

**Fig. 4 Non-linear dependency of the mobility scaling $\mathcal{S}_M$ on the erosion number $\mathcal{E}_N$ given by (26) and (28).** Three distinct mobility regimes are indicated.

with $\partial h/\partial x$, and erosion are the new contributions and were not included in the Voellmy model, and $K = 1$ therein, while in our consideration of $\mathcal{A}$, $K$ can be chosen appropriately. Thus, the Voellmy model corresponds to the substantially reduced form of $\mathcal{A}$, with $\mathcal{A} = g^x - g^z\mu^m$, and $\mathcal{P}_M = 0$.

**Complete set of dynamical landslide equations with erosion.** We have now the proper understanding of the structure of the inertial part of the momentum equation for landslide with erosion. For the full and better simulation of the erosive landslide problem, we must numerically integrate the mass and momentum balance equations that include the evolution of both the state variables, the flow velocity, and the flow depth. We can now formulate the full set of dynamical equations with erosion either in the non-conservative from, or in conservative form. However, in general, it is better to use the non-singular mechanical erosion rate as derived in Pudasaini and Fischer[27]:

$$E = \frac{\sqrt{g\cos\zeta h\left[(1-\gamma^m)\rho^m\mu^m\alpha^m - (1-\gamma^b)\rho^b\mu^b\alpha^b\right]}}{\sqrt{\nu(\rho^m\lambda^m\alpha^m - \rho^b\lambda^b\alpha^b)}}, \quad (29)$$

where, $\nu$ connects the shear velocity of the system with the flow velocity. Usually, the shear velocity is about 5–10% of the mean flow velocity. So, for simplicity, we can take a suitable value of $\nu$ in the domain (100, 400). Otherwise, we follow Pudasaini and Fischer[27] for an analytical closure relation for $\nu$.

First, we present the non-conservative formulation. The momentum balance equation (12), together with the mass balance (1) and the bed evolution (3), constitutes the set of full dynamical model equations for erosive landslide in non-conservative form

$$\frac{\partial h}{\partial t} + \frac{\partial}{\partial x}(hu) = E, \quad (30)$$

$$\frac{du}{dt} = g^x - \frac{u}{|u|}(1-\gamma^m)\alpha^m g^z\mu^m - g^z\left[((1-\gamma^m)K+\gamma^m)\alpha^m + (1-\alpha^m)\right]\frac{\partial h}{\partial x}$$
$$- g^z\frac{\partial b}{\partial x} - C_{DV}u^2 + (2\lambda^b - 1)E^P, \quad (31)$$

$$\frac{\partial b}{\partial t} = -E, \quad (32)$$

where, $E^P = Eu/h$, and for completeness, the factor $\frac{u}{|u|}$ in the Coulomb friction term and the force induced by the detailed topographic effect $-g^z\frac{\partial b}{\partial x}$ have been restored. These equations include the evolution of both the flow velocity and the flow depth for the erosional landslide.

Second, we present the conservative formulation. Due to the possible rapid spatial and temporal changes of the flow variables, in practice, the mass flow model equations are often exclusively solved in conservative form[6,48,61,62,75]. So, we need to formulate the balance equations in conservative form. First, we note that the third term on the left hand side of (2), i.e., $\frac{\partial}{\partial x}\left[(1-\gamma^m)\alpha^m g^z Kh^2/2\right]$ emerges from the stress (particularly known as the hydraulic pressure gradient), and thus not a part of the inertia of the system. Classically, this term is organized that way only for the convenience and for analytical or numerical reasons[76–78]. We have seen that, for the erosive landslide, the main essence lies in the use of the correct velocity (the entrainment velocity) in transporting the newly entrained mass (the rate of change of landslide mass) that appears in the inertial part of the momentum balance equation. With this, the non-conservative momentum equation (4) can be structurally brought back to the conservative form (2). This is achieved by legitimately and consistently re-writing (4) as:

$$h\left[\frac{\partial u}{\partial t} + u\frac{\partial u}{\partial x}\right] + (u-u^b)\left[\frac{\partial h}{\partial t} + \frac{\partial}{\partial x}(hu)\right]$$
$$= h\left[g^x - \frac{u}{|u|}(1-\gamma^m)\alpha^m g^z\mu^m - g^z\left\{((1-\gamma^m)K+\gamma^m)\alpha^m + (1-\alpha^m)\right\}\frac{\partial h}{\partial x} - g^z\frac{\partial b}{\partial x} - C_{DV}u^2\right]$$
$$+ u^b E. \quad (33)$$

The physical reason for the appearance of $(u - u^b)$, on the left hand side, for the erosional landslide has been explained previously. With the help of (30), this can be written as

$$\frac{\partial}{\partial t}(hu) + \frac{\partial}{\partial x}\left[hu^2 + (1-\gamma^m)\alpha^m g^z K\frac{h^2}{2}\right]$$
$$= h\left[g^x - \frac{u}{|u|}(1-\gamma^m)\alpha^m g^z\mu^m - \{1-(1-\gamma^m)\alpha^m\}g^z\frac{\partial h}{\partial x} - g^z\frac{\partial b}{\partial x} - C_{DV}u^2\right] + 2u^b E. \quad (34)$$

This completes the process of deriving the full set of dynamical equations (mass and momentum balances) for erosional landslide in conservative form.

As it is clear from the derivation, the momentum balance equation (34) correctly includes the erosion-induced change in inertia and the momentum production of the system. This equation is the same as that in Pudasaini and Fischer[27] except that the last term on the right hand side is now $2u^b E$ instead of $u^b E$. In (34) one $u^b E$ emerges from the momentum production derived from the effectively reduced friction (as in ref. [27]), while the other $u^b E$ originates from the correct understanding of the inertia of the entrained mass that has not yet been considered in any existing models including the one by Pudasaini and Fischer[27]. However, mechanically and dynamically, this makes a huge difference, and thus, is a great advancement in simulating landslide with erosion. Importantly, our present analysis makes a complete description of the full dynamical model equations for erosive landslide in conservative form by considering all the aspects associated with the erosion-induced reduced friction (the momentum production) and the correct handling of the inertia of the system, which now reads:

$$\frac{\partial h}{\partial t} + \frac{\partial}{\partial x}(hu) = E, \quad (35)$$

$$\frac{\partial}{\partial t}(hu) + \frac{\partial}{\partial x}\left[hu^2 + (1-\gamma^m)\alpha^m g^z K\frac{h^2}{2}\right]$$
$$= h\left[g^x - \frac{u}{|u|}(1-\gamma^m)\alpha^m g^z\mu^m - \{1-(1-\gamma^m)\alpha^m\}g^z\frac{\partial h}{\partial x} - g^z\frac{\partial b}{\partial x} - C_{DV}u^2\right]$$
$$+ 2\lambda^b Eu, \quad (36)$$

$$\frac{\partial b}{\partial t} = -E, \quad (37)$$

where, as before, the substitution $u^b = \lambda^b u$ has been made, and the dynamics of $\lambda^b \in (0,1)$ has been described in the previous sections. All the analyses (with respect to $u^b$, or $\lambda^b$) presented above about the gained or lost momentum (energy), and the enhanced or reduced mobility of the erosive landslide are also valid for the model equations (35)–(37) written in conservative form and describe the evolution of the flow depth and the flow velocity. We note that, in general, the complete and the continuum description of the dynamical landslide equations with erosion (35)–(37) are preferable over their discrete counter part (9) or (18).

The new model can be extended to multi-phase mass flow simulation. It is now so convenient that the models (35)–(37) can be directly applied to the mechanical erosion model for two-phase mass flows developed by Pudasaini and Fischer[27] just by replacing $\lambda^b$ by $2\lambda^b$ in the momentum balance equations in their model. Importantly, this also lays a foundation to further apply these methods to multi-phase mass flow model[61] to include erosion-induced mobility.

## Discussion

Here, we present discussions on the novelty, essence and implications of the new model. Erosion-induced increased or decreased mobility has been reported with the relevant data in the laboratory and/or from the field events[27–31]. However, no clear-cut mechanical derivation and explanations have been presented in the literature so far for when and how the erosion related mass flow gains energy leading to enhanced-mobility, or loses energy that consequently forces the landslide to reduce its velocity and thus its run-out distance, resulting in reduced mobility. We have presented the first-ever, analytically constructed simple and clear mechanical condition for erosion-induced energy budget that delineates energy gain or loss, and the associated enhanced or reduced mass flow mobility. We analytically derived two important dimensionless numbers, the mobility scaling, and the erosion number, as a function of the ratio between the erosion-induced force and the net driving force. These numbers provide the practitioners with the simple and direct measure of mobility by comparing the landslide dynamics without erosion from that with erosion. Furthermore, the mobility scaling offers an exact quantification of erosion in mobility.

By rigorous derivation, Pudasaini and Fischer[27] showed that appropriate incorporation of the mass and momentum productions or losses in conservative model formulation is essential for the physically correct and mathematically consistent description of erosion-entrainment-deposition processes. They proved that effectively reduced friction in erosion is equivalent to the momentum production. With this, Pudasaini and Fischer[27] demonstrated that erosion can induce a higher mass flow mobility. Although they laid the foundation for the mechanical erosion-rate model, their model is incomplete as they did not deal with the erosion-induced inertia. In the conservative formulation, the change in inertia is implicit in the inertial part of the momentum balance equation. However, the explicit and full quantification of the available energy (or, momentum), and thus the state of mobility of the erosive landslide, requires the combined analysis of the erosion-induced changed inertia of the system and the produced momentum. This is vital for applications in real-world problems as it enables us to determine the actual change in the momentum and consequently the energy, and the mobility of the landslide with erosion. Here, we have achieved this by utilizing the non-conservative formulation which made it possible to explicitly express the changed inertia of the system that combines the erosion-related rate of change of mass (resulting in the mass production), $E$, and the relative velocity of

the eroded particles with respect to the landslide velocity, namely the entrainment velocity, $u^{ev} = u - u^b$. This combination uniquely yielded the actual change in the inertia of the system, $(u - u^b)E$. As shown above, this, together with the produced momentum, $u^b E$, provides the first-ever explicit and complete mechanical quantification of the (overall) state of induced momentum or energy, $(2\lambda^b - 1)E^P$, associated with the erosive landslide, and hence, the precise description of its mobility. This, in fact, fully addresses the long-standing scientific and engineering question of why and when some erosive landslides can have higher mobility, while others have their mobility reduced even in the erosive situation.

Most of the existing and influential erosion models[25,35,37,46,47,63] do not include the momentum production in the momentum balance equation. These aspects have also been partially discussed with a mechanical erosion model for two-phase mass flow by Pudasaini and Fischer[27]. Moreover, none of the existing models deal with erosion-induced changed inertia of landslide. Essentially, Perla et al.[63], McDougall and Hungr[35], and Iverson[37] directly replaced $\frac{d}{dt}(mu)$ with $m\frac{du}{dt} + u\frac{dm}{dt}$ in the inertial part of the momentum equation, which is mechanically invalid for the erosive landslide. Therefore, these types of models result in the unphysical loss of energy associated with the erosive landslide, and hence, cannot properly explain the state of energy and mobility.

So, from the physical point of view, most of the existing and dominant erosion models for mass movements are erroneous because of two reasons: first, incorrect formulation of the inertial part of the momentum equation. And, second, the neglection of the momentum production, associated with the erosion velocity, in the momentum balance equation. Together with the Pudasaini and Fischer[27] model, we solved this fundamental problem in landslide motion with erosion paving now the way for the correct modeling and prediction of landslide dynamics, that erosion essentially changes the state of energy and mobility, and thus the run-out, dynamic impact and inundation. So, the present contribution fundamentally enhances our understanding of mobility of erosive mass movements.

We have also derived a full set of dynamical equations in conservative form in which the momentum balance correctly includes the erosion-induced change in inertia and the net energy gain or the net momentum production, $2\lambda^b Eu$. This offers a legitimate simulation of landslide motion with erosion. The new effectively single-phase mass flow model with erosion (35)–(37) can be directly extended to crucially enhance the existing two-phase erosion model[27], or to the multi-phase mass flow model[61]. This allows us to simulate much more realistic erosion related mass flow mobility in real events. The application of the novel model to experimental and complex natural events of landslides, debris and avalanche motions would require substantial additional work, and corresponding parameter estimates, either derived from field measurements or back calculations, involving observation data, which, therefore, has to be deferred to some future contributions.

Finally, we summarize our findings. Mobility of an erosive landslide can be attributed to its excessive volume and material properties, and is marked by rapid motion, exceptional travel distance, and the inundation area. Proper knowledge of mobility is required in accurately determining the dynamics and enormous impact force. However, most of the existing influential erosive landslide models do not include momentum production and none of them deal correctly with erosion-induced changed inertia. The correct consideration of inertia is crucial for the precise derivation of the dynamical landslide model with erosion. A novel understanding is that the increased (changed) inertia is not related to the landslide velocity, but it is associated with the

distinctly different entrainment velocity emerging from the inertial frame of reference of the landslide. The classical representation of inertia appeared to be wrong for erosional situations. We eliminated the existing erroneous perception on erosive landslides and correctly determine the energy and thus the mobility of the erosive landslide, that erosion fundamentally changes the state of energy and mobility, and consequently the dynamic impact and inundation. The actual change in inertia together with the produced momentum provides the first-ever explicit and full mechanical quantification of the state of energy, and thus, the precise description of mobility of the erosive landslide. This addresses the long-standing scientific question of why and when some erosive landslides can have higher mobility, while the others have reduced mobility.

We revealed, that the erosion velocity plays an outstanding role in appropriately determining the energy budget and mobility of an erosive landslide. The mobility is clearly defined and fully controlled by the erosion velocity. If the erosion velocity is greater than one half of the flow velocity, the mobility is enhanced. Erosion velocity closer to the flow velocity results in the most mobile flow. If the erosion velocity is less than one half of the flow velocity then, the mobility is reduced. The landslide consumes the highest amount of energy if the erosion velocity is much smaller than the flow velocity. The erosion will not change the energy status, and hence the mobility of the landslide, if the erosion velocity is one-half of the flow velocity. Based on the newly constructed energy generator, we extracted an important conclusion: whether the erosion-related mass flow mobility will be enhanced, reduced or remains unaltered depends exclusively on whether the energy generator is positive, negative or zero. This becomes the game-changer and explicitly tells us the state of mobility, and ultimately regulates the destructive power of the landslide. With the knowledge of the energy generator, the involved energy in landslide erosion can now be quantified from the overall mobility parameter. Such an explicit and fully mechanical description of the state of mobility of an erosive landslide is seminal. We introduced three principally novel mechanical concepts: the erosion-velocity, entrainment-velocity, and the energy-velocity, and demonstrated that the erosion and entrainment are essentially different processes. With this, we drew a central inference: the landslide gains energy, and thus enhances its mobility, if the erosion velocity is greater than the entrainment velocity. We call this phenomenon the landslide-propulsion, emerging from the net momentum production, that provides the erosion-thrust to the landslide. The energy velocity associated with the net momentum production clearly delineates the three excess energy regimes: positive, negative, or zero, resulting in the corresponding enhanced, reduced or unaltered mobility of the erosional landslide.

Based on the newly derived basic erosional landslide equation, we constructed a simple landslide mobility equation. The great advantage of the new mobility equation is that the erosion enhanced mobility can now be directly quantified. This is the first-ever physics-based model to do so. We explicitly obtained the landslide velocity by analytically solving the mobility equation for the erosion-induced increased landslide velocity. This form of the velocity is very useful in quickly solving relevant engineering and applied problems and has enormous application potential. Analytically obtained velocities demonstrate that erosion can have the major control on the landslide dynamics. As the dynamic pressure is proportional to the square of the velocity, the dynamic pressure with erosion can be much higher than the same without erosion. Similarly, the large contrast in velocity with and without erosion can result in completely different run-out scenarios, much more extensive for erosive landslides. Technically, this provides very important information for landslide practitioners in accurately determining the landslide velocity with erosion. We constructed two innovative and useful dimensionless numbers, the mobility scaling and erosion number, providing a direct measure of landslide mobility with erosion. This offers the unique possibility to precisely quantify the significance of erosion in mobility. Importantly, we have also derived the full set of dynamical equations for landslide in conservative form in which the momentum balance correctly includes the erosion-induced change in inertia and the momentum production. This is a great advancement in fully simulating landslide with erosion. This clearly suggests the importance of the correct inclusion of erosion in modeling the landslide dynamics and run-out.

## Data availability

The relevant data supporting the findings of this study are available within the paper.

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

## Acknowledgements

Shiva P. Pudasaini acknowledges the financial support provided by the Technical University of Munich with the Visiting Professorship Program, and the international research project: AlpSenseRely—Alpine remote sensing of climate-induced natural hazards—from the Bayerisches Staatsministerium für Umwelt und Verbraucherschutz, Munich, Bayern. This paper is based on: arXiv:2103.14842v1, https://arxiv.org/pdf/2103.14842.pdf.

## Author contributions

The physical-mathematical models were developed by S.P.P. who also designed and wrote the paper. M.K. further improved the paper, contributed to the analysis, and discussions of the results with enhanced descriptions to better fit the broader geosciences audiences. Both S.P.P. and M.K. interpreted the results and additionally edited the paper through reviews.

## Funding

## Competing interests

The authors declare no competing interests.
