## [Peer Review File · Nature Communications]

Reviewers' Comments:

Reviewer #1:

Remarks to the Author:

Excellent paper. Congratulations.

The framework proposed combines in a unique model the previous numerous attempts to understand and model erosion, entrainment, energy consumption for long-runout landslides.

I am just recommending a text definition or a conceptual description of the newly introduced velocities: Erosion, Entrainment, Energy...

The equations are clear but a definition could help people in transferring these new concepts to other scholars, students, or professionals.

Reviewer #2:

Remarks to the Author:

The erosion of basal material play a important role in landslides, debris flow and floods. In the past decades, lots of researchers obtained amount of achievements in this domain. In my opinion, two different ways of attempts are approaching and gain broad support. One is to build an empirical and operable erosion rate computational equation to taking the erosion into account. The other is to build mechanical-based erosion rate from strict physical derivation. In this manuscript, the authors tried to re-analyse their derived equations to explain the mobility mechanism of landslide when considering the erosion. Generally, considering the high-quality requirement of NC, the manuscript can not be accepted as its novelty is not sound.

First of all, the most idea or concepts such as the difference of erosion and entrainment, erosion-velocity, entrainment-velocity, the positive or negative effect from the erosion, were provided or described in their own or previous published documents. In other aspects, landslides is a specific mass movement of two-phase flow. For the same reason, the mass and momentum equations should be a simplified equations which could be derived from their two-phase equations by neglecting some terms or some discrepancies.

To consider the interaction between the flow mass and the basal material, it is essential to clearly define the velocity in the erosion boundary, for instance, the velocity of landslide in the boundary and the velocity of erodible material. In this manuscript, the u which is averaged velocity of landslide, the u_b represents the velocity of erodible material. The u can not represent the velocity in the boundary. The dealing of this kind without a definite statement in the boundary is not clear for readers.

The authors stated that the expression $F=d(\mu)/dt=m*du/dt+u*dm/dt$ is wrong. It could easily mis-guide readers. Undoubtedly, the expression is correct as it is just the derivative operator. The way it could be wrong only people improperly use it. Currently, the authors did not clearly explain it to readers.

Lastly, the writing style is not proper for NC as too many equations derivation in detail. It is more proper for a professional journal not a comprehensive journal.

Reviewer #3:

Remarks to the Author:

In the attached file, it is possible to find the review report.

Response to Reviewer # 1: MS #: NCOMMS-21-19648

Reviewer's comments and suggestions are in plane text indicated by C, our responses and enhancements are in blue text as indicated by R. Please see the improved manuscript text in accordance with the responses here in which the removed portions are in red, and edited and added texts are in blue colour. The manuscript is now in the format compatible with NC. The Abstract has been significantly condensed to approach the requirement of NC. P refers to the page number in the revised manuscript with track changes in colour.

C: Excellent paper. Congratulations.

R: We thank the reviewer very much for the appreciation on our new model and the new scientific aspects we brought in this manuscript. We duely admire your valuable time and interest on our work and suggestions that truly resulted in the substantially improved revised manuscript in which we appropriately addressed all the concerns, we believe turning it now in to a suitable paper to appear in Nature Communications soon.

C1. The framework proposed combines in a unique model the previous numerous attempts to understand and model erosion, entrainment, energy consumption for long-runout landslides.

R1: As recognized by the reviewer, this manuscript addresses the long-standing scientific question of why many erosive landslides generate higher mobility, while others reduce mobility. We have provided the first-ever explicit mechanical quantification of the state of erosional energy and a precise description of mobility. These are genuinely new and significant achievements associated with the erosional landslide.

C2. I am just recommending a text definition or a conceptual description of the newly introduced velocities: Erosion, Entrainment, Energy...

R2: Following the reviewer's important suggestion, in the revised manuscript, we have defined [P11]: "We call these, the-three- E mechanical-concepts."

The conceptual description of the newly introduced velocities have now been added as [P4]: "Next, we define some (mostly) fundamentally new mechanical concepts and terminologies as follows. This will help to comprehensively streamline the paper. Erosion: a mechanical process by which the bed material is mobilized by the flow. Entrainment: a mechanical process by which the eroded material is incorporated (entrained) and taken along with by the flow. Erosion-velocity: $u^e := u^b$, the velocity of the eroded material from the basal substrate. Entrainment-velocity: $u^{ev} := u - u^b$, the velocity of the entrained mass, or the velocity of the landslide minus the velocity of the eroded mass. Energy-velocity: $u^{ev} := u^b - (u - u^b) = u^e - u^{ev}$, the velocity associated with the net momentum production due to erosion that generates the excess (kinetic) energy, or the erosion velocity in excess to the entrainment velocity. Energy: associated with the erosion-induced net momentum production that results in the excess (kinetic) energy. Energy-generator (or, the mobility generator): the parameter that generates the excess energy (or the mass flow mobility) due to erosion. Energy-budget: the state of available excess energy generated by the net momentum production due to erosion. Erosion-rate: $E = -\partial b/\partial t$, the negative time rate of change of the basal topography, or the time rate of change of mass resulting in the mass production. The mechanical significance and the mathematical structures of these concepts and terminologies will be clearer in the due places when they appear in the text."

C3. The equations are clear but a definition could help people in transferring these new concepts to other scholars, students, or professionals.

R3: To make the text clearer and understandable to the broader scholars, students, or professionals, and the audiences of NCOMMS, we have now clearly defined the variables and parameters and the adopted processes as follows [P3-4]: "First we define variables and parameters. The landslide mass is denoted by m . The superscripts m and b represent the parameters or the quantities in the landslide (mixture), and the erodible basal substrate, respectively. For simplicity, we consider a geometrically two-dimensional flow. Let t be the time, (x, z) be the coordinates and (g^x, g^z) the gravity accelerations along and perpendicular to the slope, respectively. Let, h

and u be the flow depth and the mean flow velocity along the slope. Similarly, $\gamma^m, \alpha^m, \mu^m$ be the density ratio between the fluid and the particles ($\gamma^m = \rho_f^m / \rho_s^m$), volume fraction of the solid particles (coarse and fine solid particles is represented by a single solid-phase: $\alpha^m := \alpha_s + \alpha_{fs}$, and $\rho^m := \rho_s^m$ is the effective density of the solid in the mixture), and the basal friction coefficient ($\mu^m = \tan \delta^m$) related to the basal friction angle δ^m , in the mixture. Analogously, $\rho^b = \rho_s^b$ is the density of the solid, $\gamma^b = \rho_f^b / \rho_s^b$ is the density ratio between the fluid and the particles and α^b is the solid volume fraction in the bed material. $\mu^b = \tan \delta^b$ is the Coulomb friction parameter associated with the erodible bed with the friction angle δ^b . Note that differences in μ^m and μ^b can result due to the difference in the landslide material and the bottom material. In simple situation, μ^m and μ^b can have similar values²⁷. Furthermore, $b = b(x; t)$ is the erodible bed surface that evolves in space and time, E is the basal erosion rate, u^b is the erosion velocity, K is the earth pressure coefficient as a function of the internal (ϕ^m) and the basal (δ^m) friction angles, and C_{DV} is the viscous drag coefficient.” And [P4]:

“The Eulerian and Lagrangian descriptions are adopted here by purpose that will be clear in due places. First, we write the existing balance equations in standard and usual Eulerian form. Then, we utilize the most simple and convenient discrete procedure for the correct handling of the true inertia of the landslide with erosion. Classically, similar approaches have been used in deriving rocket equations. The Lagrangian descriptions are the most simple form that clearly demonstrate how erosion can result in substantially increased landslide velocity leading to enhanced mobility. Finally, the full set of erosional landslide model equations are written in the Eulerian description in the more sophisticated form that was largely facilitated by the Lagrangian descriptions.”

Response to Reviewer # 2: MS #: NCOMMS-21-19648

Reviewer's comments and suggestions are in plane text indicated by C, our responses and enhancements are in blue text as indicated by R. Please see the improved manuscript text in accordance with the responses here in which the removed portions are in red and edited and added texts are in blue colour. The manuscript is now in the format compatible with NC. The Abstract has been significantly condensed to approach the requirement of NC. P refers to the page number of the revised manuscript with track changes in colours.

C1: The erosion of basal material play a important role in landslides, debris flow and floods. In the past decades, lots of researchers obtained amount of achievements in this domain. In my opinion, two different ways of attempts are approaching and gain broad support. One is to build an empirical and operable erosion rate computational equation to taking the erosion into account. The other is to build mechanical-based erosion rate from strict physical derivation. In this manuscript, the authors tried to re-analyse their derived equations to explain the mobility mechanism of landslide when considering the erosion. Generally, considering the high-quality requirement of NC, the manuscript can not be accepted as its novelty is not sound.

R1: With due respect, we thank the reviewer very much for your time and interest on our manuscript. We have now revised the manuscript by improving it as much as applicable with respect to your comments. However, for simple reasons made clear below with respect to the initial submission and the excerpts therein, and also the revised manuscript, we cannot accept most of the comments and criticisms raised by the reviewer. There can be one or several reasons that the reviewer could not be positive on our manuscript. First, not careful and thorough reading of the manuscript and probably not properly understanding it. Second, seemingly inobjective assessment of the manuscript. Third, illegitimately neglecting several very important novel and seminal scientific findings we reported in this manuscript. So, we can not agree with this. Nevertheless, with the following clarifications, we hope that the revised manuscript is suitable to appear in Nature Communications soon.

On "In this manuscript, the authors tried to re-analyse their derived equations to explain the mobility mechanism of landslide when considering the erosion." This is incorrect. The main aspect of this manuscript is not to try to re-analyse the (previously) derived equations to explain the mobility mechanism, rather to fundamentally advance the science (mechanics) of mobility of erosional landslide. Please check the following, carefully written and proven, statements excerpted from the text that sufficiently dismiss the improper claim of the reviewer.

On "the high-quality requirement of NC", and "its novelty is not sound": We fully agree that NC publishes high-quality research results with novelty. That was exactly the reason we submitted our manuscript to NC. The following new scientific contents clearly manifest the high-quality, novel and very important results in our manuscript. This becomes crystal-clear by carefully and thoroughly reading the paper. So, the reviewer's claims are unfounded.

[P3]: "The Pudasaini and Fischer²⁷ model built a foundation by mechanically including the momentum production. Nevertheless, their model appears incomplete as they did not deal with the inertia of the entrained mass and could not present a clear mechanical condition for when and how the mobility of an erosive landslide will be enhanced or reduced and how to quantify it."

"Thus, whether erosion will result in the enhanced mass flow mobility and the quantification of its influence remains an unsolved problem. Extending the Pudasaini and Fischer²⁷ model, here, we have addressed this important issue by explicitly deriving mechanical conditions for the mobility of erosive landslides. By introducing a simple landslide mobility equation, we mechanically explain how and when erosive landslides enhance or reduce their mobility. This has been made possible by physically correctly considering the inertia of the erosive landslide. The model offers the first-ever opportunity to distinctly quantify the mobility of an erosive landslide."

"Importantly, a novel mechanism of landslide-propulsion has been identified that emerges from the net momentum production, providing the erosion-thrust to the landslide."

“We constructed two dimensionless numbers, the mobility scaling and the erosion number, the first of this kind in mass flow modelling, delivering the explicit measure of mobility.”

“This constitutes a foundation for legitimate and physically plausible simulation of landslide motion with erosion.”

[P9]: “The great advantage of (16) is that the erosion-enhanced flow mobility can now be explicitly evaluated in terms of velocity, as all the quantities (except u) on the right hand side of (16) are measurable, or given. This is the first-ever physics-based model to do so.”

“This is how the energy generator changes the game and fully controls the mobility of the erosive landslide.”

[P10]: “Such an explicit and fully mechanical description of the state of energy (or, momentum) and the associated mobility of an erosive landslide is seminal.”

[P11]: “So, the present paradigm further enhances the mechanical erosion model by Pudasaini and Fischer²⁷ and offers a complete and legitimate model for landslide erosion.”

“While the erosion velocity was first introduced by Pudasaini and Fischer²⁷, the entrainment velocity and the energy velocity are completely new concepts.”

“The existing literature could not distinguish between the erosion and entrainment as these terms are used interchangeably. However, here, we have made it very clear with the mechanical expressions, that the erosion and entrainment are essentially different phenomena. Erosion is a process by which the bed material is mobilized by the flow with the velocity $u^e = u^b$, while entrainment is intrinsically another process by which the eroded material is incorporated (entrained) and taken along with by the flow with the velocity $u^{ev} = u - u^b$. This fundamentally enhances our understanding of basic, but different processes in erosion related phenomena in landslide by clearly defining, and distinguishing the mechanisms of erosion and entrainment. These are important novel aspects.”

[P16]: “As \mathcal{S}_M exactly quantifies the contribution of erosion in landslide mobility, technically, this is the most attractive and pleasant feature of the mobility scaling.”

“The erosion number \mathcal{E}_N is the second novel dimensionless number presented here”

[P18]: “This equation is the same as that in Pudasaini and Fischer²⁷ except that the last term on the right hand side is now $2u^bE$ instead of u^bE . In (34) one u^bE emerges from the momentum production derived from the effectively reduced friction (as in²⁷), while the other u^bE originates from the correct understanding of the inertia of the entrained mass that has not yet been considered in any existing models including the one by Pudasaini and Fischer²⁷. However, mechanically and dynamically, this makes a huge difference, and thus, is a great advancement in simulating landslide with erosion. Importantly, our present analysis makes a complete description of the full dynamical model equations for erosive landslide in conservative form by considering all the aspects associated with the erosion-induced reduced friction (the momentum production) and the correct handling of the inertia of the system, ...”

[P20]: “The classical representation of inertia appeared to be wrong for erosional situations. We eliminated the existing erroneous perception on erosive landslides and make a breakthrough in correctly determining the energy and thus the mobility of the erosive landslide, that erosion fundamentally changes the state of energy and mobility, and consequently the dynamic impact and inundation.”

C2: First of all, the most idea or concepts such as the difference of erosion and entrainment, erosion-velocity, entrainment-velocity, the positive or negative effect from the erosion, were provided or described in their own or previous published documents. In other aspects, landslides is a specific mass movement of two-phase flow. For the same reason, the mass and momentum equations should be a simplified equations which could be derived from their two-phase equations by neglecting some terms or some discrepancies.

R2: On “the most idea or concepts such as the difference of erosion and entrainment, erosion-velocity,

entrainment-velocity, the positive or negative effect from the erosion, were provided or described in their own or previous published documents.”: This is not true. We have made it very clear in the manuscript with the following strong mechanical derivations and physically-based statements. The following are the genuinely first-ever novel aspects presented in this manuscript:

[P1]: “This depends on whether the newly developed energy generator is positive, negative or zero. The energy-velocity delineates the three excess-energy-regimes: positive, negative and zero.”

“This provides the first-ever explicit mechanical-quantification of the state of erosional-energy and a precise description of mobility. This addresses the long-standing scientific question of why many erosive-landslides generate higher mobility, while others reduce mobility.”

[P3]: “This has been made possible by physically correctly considering the inertia of the erosive landslide. The model offers the first-ever opportunity to distinctly quantify the mobility of an erosive landslide.”

“We constructed two dimensionless numbers, the mobility scaling and the erosion number, the first of this kind in mass flow modelling, delivering the explicit measure of mobility. The mobility scaling (precisely) quantifies the contribution of erosion in landslide mobility.”

[P9]: “The great advantage of (16) is that the erosion-enhanced flow mobility can now be explicitly evaluated in terms of velocity, as all the quantities (except u) on the right hand side of (16) are measurable, or given. This is the first-ever physics-based model to do so. Thus, it has enormous application potential.”

[P10]: “With the knowledge of the energy generator $\mathcal{P}_{Meg} = (2\lambda^b - 1)$, the involved net energy in landslide erosion can now be quantified from the mobility parameter \mathcal{P}_M . Such an explicit and fully mechanical description of the state of energy (or, momentum) and the associated mobility of an erosive landslide is seminal.”

[P11]: “Here, we formally introduce three important concepts with their underlying mechanics. These are: (i) the erosion-velocity, $u^e = u^b$, (ii) the entrainment-velocity, $u^{ev} = u - u^b$, and (iii) the energy-velocity, $u^{env} = u^b - (u - u^b)$. We call these, the-three- E mechanical-concepts.”

“Understanding the difference between erosion and entrainment is important. The existing literature could not distinguish between the erosion and entrainment as these terms are used interchangeably. However, here, we have made it very clear with the mechanical expressions, that the erosion and entrainment are essentially different phenomena. Erosion is a process by which the bed material is mobilized by the flow with the velocity $u^e = u^b$, while entrainment is intrinsically another process by which the eroded material is incorporated (entrained) and taken along with by the flow with the velocity $u^{ev} = u - u^b$. This fundamentally enhances our understanding of basic, but different processes in erosion related phenomena in landslide by clearly defining, and distinguishing the mechanisms of erosion and entrainment. These are important novel aspects.”

[P12]: “We can now explain the novel mechanism of landslide-propulsion and erosion-thrust. With the definitions, their mechanics and the discussions in the previous sections, we draw a central conclusion: the landslide gains energy, and thus enhances its mobility if the energy-velocity is positive, specifically, if the erosion velocity is greater than the entrainment velocity, i.e., $u^e > u^{ev}$. We call this phenomena the landslide-propulsion, emerging from the net momentum production, that provides the erosion-thrust to the landslide. This means, if the erosion velocity is greater than one half of the flow velocity, i.e., $u^b > u/2$, the mobility is enhanced. This is equivalent to the condition $\lambda^b > 1/2$. In other words, the landslide gains energy to enhance its mobility if the eroded material is easily entrainable with the velocity lower than the erosion velocity. These are quite natural phenomena, but revealed here for the first time.”

[P15-16]: “We call \mathcal{S}_M the mobility scaling. Both mechanically and technically, \mathcal{S}_M has a great significance. First, it is simple, and exclusively depends on all the measurable physical and mechanical parameters of the landslide, the net driving force \mathcal{A} and the mobility parameter \mathcal{P}_M . Second, it is a novel dimensionless number that scales the landslide mobility through velocity.”

[P16]: “We call \mathcal{E}_N the erosion number. The erosion number \mathcal{E}_N is the second novel dimensionless number

presented here as a ratio between the erosion-induced force \mathcal{P}_M (also called the mobility parameter) and the net driving force \mathcal{A} (per unit mass). Depending on whether \mathcal{P}_M is positive, zero or, negative, \mathcal{E}_N can be positive, zero or, negative implying the enhanced, unaltered or, reduced mobility.”

[P20-21]: “Based on the newly constructed energy generator, we extracted an important conclusion: whether the erosion related mass flow mobility will be enhanced, reduced or remains unaltered depends exclusively on whether the energy generator is positive, negative or zero. This becomes the game-changer and explicitly tells us the state of mobility, and ultimately regulates the destructive power of the landslide.”

[P21]: “The energy velocity associated with the net momentum production clearly delineates the three excess energy regimes: positive, negative, or zero, resulting in the corresponding enhanced, reduced or unaltered mobility of the erosional landslide.”

On “two-phase flow”: We appreciate for these comments. We know these aspects very well. Generally speaking, mass flows are multi-phase flows, as we pioneered the first-ever, multi-mechanical, multi-phase mass flow model in Pudasaini and Mergili (2019) by extending the leading general two-phase mass flow model by Pudasaini (2012). Moreover, we are the first to present the mechanical erosion model for two-phase mass flows (Pudasaini and Fischer, 2020), but without the possibility to quantify the state of mobility. Due to the complexity of the erosion associated mass flows, here we focus on the first-ever mechanical model to truly describe the mechanics of landslide mobility with erosion. Once we establish the foundation for a single-phase landslide, later we can further develop more complex erosion-induced mobility models for two-, or even multi-phase mass flows. These aspects have been clearly mentioned in the text as:

[P19]: “The new model can be extended to multi-phase mass flow simulation. It is now so convenient that the models (35)-(37) can be directly applied to the mechanical erosion model for two-phase mass flows developed by Pudasaini and Fischer²⁷ just by replacing λ^b by $2\lambda^b$ in the momentum balance equations in their model. Importantly, this also lays a foundation to further apply these methods to multi-phase mass flow model⁶¹ to include erosion-induced mobility.”

We must carefully move step-by-step for the meaningful scientific discovery with the clear vision of application and solving real problem. This is exactly what we are doing.

C3: To consider the interaction between the flow mass and the basal material, it is essential to clearly define the velocity in the erosion boundary, for instance, the velocity of landslide in the boundary and the velocity of erodible material. In this manuscript, the u which is averaged velocity of landslide, the u^b represents the velocity of erodible material. The u can not represent the velocity in the boundary. The dealing of this kind without a definite statement in the boundary is not clear for readers. The authors stated that the expression $F = d(mu)/dt = m * du/dt + u * dm/dt$ is wrong. It could easily mis-guide readers. Undoubtedly, the expression is correct as it is just the derivative operator. The way it could be wrong only people improperly use it. Currently, the authors did not clearly explain it to readers.

R3: On the definitions of the velocities on the boundary: This is strange that the reviewer could not see how clearly we have defined and mechanically considered the velocities on either sides of the erosion boundary, namely, the velocity at the flow bottom (on the upper side of the erosion interface), the averaged velocity u (which is the standard consideration in any admired models used in mass flow simulations in the literature), and the erosion velocity at the erodible basal substrate, u^b (on the lower side of the erosion interface). That was one of the main achievements in Pudasaini and Fischer (2020), that we utilized in this manuscript. The very essence of the erosion velocity, namely $u^b = \lambda^b u$, has been closed with the mechanically reach well defined analytical erosion drift model for λ^b in Pudasaini and Fischer (2020). This uniquely connects the dynamically derived flow velocity u in the upper side of the erosion interface with the mechanically obtained erosion velocity u^b on the lower side of the erosion interface. So, we have clearly defined boundary velocities.

On the use of the wrong momentum equation for the erosional landslide: The reviewer does not appear to perceive one of the main essences of this manuscript. We did not say that ‘the expression $F = d(mu)/dt = m * du/dt + u * dm/dt$ is wrong’. But rather we said:

[P7]: “The fundamental understanding here, as revealed by (9) is, that the increased inertia due to the increase in the mass of the landslide is not related to the velocity of the landslide u , but it is associated with the entrainment velocity, u^{ev} . Depending on the erosion situation, that we will discuss later, u^{ev} can be substantially less than the landslide velocity u . Thus, the true increased inertia $u^{ev} dm/dt$ can be much less than incorrectly proposed previously, $u dm/dt$. Hence, the classical, or the direct representation of $F = \frac{d}{dt}(mu)$ as $F = m\frac{du}{dt} + u\frac{dm}{dt}$ is fundamentally wrong for erosional situation.” This was the triggering point of this manuscript. And:

“However, it is evident, that the law (9) cannot be obtained directly by rearranging the inertial terms in the Newton’s second law of motion, but rather must be derived carefully by correctly considering the conservation of momentum for an erosional landslide as done above.” To further strengthen our statements, and make it clearer, we have added the following text in the revised manuscript [P6]: “It is important to note that $P_2 - P_1 = m\Delta u + (u - u^b)\Delta m$ is the main structure that any physically correct derivation must produce for the erosional landslide. This is clear from the two alternative derivations presented above. Moreover, at this point, it is crucial to realize, that the momentum equation (9) for the erosional landslide must be derived rigorously as done here by following the first-principles, but cannot just be speculated arbitrarily.” These are the scientific facts we have rigorously proven here for the first time for the erosional landslide.

C4: Lastly, the writing style is not proper for NC as too many equations derivation in detail. It is more proper for a professional journal not a comprehensive journal.

R4: We can only partly understand this concern. However, our mathematical derivations are simple, rigorous, and follow the physical facts of the erosional landslide. We have sufficiently explained every derivations and the mathematical-physical structures such that the targeted and wider geosciences audiences can easily follow the paper. We were very concerned about this. Because of its high-quality scientific contents, genuinely novel results of first-kind, potentially immediate, massive and sustained impacts, we are sure that our manuscript best fits to the NC. This way, the scientific community will benefit quickly and widely by publishing this paper in NC that can boost the results of great values.

Response to Reviewer # 3: MS #: NCOMMS-21-19648

Reviewer's comments and suggestions are in plane text indicated by C, our responses and enhancements are in blue text as indicated by R. Please see the improved manuscript text in accordance with the responses here in which the removed portions are in red cancel and edited and added texts are in blue colour. The manuscript is now in the format compatible with NC. The Abstract has been significantly condensed to approach the requirement of NC. P refers to the page number of the revised manuscript with track changes in colours.

Summary and key concepts

C: The present manuscript focuses on the mathematical modelling of landslides whose motion is affected by the entrainment of eroded material in the flowing mixture. It aims to provide a theoretical condition that may define when a landslide increases or reduces its velocity due to this physical process.

To achieve this goal, the authors considered as a starting point of their study a one-dimensional depth-integrated single-phase model written using the Eulerian description (Section 2). As a single-phase model, it is composed of a mass and a momentum balance principle, where the contribution of the eroded material to the landslide motion is evaluated in terms of mass and momentum production rates that are defined by specific algebraic closure relations. Although the effect of the entrainment of the eroded material has been already taken into account, the authors supposed that the inertial term of the momentum balance is incorrect since it does not consider that the mixture and the eroded material can have two distinct longitudinal velocities. Therefore, the authors derived a different formulation of the landslide inertia by using a discrete approach (Section 3), namely by identifying the mass and the velocity of both the landslide mixture and the eroded material at a first time (t_1) and the mass and the velocity of the landslide mixture at a subsequent time ($t_2 = t_1 + \Delta t$) after the entrainment of the eroded material. By writing the difference quotient of the landslide momentum and by approaching the time step Δt to zero, the landslide inertia is formulated in the Lagrangian description and depends on the relative velocity between the landslide mixture and the eroded material. This inertial term is defined by the authors as the correct formulation of the inertia of a landslide whose motion is affected by the entrainment of eroded material.

By introducing the reformulated inertia into the original momentum balance equation and by substituting in the resulting equation the mass balance principle, the authors derived an equation for the material derivative of the landslide velocity (Section 4). In this new equation, the contribution of the entrainment of the eroded material to the landslide velocity appear in a term that contains the factor $2u^b - u$, where u^b and u represent the longitudinal velocities of the eroded material and the landslide mixture, respectively. By comparing the two distinct longitudinal velocities appearing in the factor $2u^b - u$, the authors derived the theoretical condition that explains when a landslide may increase or reduce its velocity due to the entrainment of the eroded material (Sections 5 - 7). To show that the proposed formulation can detect an increase in the landslide velocity due to the entrainment of the eroded material, the authors computed the analytical solution for the equation of the material derivative of the landslide velocity and showed the results of this analytical solution for specific values of the parameters contained in the equation.

R: We very much appreciate the reviewer for precisely summarizing our work with important key concepts and novel findings that genuinely unveils a new science of mobility of an erosive landslide.

General considerations

C: The topic of the manuscript is interesting and might be useful for researchers working in the field of mass flow movements. However, I made a huge effort to understand the whole theory, since the way the manuscript is written and designed confuses the reader. The confusion arises from the fact that:

C 1): The definitions of some technical words used and/or first introduced in this work do not appear immediately in the text but after many pages. For example:

- a. in the Abstract and the Introduction sections, the words "erosion velocity", "entrainment velocity",

“energy generator” and “energy velocity” are used largely, but the explanations of the meaning of these words are provided in the subsequent pages (“erosion velocity” → page 5, “entrainment velocity” → page 6, “energy generator” → page 8, “energy velocity” → page 11). In this way, the comprehension of the text becomes difficult.

b. the definition of what the authors meant for erosion and entrainment as physical processes is reported only on page 11 in paragraph 7.2.

R: Thank you very much for supporting our work. In the revised manuscript, we have exclusively improved the writing style and eliminated all the confusions as far as we could spot them. Importantly, our sincere thanks to the reviewer for such an extraordinary review with outstandingly detailed and very constructive comments and explicit suggestions that truly resulted in the substantially improved revised manuscript in which we appropriately addressed all the concerns raised by the reviewer. We believe that the revised manuscript is now suitable to appear in Nature Communications soon.

R 1): a., b.: We fully agree with these comments. Due to the length constraint, it was not possible to define all these terms in the Abstract. So, we used the word ‘introducing’. The text has been revised and now reads: “By introducing three key-mechanical-concepts, we demonstrate that the erosion and entrainment are essentially different processes.” The definitions of all the technical words used and/or first introduced in this work now appear immediately in the text, 2nd paragraph in Results section [P4]:

“Next, we define some (mostly) fundamentally new mechanical concepts and terminologies as follows. This will help to comprehensively streamline the paper. Erosion: a mechanical process by which the bed material is mobilized by the flow. Entrainment: a mechanical process by which the eroded material is incorporated (entrained) and taken along with by the flow. Erosion-velocity: $u^e := u^b$, the velocity of the eroded material from the basal substrate. Entrainment-velocity: $u^{ev} := u - u^b$, the velocity of the entrained mass, or the velocity of the landslide minus the velocity of the eroded mass. Energy-velocity: $u^{ev} := u^b - (u - u^b) = u^e - u^{ev}$, the velocity associated with the net momentum production due to erosion that generates the excess (kinetic) energy, or the erosion velocity in excess to the entrainment velocity. Energy: associated with the erosion-induced net momentum production that results in the excess (kinetic) energy. Energy-generator (or, the mobility generator): the parameter that generates the excess energy (or the mass flow mobility) due to erosion. Energy-budget: the state of available excess energy generated by the net momentum production due to erosion. Erosion-rate: $E = -\partial b/\partial t$, the negative time rate of change of the basal topography, or the time rate of change of mass resulting in the mass production. The mechanical significance and the mathematical structures of these concepts and terminologies will be clearer in the due places when they appear in the text.”

Moreover, we have now defined all the variables and parameters in the beginning of the Results section, the 1st paragraph [P3-4]: “First we define variables and parameters. The landslide mass is denoted by m . The superscripts m and b represent the parameters or the quantities in the landslide (mixture), and the erodible basal substrate, respectively. For simplicity, we consider a geometrically two-dimensional flow. Let t be the time, (x, z) be the coordinates and (g^x, g^z) the gravity accelerations along and perpendicular to the slope, respectively. Let, h and u be the flow depth and the mean flow velocity along the slope. Similarly, $\gamma^m, \alpha^m, \mu^m$ be the density ratio between the fluid and the particles ($\gamma^m = \rho_f^m/\rho_s^m$), volume fraction of the solid particles (coarse and fine solid particles is represented by a single solid-phase: $\alpha^m := \alpha_s + \alpha_{fs}$, and $\rho^m := \rho_s^m$ is the effective density of the solid in the mixture), and the basal friction coefficient ($\mu^m = \tan \delta^m$) related to the basal friction angle δ^m , in the mixture. Analogously, $\rho^b = \rho_s^b$ is the density of the solid, $\gamma^b = \rho_f^b/\rho_s^b$ is the density ratio between the fluid and the particles and α^b is the solid volume fraction in the bed material. $\mu^b = \tan \delta^b$ is the Coulomb friction parameter associated with the erodible bed with the friction angle δ^b . Note that differences in μ^m and μ^b can result due to the difference in the landslide material and the bottom material. In simple situation, μ^m and μ^b can have similar values²⁷. Furthermore, $b = b(x; t)$ is the erodible bed surface that evolves in space and time, E is the basal erosion rate, u^b is the erosion velocity, K is the earth pressure coefficient as a function of the internal (ϕ^m) and the basal (δ^m) friction angles, and C_{DV} is the viscous drag coefficient.” These help to comprehensively and smoothly follow the new text.

C 2): The theoretical condition that the authors want to derive is obtained from equations that are written using different descriptions of the flow, i.e., the Eulerian (Section 2), the discrete (Section 3) and the Lagrangian descriptions (Sections 3-8).

R 2): Use of the Eulerian and Lagrangian descriptions has been made clear now in a new paragraph [P4]: “The Eulerian and Lagrangian descriptions are adopted here by purpose that will be clear in due places. First, we write the existing balance equations in standard and usual Eulerian form. Then, we utilize the most simple and convenient discrete procedure for the correct handling of the true inertia of the landslide with erosion. Classically, similar approaches have been used in deriving rocket equations. The Lagrangian descriptions are the most simple form that clearly demonstrate how erosion can result in substantially increased landslide velocity leading to enhanced mobility. Finally, the full set of erosional landslide model equations are written in the Eulerian description in the more sophisticated form that was largely facilitated by the Lagrangian descriptions.”

Moreover, the text already mentions:

[P8]: “For the full and better simulation of the erosive landslide, we must numerically integrate (12) together with (1) that includes evolution of both the flow velocity and the flow depth. This will be discussed later. Here, we are mainly interested in developing a simple model that can be solved analytically to highlight the main essence of erosion-induced energy (momentum) and the associated mobility of the landslide in terms of its velocity.”

[P12]: “Since here we aim to explicitly quantify the effect of erosion in landslide velocity, for simplicity, we consider (18) in Lagrangian form and obtain the exact landslide velocity. However, we mention, that the exact solution of (18) can also be obtained in Eulerian form, but is very demanding mathematically.”

C 3): The derivation of most of the equations is not rigorous since some hypotheses and some variables are not defined or are not correct. This will be clearer by reading the “Major comments” section of this document.

R 3): We thank the reviewer very much for pointing out this. There were some printing errors and some terms were not written correctly. As we will see later in the specific comments and responses (in “Major comments”), these have now been properly fixed. However, those deficiencies have no influence on the final form of the main model equations, only in some minor intermediate expressions, and all the important model equations were correct. All the hypotheses and variables have been defined now and are correct. These guarantee the rigourness of the derivations of all the equations in the revised manuscript.

C 4): Regarding the discussion of the theoretical condition for the increased or reduced mobility of the landslide due to the entrained material, the same concepts are presented in many different sections and what is changed is only the variable considered. In fact, the condition of increased/reduced mobility is presented:

- a. Firstly, in Section 5 (pages 8 and 9) using the ratio λ^b/λ^m and the factor $(2\lambda^b - 1)$ called “energy generator”.
- b. Secondly, in paragraph 7.3 (page 11) using the “energy velocity” u^{env} defined as $u^{env} = 2u^b - u = (2\lambda^b - 1)u$.

The reader might ask why there is the need to discuss in two parts the same results, since no extra information is presented.

R 4): We can fully understand this concern. However, for the completeness and to delineate three energy regimes, and make the text follow smoothly, we had some minimal overlap. The related text has been adequately improved/re-written, including the newly inserted text [P11]: “However, these concepts are systematically collected here for the better logical sequence and structural reasons. This comprehensively helps to distinguish between erosion and entrainment, and to formally delineate different energy or mobility regimes that will be explained later.” Furthermore, we tried to remove the repeated texts and information in the revised manuscript.

C: In addition to these aspects, in the “Major comments” section some specific and conceptual comments on

the work are reported and explained. The reference to the lines of the manuscript is made introducing a line numbering that starts for each page from 1.

R: Thank you very much for the many major and important comments. These really helped to increase the quality of the revised version of the manuscript. Below, we address all the concerns raised in the Major comments, which genuinely boosted the readability of the paper with proper and new concepts associated with the physical aspects, mathematical structures, and derivations. Furthermore, the revised manuscript is now more targeted to the broad audiences of the Nature Communications, and Geosciences scholars in particular.

Major comments

C 1): The title of this work is “The Mechanics of Landslide Mobility with Erosion”. I think that it is much more appropriate to use, not only in the title but also in the whole manuscript, the word “entrainment” rather than the term “erosion” for two reasons:

- The erosion, intended as a process that removes particles from the bottom (Frank et al., 2017; <https://doi.org/10.5194/nhess-17-801-2017>), produces changes in the bed topography, but this study does not focus the attention on how the changes in the bottom topography affect the mobility of the landslide;
- The aim of the present manuscript consists in analyzing how the mean velocity of the landslide is affected by the incorporation into the flow of a mass that moves with a specific velocity in the same direction of the flowing mixture. It is in this specific case that the entrained mass arises from an erosive process.

R 1): We can perceive these concerns. This seems to be so, and we also thought this way in the beginning. However, we prefer to retain the title, because it is clear from the process, derivation and the text that the basal erosion is the prime, that in a moment later, induces entrainment. The subheading [P10]: “The Outstanding Role of Erosion Velocity in Mass Flow Mobility” is entirely devoted for such description without any ambiguity. The way the erosion-velocity (u^e), entrainment-velocity (u^{en}) and energy-velocity (u^{env}) are defined later in a subheading [P11]: “Erosion-, Entrainment-, Energy-Velocity: New Mechanical Concepts”, their structures and evolving complexities demonstrate the fundamental and triggering role of erosion leading to entrainment and later to energy generation. The whole process is induced by erosion and it controls the landslide dynamics.

Bed erosion that directly produces changes in the bed topography (that has been explained in Pudasaini and Fischer, 2020) is also the case in this study. To make it explicit on how the changes in the bottom topography affect the mobility of the landslide, we have now explicitly mentioned (in several places, wherever necessary) on how the bed topography evolves with erosion as $\partial b/\partial t = -E$, this includes equations (3), (32), (37). Writing has been strengthened by inserting the following text [P5]: “Equation (3) makes a direct connection between the erosion process and the evolution of the bed by updating the basal topography with erosion, that in turn, explains how the changes in the bottom topography affect the mobility of the landslide.”

C 2): In the whole manuscript (Abstract and Introduction sections, Sections 5, 7 and 10, Summary section), the term “energy” is used largely, but a clear definition of what the authors meant with this term is missing. By reading the manuscript, it is possible to notice that a variable defined as a difference between two distinct velocities is used to define the “state of energy” of the landslide (Section 7, pages 10-11). However, the energy is a concept that is quite different from a velocity. Moreover, no analyses on the energy balance principle written either as a partial differential equation or as an algebraic balance like the Bernoulli’s principle are reported in this work. Thus, it is not possible to justify both the connection of the energy to a variable that has the dimension of a velocity and the discussion on the increase/decrease in the landslide energy due to the entrainment of mass.

R 2): We can fully understand the concern raised “the energy is a concept that is quite different from a velocity”. There might have been some confusions in the previous manuscript. First, in several places of the revised manuscript, to avoid confusion, the word “energy” has been replaced by “force” that is more appropriate (in Abstract, Introduction and Discussion). Second, we have now provided a clear link between the energy-velocity and energy. Erosion can result to increase the momentum of the landslide that escalates the kinetic energy of the slide. Kinetic energy involves (and is dominated by the) landslide velocity. So, the use of the

term energy-velocity (a term defined here and does not exist in the literature) is quite legitimate. Because this is an entirely new concept that may take a while until the scientific community will fully recognize its importance. This is how every new concept emerges, evolves, and establishes. The energy velocity is associated with the kinetic energy and has been clearly explained now in a subheading [P11]: “Erosion-, Entrainment-, Energy-Velocity: New Mechanical Concepts”. To make it clearer, we have added the following text in the revised manuscript, with a new paragraph [P11]: “The perception and structure of the energy-velocity is important. With the erosion rate E , for the landslide mass $m (= \rho^m h)$ (per unit flow length), the excess kinetic energy generated by u^{env} can be realized as $\mathcal{E}^{env} = \frac{1}{2} m u^{env} E$ which has the dimension of energy, because both u^{env} and E have the dimension of velocity. An expression similar to this can be obtained from the erosion-induced net momentum production, the last term on the right hand side of (11), which, when multiplied by the landslide mass ($m = \rho^m h$) results in $\mathcal{E}_h^{env} = (\rho^m h) (2u^b - u) E \frac{1}{h} = m(2u^b - u) E \frac{1}{h} = m u^{env} E \frac{1}{h}$. So, we can write, $\mathcal{E}_h^{env} = 2\mathcal{E}^{env} \frac{1}{h}$, which is twice the erosion-induced (excess) kinetic energy (resulting from u^{env}) normalized by the flow depth. This physically justifies the use of the term energy-velocity for the expression u^{env} , because it generates the erosion-induced excess energy and has the dimension of velocity. Note that depending on the sign of u^{env} , \mathcal{E}_h^{env} can be positive (for $u^{env} > 0$, or $\lambda^b > 1/2$) or negative (for $u^{env} < 0$, or $\lambda^b < 1/2$) resulting in the enhanced or the reduced mobility of the landslide. This has further been explained below. The energy-velocity $u^{env} = u^b - (u - u^b)$ is associated with the total (net) momentum production, with contribution u^b emerging from the reduced friction and $-(u - u^b)$ from the changed (reduced) inertia. It plays exclusively unique and dominant role in formulating the mobility equation and in determining the state of energy. Thus, the energy-velocity provides the universal picture of the erosion-induced mobility.”

On: “no analyses on the energy balance principle written”: The following text has been added with a new paragraph [P12]: “To deal with the more sophisticated situation we might also include the energy balance equation to complement the mass and the momentum balances. However, for now, we begin with the first-ever simple mechanical model capable of describing the excess energy of the landslide associated with erosion, where erosion induces the net momentum production giving rise to the excess kinetic energy.”

C 3): In the Abstract and Summary sections (page 1 line 22 and page 20 line 35), there appears the sentence “we demonstrate that the erosion and entrainment are essentially different processes”. A physical and mathematical proof is not reported in the manuscript. The distinction between the two processes (Section 7, pages 10-11) seems indeed to arise only from the definitions of:

- the erosion velocity u^b , that is the velocity of the eroded material incorporated into the landslide mixture and moving in the same direction of the landslide;
- the entrainment velocity $u - u^b$.

R 3): We have indeed proven this with the physical processes and the mathematical structures that the erosion and entrainment are essentially different phenomena. This is evident from the fact that erosion $u^e = u^b$ and entrainment $u^{en} = u - u^b$ are different in general. They can be the same only in a very special situation for which $u^b = \frac{1}{2}u$, which is very unlikely to occur in nature. This has been made clear now [P9].

C 4): In the Introduction (page 2, second paragraph, lines 16-34), the literature on erosion models is analyzed quickly without providing sufficient information on how these models take account of the processes of erosion and/or entrainment. In this way, the reader does not understand which is the precise lack in the literature. Moreover, some of the cited erosion models (Fraccarollo and Capart, 2002; Armanini et al., 2009; Iverson, 2012) are defined as single-phase models (lines 25-27), although they aren’t. They are indeed composed of two mass balances and one momentum balance rather than of one mass and one momentum balance principles. As a consequence, the cited models describe the flow of two layers or two phases that move with the same velocity, but this aspect does not mean that these models consider single-phase flows.

In addition, at lines 32-34 the erosion rates reported in some works are defined as “not based on the physical principles” or “physically inconsistent”, but no explanations for this is presented.

R 4): On “without providing sufficient information on how these models take account of the processes of erosion and/or entrainment”: This has been exclusively discussed in Pudasaini and Fischer (2020) that has been referred here. We could further elaborate on this also here, but think that that is not necessary, and hope the reference would serve for the interested readers.

On “but this aspect does not mean that these models consider single-phase flows”: We know well about all these models. This is why we said “effectively single-phase”, but not “single-phase”. The text has now been improved by writing softly as [P2] “effectively single-phase or at most quasi two-phase flows” instead of “effectively single-phase flows.”

On “not based on the physical principles” and “physically inconsistent”: We have sufficiently explained with references (Pudasaini and Fischer, 2020; de Haas et al., 2020). Our statement has been justified as we said in the Introduction [P2]: “However, the erosion rates presented and utilized in these works are either not based on physical principles or, these are physically inconsistent^{13,27}.” Our statements have been made further clearer by mentioning [P2]: “Specific physical shortcomings will be presented in Results and Discussion sections.” in the revised manuscript. This includes (mostly already existing descriptions):

[P6]: “This appears from an erroneous understanding of the erosional landslide, but prevails in many existing models, see, for example^{35,37,63}.”

[P7]: “Which, effectively means, that the classical approach works only for non-erosional situation, but not for landslide with erosion. So, those models, which are based on the unphysical formulation of the momentum balance, for example^{35,37,63}, are not appropriate in simulating landslide motion with erosion.”

[P11]: “Physically, this is impossible as proved by Pudasaini and Fischer²⁷, as $\lambda^b \neq 0$, however, this refers to the situation in many existing and influential erosion models^{25,35,37,46,47,61}.”

[P20]: “Most of the existing and influential erosion models^{25,35,37,46,47,63} do not include the momentum production in the momentum balance equation. These aspects have also been partially discussed with a mechanical erosion model for two-phase mass flow by Pudasaini and Fischer²⁷. Moreover, none of the existing models deal with erosion-induced changed inertia of landslide. Essentially, Perla et al.⁶³, McDougall and Hungr³⁵, and Iverson³⁷ directly replaced $\frac{d}{dt}(mu)$ with $m\frac{du}{dt} + u\frac{dm}{dt}$ in the inertial part of the momentum equation which is mechanically invalid for the erosive landslide. Therefore, these types of models result in the unphysical loss of energy associated with the erosive landslide, and hence, cannot properly explain the state of energy and mobility.”

C 5): In the whole manuscript, there is a large usage of the terms “correctly” (e.g. page 1, line 30), “precisely” (e.g. page 3, line 46), “well represent” (e.g. page 20, line 46), “physics-based” (e.g. page 8, line 17; page 20, line 43), “correct inclusion of the erosion” (e.g. page 14, line 6; page 21, line 10) in association with the model, the mechanical condition proposed and the results obtained in this work. However, it is not possible to state that both the model and the theoretical condition proposed are correct for sure, since no comparison of this study with laboratory experiments and real cases are reported.

R 5): Since the new model is based on the natural and physical first principles with rigorous mathematical formulations we are sure that the model will withstand the scrutiny against laboratory and field data. This is a fundamental paper with truly new concepts and principles. In general, it would be nice also to include some comparisons against the data, this is desirable to manifest the essence of the presented new science. This important aspect has already been mentioned: At the end of the Abstract [P1]: “We present a full set of dynamical equations in conservative form in which the momentum balance correctly includes the erosion-induced net momentum production.” And, at the Discussion section [P20]: “The application of the novel model to experimental and complex natural events of landslides, debris and avalanche motions would require substantial additional work, and corresponding parameter estimates, either derived from field measurements or back calculations, involving observation data, which, therefore, has to be deferred to some future contributions.”

However, it is not always possible to include all the aspects as we think in a single manuscript. Nevertheless, we

hope that the model will soon be applied to the laboratory and/or real field events. We expect this to happen soon after the publication of this paper, we are sure about this, as have been proven by some of our previous physical-mathematical articles that quickly became leading papers in the field and increasingly attracting the researchers in geosciences, engineering and applied mathematics in solving many great problems.

C 6): In Section 2.1 (page 4, lines 16-22), the equations (1)-(2) are derived from the three-phase model of Pudasaini and Mergili (2019), which in the one-dimensional depth-integrated case is composed of three mass balances and three momentum balances. The equations (1)-(2) are then derived assuming that the three phases move with the same velocity and, due to this aspect, the model is defined as single-phase. However, the assumption that three phases move with the same velocity does not mean that the mixture is a single-phase flow, and the model is single-phase. In addition to this aspect, there are some hypotheses that are missing.

- a. Assuming that the difference between the phase velocities is negligible, the number of momentum balances is reduced from 3 to 1. However, no hypotheses on how to reduce the number of mass balances is reported. Is this because the concentrations are assumed to be constant in time and space? If the answer to this question is affirmative, the real physics of “erosion” cannot be detected since this physical process produces changes over time and space in the concentrations of the different phases. Thus, this assumption should be declared.
- b. Also the viscous deformation of the fine solid phase needs to be assumed negligible;
- c. How are the densities of the coarse and fine solid phase? Are they assumed to be equal to each other?
- d. How is defined in the model (1)-(2) the bottom?

R 6: On “single-phase”: We have further assumed, for simplicity, that the solid and fine solid can be represented as a single solid-phase. This turns the mixture as composed of solid and fluid. We further assumed that the fluid viscous forces and deformation is negligible. Then, with the assumption of no relative velocity between the phases, the mixture effectively behaves as a single-phase material in which the fluid phase can still be parameterized by retaining the information of solid volume fraction, in simple situation, considered as a homogeneous mixture. So, at the moment, in total, we are dealing with the single-phase debris flow. To better represent this situation, in the revised manuscript we have replaced “single-phase” by “effectively single-phase”. The text has been improved accordingly [at P4] before equation (1).

- a. The same hypothesis produces the single mass balance without any extra conditions on concentration. This is so, because the true densities have already cancelled out from those mass balances. Then, when phase velocities are the same ($u_s = u_{fs} = u_f = u$), with the natural hold-up $\alpha_s + \alpha_{fs} + \alpha_f = \alpha_s^m + \alpha_f = 1$, this is straightforward to obtain the single mass balance for the mixture, as can be seen from the three mass balances in Pudasaini and Mergili (2019). To make it clear, we have now added a phrase [P4]: “since $\alpha_s + \alpha_{fs} + \alpha_f = \alpha_s^m + \alpha_f = 1$ ”, before equation (1). However, as in Pudasaini and Fischer (2020), erosion can be modelled for a single-, or a multi-phase mixture. As mentioned at the end of the Results section [P19], that the present model can thus be extended to a multi-phase erosion model. “The new model can be extended to multi-phase mass flow simulation. It is now so convenient that the models (35)-(37) can be directly applied to the mechanical erosion model for two-phase mass flows developed by Pudasaini and Fischer²⁷ just by replacing λ^b by $2\lambda^b$ in the momentum balance equations in their model. Importantly, this also lays a foundation to further apply these methods to multi-phase mass flow model⁶¹ to include erosion-induced mobility.”
- b. The solid and fine-solid phase is represented by a single solid phase. So, we don’t need to further assume the viscous aspect of the fine solid. It was a bit confusing in the previous text, but now it is clear.
- c. Again, as the solid and fine-solid phase is represented by a single solid phase, they have a representative solid density, ρ^m . To make it clear, in the revised text, we have defined it as [P4]: “(coarse and fine solid particles is represented by a single solid-phase: $\alpha^m := \alpha_s + \alpha_{fs}$, and $\rho^m := \rho_s^m$ is the effective density of the solid in the mixture)”. Furthermore, because the true densities are constant, these cancel out from the balance equations, what remains as a parameter is the ratio between the solid (single for solid plus

fine-solid) and the fluid density in the mixture defining the density ratio, $\gamma^m = \rho_f^m / \rho_s^m$ as a parameter.

d. The bottom topography and its effect has been consistently incorporated in the momentum balance in the revised manuscript as follows [P5]: “In simple situation, the bottom is defined with the locally inclined x -axis. This is a standard procedure involving erosion along the slope. To make the model more general, following Pudasaini and Mergili (2019)⁶¹, we have also included the term $-g^z \partial b / \partial x$ in the momentum balance (2) that incorporates the effect of the local variation of the topography (on top of the x -axis) on the flow dynamics. This superposition accounts for the detailed topographic information on top of the mean slope. Moreover, the time evolution of the basal surface due to erosion is given by $\partial b / \partial t = -E$ in (3).”

C 7): In Section 2.1 (page 4), the equation (2) is derived from the model of Pudasaini and Mergili (2019) by summing up the three momentum balance principles related to the three phases. However, the resulting equation is different from that derived in this manuscript. Defining α_s and α_{fs} (Pudasaini and Mergili, 2019) as the concentrations of the coarse and fine solid phases respectively, the term dependent on K at the left-hand side of the equation and the Coulomb friction at the right-hand side of the equation should depend on α_s rather than on α^m , which is defined (lines 11-12, page 4) as “the volume fraction of the solid particles (coarse and fine solid particles)”. Is there any assumption not declared?

R 7): Equation (2) is derived directly from the model of Pudasaini and Mergili (2019) by summing up the three momentum balance principles related to the three phases. The equation is correct, we have again checked it. As explained above, the solid and fine-solid is represented by a single solid phase with the volume fraction $\alpha^m = \alpha_s + \alpha_{fs}$. There is no extra assumption. To make it clear, we have now explicitly defined $\alpha^m := \alpha_s + \alpha_{fs}$ in the revised text [P4].

C 8): In Section 2.1 (page 4), the equation (3) is derived writing the momentum balance equation (2) in the non-conservative form. The term that in (2) depends on K should produce two terms: one containing $\partial h / \partial x$ and the other containing $\partial \alpha^m / \partial x$. However, the term depending on the gradient of the concentration does not appear in the equation. Is there any assumption on this concentration?

R 8): The revised text has been improved by explaining this aspect [P5]: “Because of the effectively single-phase nature of the model, either we can consider an extra closure relation for α^m , or parameterize or consider it as a constant. To keep the present basic model simple, we have assumed locally uniform mixture, so the local spatial variation of α^m can be ignored. However, it has been restored while constructing the full model at the end of this section.”

C 9): Section 2.2 (page 5) is confusing since lines 6-9 and 15-16 declare that the substitution of the mass balance equation (1) into the momentum balance equation (3) is wrong. As a consequence of this aspect, one could understand that the mass balance equation (1) is wrong rather than the momentum balance equation (2).

R 9): Thank you very much for this suggestion. The confusion has been eliminated by improving the writing in the new text, which now reads [P6]: “Below, we derive a physically correct momentum balance equation for an erosional landslide and prove that the direct substitution $u \left[\frac{\partial h}{\partial t} + \frac{\partial}{\partial x} (hu) \right] = uE$ in the inertial part of (4) results in a physically wrong momentum balance equation.”

C 10): In Section 3.1 (pages 5), the inertial term of the momentum balance principle is derived considering a discrete description of the landslide and the eroded material at two different times. The derivation creates confusion, since:

a. In this discrete approach, the entrained mass is defined as $-\Delta m$ (line 26).

b. The quantity P_2 is defined as “momentum of the landslide and the eroded mass at time t_2 ” (lines 25-26), but it is difficult to understand why this variable represents the momentum of both the landslide and the eroded mass as a whole. In fact, looking at the equation (5), it seems that P_2 expresses the

momentum of the landslide at the final time purified from the contribution of the entrained material rather than the momentum of both the landslide and the eroded mass as a whole. Is this correct or not? Why?

c. The sentence reported at lines 31-32, i.e., “In fact, the negative sign is not due to the negative change of the mass but rather due to the relative erosion velocity as compared to the actual velocity of the landslide”, is not clear.

R 10): We can understand the possible cause of confusion, and since this is the first paper to correctly handle the inertia for the erosive mass transport, not all aspects might have been clear, we hope in time it will be more clearer as the same expressions as presented here can, and will be derived later in many different ways. At a first glance, it might seem to be confusing. But, we have proven this alternatively by obtaining the same result below (6). The main point is the emergence of the physically valid structure for $P_2 - P_1 = m\Delta u + (u - u^b)\Delta m$ after (8). This is the main structure, that any physically correct derivation must produce. The revised text clearly mentions it as [P6]: “It is important to note that $P_2 - P_1 = m\Delta u + (u - u^b)\Delta m$ is the main structure that any physically correct derivation must produce for the erosional landslide. This is clear from the two alternative derivations presented above. Moreover, at this point, it is crucial to realize, that the momentum equation (9) for the erosional landslide must be derived rigorously as done here by following the first-principles, but cannot just be speculated arbitrarily.” So, different components of the landslide mass and eroded mass and the velocities can be arranged to establish their associations in many different ways resulting in the same model equation for landslide mobility that correctly incorporates the erosion induced inertia.

In the mean time, we have derived it in a further simple, yet more intuitive way as follows:

Let at time t_1 the landslide of mass m moves with the velocity u and the eroded mass Δm that has just been mobilized a moment ago moves with the erosion velocity u^b in which $u^b < u$. At the later time t_2 , the landslide with mass m strikes the little mass Δm with a perfectly inelastic collision, which is natural to happen. Consequently, the mass Δm is embedded in the landslide resulting in the entrainment. At this time, the total of the landslide mass and the entrained mass ($m + \Delta m$) moves together with a single velocity $(u + \Delta u)$, where Δu is the increment in the landslide velocity u . So, in the frame of reference of a stationary observer, the momentum P_1 at time t_1 and the momentum P_2 at time t_2 , respectively, are:

$$P_1 = m u + \Delta m u^b, \quad (1)$$

$$P_2 = (m + \Delta m) (u + \Delta u). \quad (2)$$

Since $P_2 - P_1 = mu + m\Delta u + u\Delta m + \Delta u\Delta m - mu - u^b\Delta m = m\Delta u + \Delta u\Delta m + (u - u^b)\Delta m \approx m\Delta u + (u - u^b)\Delta m$, we obtain the momentum equation for an erosional landslide:

$$F = m \frac{du}{dt} + (u - u^b) \frac{dm}{dt}, \quad (3)$$

in which, the higher order term $\Delta u \Delta m \ll 1$ is ignored. The above equation is the same as the erosional landslide equation in the present manuscript, however, now the derivation is much simpler. Nevertheless, we prefer to report this new derivation elsewhere as a (brief) paper. We hope the reviewer agrees with us. Otherwise, if the reviewer suggests it would be better also to put the third derivation in the manuscript we are equally happy to do that.

So, all three derivations, two in the paper, and this extra presented above, lead to the same result. This proves that we are physically and mathematically fully consistent. This is why we call (5)-(6) the most elegant formulation. Physical correctness of the model derivation with subheading “Correct Derivation of the Relevant Momentum Balance Equation” [P6] has been clearly demonstrated by (9) and all further derivations and discussions.

a. The entrained mass has been defined legitimately, because, $m + (-\Delta m) + \Delta m = m$ is the actual mass after entrainment. So, Δm is later added by erosion.

b. On “Is this correct or not?”: No. Because, at time t_2 , $m + (-\Delta m) + \Delta m = m$ is the landslide mass and Δm is the eroded mass. This has been explicitly proven in (7). Please also see the new derivation above and physical justification.

c. This has been made clear by improving the writing [P6]: “In fact, the negative sign is not due to the negative change of the mass but rather due to the component $(-\Delta m)$ in the mass at time t_1 . This is consistent with the momentum conservation for the erosional landslide. We make this clearer now.” This is so because, the alternative derivation in (7), including the text afterwards, and also the new derivation above, which are more intuitive than in (5)-(6), all prove this.

C 11): In Section 3.1 (pages 5-6), is there any assumption concerning the velocity of the eroded mass u^b ? The reason for this question lies in the fact that the equation (8) can be derived from the classical definition of the inertia as follows: $\frac{d(mu)}{dt} = \frac{d}{dt}[m(u - u^b)] = m\frac{d}{dt}(u - u^b) + (u - u^b)\frac{dm}{dt} = m\frac{du}{dt} + (u - u^b)\frac{dm}{dt}$ where u^b is assumed to be constant in time and space.

If this assumption is correct also for the formulation reported in the manuscript, it is in contradiction with the closure relation used, i.e., $u^b = \lambda^b u$, since the mean flow velocity u can change over time and space.

R 11): On “is there any assumption concerning the velocity of the eroded mass u^b ?”: No, there is no assumption on the velocity of the eroded mass u^b .

On: “where u^b is assumed to be constant in time and space”: No. So, since u^b in general is a variable, the reviewer’s proposed equation is not valid for erosive landslide, because: $\frac{d(mu)}{dt} \neq \frac{d}{dt}[m(u - u^b)]$ and $m\frac{d}{dt}(u - u^b) + (u - u^b)\frac{dm}{dt} \neq m\frac{du}{dt} + (u - u^b)\frac{dm}{dt}$.

On “the closure relation used, i.e., $u^b = \lambda^b u$ ”: This closure relation is valid. Also see the text before (12) for the generality of this expression.

C 12): In Section 3.1 (page 6, line 7), the variable $u^{ev} = u - u^b$ is defined as “the velocity of the entrained mass”. But on page 5 at line 24, it is declared that the mass entrained is the erosion velocity u^b . This creates confusion.

R 12): This confusion has been eliminated now by writing [P6]: “the other portion of the mass $(-\Delta m)$ moves with the erosion velocity u^b .”

C 13): In Section 4 (page 7) the inertial term derived is substituted into the momentum balance principle. This substitution is not rigorous, since:

a. In the equation (9) no definition of the variables contained in m is reported.

b. At line 11, the definition of m as $m = 1$ is not correct, since using $m = 1$, $\frac{dm}{dt} = 0$ and not $\frac{dm}{dt} = E$.

R 13): We thank the reviewer very much for pointing out these carelessness, and writing mistakes, which have been fixed and the revised manuscript is rigorous now. However, the main model remains unchanged.

a. As a notation, m has been mentioned early in the Results section [P3], and now recalled again [P7] [after equation (10)], as the landslide mass.

b. The landslide mass m has now been correctly defined as $m = \rho h$, per unit length (because of two dimension). Since the density ρ is not directly involved in the momentum balance equation, we effectively have $m = h$. With this, $\frac{dm}{dt} = \frac{dh}{dt} = \frac{\partial h}{\partial t} + \frac{\partial}{\partial x}(hu) = E$, and equation (11) follows as it was before. So, the final result (11) remains unchanged. The related text around (10) and (11) has been amended accordingly: “mass (per unit channel length) $m = h$ (because the material density cancels out in the momentum balance (2) or (4) under consideration, so we should take $m = h$ instead of $m = \rho^m h$) resulting in $\frac{dm}{dt} = \frac{dh}{dt} = \frac{\partial h}{\partial t} + \frac{\partial}{\partial x}(hu) = E$, which is (1),”

C 14): In the equation (16) of Section 4 (page 8), there appear two variables that were not declared in the previous pages and that are not defined in this point. These variables are ρ^m and ρ^b . Since at page 4 (line 11), the reader can see that the phases can have different densities (ρ^f, ρ^s), how are defined ρ^m and ρ^b ? Moreover, from the physical point of view, which is the difference between the Coulomb friction coefficients μ^m and μ^b appearing in the equation (16)? Is this because the landslide material and the bottom material could be different? If the two materials are different, how is this difference modelled when the material from the bottom is incorporated into the flowing mixture?

R 14): At the beginning of the Results section, we have mentioned that all the variables/parameters with m represent those in the flow material and all the variables/parameters with b represent those in the bed material. To make it clearer, the text has been improved as [P4]: “volume fraction of the solid particles (coarse and fine solid particles is represented by a single solid-phase: $\alpha^m := \alpha_s + \alpha_{fs}$, and $\rho^m := \rho_s^m$ is the effective density of the solid in the mixture), and the basal friction coefficient ($\mu^m = \tan \delta^m$) related to the basal friction angle δ^m , in the mixture. Analogously, $\rho^b = \rho_s^b$ is the density of the solid, $\gamma^b = \rho_f^b / \rho_s^b$ is the density ratio between the fluid and the particles and α^b is the solid volume fraction in the bed material. $\mu^b = \tan \delta^b$ is the Coulomb friction parameter associated with the erodible bed with the friction angle δ^b . Note that differences in μ^m and μ^b can result due to the difference in the landslide material and the bottom material. In simple situation, μ^m and μ^b can have similar values²⁷. Furthermore, $b = b(x; t)$ is the erodible bed surface that evolves in space and time, E is the basal erosion rate, u^b is the erosion velocity, K is the earth pressure coefficient as a function of the internal (ϕ^m) and the basal (δ^m) friction angles, and C_{DV} is the viscous drag coefficient.”

On the difference between μ^m and μ^b : This has been explained now as [see above response]: “Note that differences in μ^m and μ^b can result due to the difference in the landslide material and the bottom material. In simple situation, μ^m and μ^b can have similar values²⁷.”

On the differences between the parameters and the parameter/variable updating in the mixture: These important aspects have now been added in the revised text as [P9, after equation (17)]: “Differences in the material parameters across the erosion interface (between the landslide and the bed material) results in erosion²⁷. Furthermore, with entrainment, values of ρ^m, γ^m, μ^m and α^m should be appropriately updated in proportion to the newly entrained material to the sliding material.”

C 15): In Section 4 (page 8, lines 24-29), there appears the sentences “First, in reality, λ^b lies in a close or broader neighborhood of $1/2$ that is contained in $(0, 1)$. So, the legitimate values of λ^b is around $1/2$ ”. There is no explanation and proof of these two sentences.

R 15): In the manuscript, we have provided reference for this and think this should be sufficient [P9]: “This has been proven and explicitly explained in Pudasaini and Fischer²⁷.” [Section 5.1 therein].

C 16): At the end of paragraph 5.4 (page 10), there is a discussion of the “technically suitable natural domain of λ^b ”, which is defined as “ $(-\Delta\lambda + 1/2, 1/2 + \Delta\lambda)$, where $\Delta\lambda$ is a small positive number, say, typically $1/4$ ”. Have been the range for λ^b and the “typical” value of $\Delta\lambda$ validated? Do they derive from results of laboratory experiments?

R 16): This is a mathematically nice way to form a close neighborhood of $1/2$. The text has now been improved with [P10]: “The range of values of λ^b and $\Delta\lambda$ may depend on the erosion situation. This is a technical aspect that needs to be properly handled during the model application to laboratory and/or field data. The drift equation (19) provides a practically useful mechanical closure for λ^b , and thus for $\Delta\lambda$.”

C 17): In Section 7 (page 10):

- a. The “erosion-velocity” is defined as $u^e = u^b$. It is difficult to understand why it is necessary to define with a new different symbol for the same physical variable.
- b. The “energy velocity” is defined as $u^{env} = 2u^b - u$ and as the variable that expresses the state of energy of the landslide. As specified in the comment 2 of the “Major comments” section, this is a definition provided only in this manuscript and does not derive from proofs on the energy balance

principle. Moreover, talking about this variable, at page 11 at lines 1-2, it is declared that energy velocity enhances the mobility, but this aspect is true only if $\lambda^b > 1/2$ as declared in paragraph 7.3.

R 17): On the definition of the “erosion-velocity”, and “energy velocity”:

a. The definition of the new variable is due to the requirement of the logical sequence: u^e, u^{ev}, u^{env} , for erosion-, entrainment- and energy-velocity. This is structurally nice. The text at the beginning of subheading “Erosion-, Entrainment-, Energy-Velocity: New Mechanical Concepts” has been improved as [P11]: “However, these concepts are systematically collected here for the better logical sequence and structural reasons. This comprehensively helps to distinguish between erosion and entrainment, and to formally delineate different energy or mobility regimes that will be explained later.”

b. The energy-velocity is an entirely new concept introduced in this manuscript. It’s relevance and importance has been stated in the text [P11] and also in the previous response Comment C2, Major comments. Here, we have provided a simple structure with the physical process on how this energy-velocity in fact relates to the excess energy associated with the erosion induced net momentum production in the landslide. This should serve as the first-ever physics-based simple model. However, in later developments involving the more sophisticated situations, the balance equation for energy can be added to the mass and momentum balance equations developed in this manuscript, which, however, is not within the scope here.

Moreover, the manuscript has been improved by inserting [P11]: “Note that depending on the sign of u^{env} , \mathcal{E}_h^{env} can be positive (for $u^{env} > 0$, or $\lambda^b > 1/2$) or negative (for $u^{env} < 0$, or $\lambda^b < 1/2$) resulting in the enhanced or the reduced mobility of the landslide. This has further been explained below.”

C 18): In paragraph 8.1 (page 11, lines 38-40), the sentences “However, numerical solutions are always subject to questions as such solutions are based on some assumptions and applied approximations that may not follow the laws of nature. So, the physical relevant exact solutions are superior over the numerical simulations (Pudasaini and Krautblatter, 2021)” are not totally correct, since the analytical solutions can be used in order to derive which numerical method is the most accurate in that specific situation. There could be indeed numerical methods that are very accurate.

R 18): Following this suggestion, the text has been improved also to indicate the importance of the numerical simulations and their connections to the analytical solutions as follows [P12]: “So, in general the physically relevant exact solutions are superior over the numerical simulations⁶⁷. Nevertheless, the numerical solutions can cover the broad spectrum of the complex flow dynamics, and once tested and validated against the analytical solutions, may provide even more accurate results than the simplified analytical solutions.”

C 19): In paragraph 8.1 (page 12), the analytical solution (19) is derived from the equation (15), which contains a term depending on $\partial h / \partial x$. At page 7, it is declared that the authors “parametrize the landslide (or, the flow) depth”. It is not clear if this parametrization means either a constancy in time and space in the depth or a constancy in time or a constancy in space. Moreover, it is not declared how this parametrization affects the mass balance equation.

R 19): The revised text now makes it clear that for the main model (around equation (13)): “the parametrization (mainly in space, see later the main model (18))”. The main landslide model equation (18) is derived for the erosion enhanced velocity by combining the mass and momentum balance equations and the parametrization is implicit in \mathcal{A} . By combining (18) with the mass balance equation (1), one can investigate the effect of such parametrization. We are working on this separately that would be too much to report here. However, once we know the structure of the erosion induced mobility, the complete set of dynamical equations later in this section can be used without any parameterization. This clearly demonstrates the importance of parametrization in better understanding the intrinsic structure of the erosion induced process that later helped to present the full model without such parametrization. This has been explained after (12) as: “For the full and better simulation of the erosive landslide, we must numerically integrate (12) together with (1) that includes evolution of both the flow velocity and the flow depth. This will be discussed later. Here, we are mainly interested in

developing a simple model that can be solved analytically to highlight the main essence of erosion-induced energy (momentum) and the associated mobility of the landslide in terms of its velocity.”

C 20): In paragraph 8.2 (page 12, line 15), the equation (20) defines the “steady-state (uniform) velocity of the landslide”. In this definition, the two terms “steady” and “uniform” seem to be used as synonyms. However, “steady” means a constancy in time, while “uniform” means a constancy in space.

R 20): The text has been revised as suggested by removing “(uniform)” [P13].

C 21): In paragraph 8.2 (page 12, lines 42-43), there appears the sentence “By properly selecting the model parameters such exceptionally high velocities as inferred from the field can be obtained from the new model”. This sentence cannot be demonstrated since no definitions of the physical range of these parameters are reported. It is necessary to state that the parameters need to be calibrated carefully for example by reproducing laboratory experiments or by computing back analyses.

R 21): We agree, and the text has been improved accordingly, and now reads as [P13]: “By properly selecting the model parameters such exceptionally high velocities as inferred from the field (as mentioned above) can be obtained from the new model. Yet, the model must further be scrutinized with carefully calibrated parameters by reproducing laboratory experiments and back analyzing the natural events, which however, is not within the scope here.”

C 22): In the Summary section (page 20, lines 45-47), there appears the sentence “Analytically obtained values well represent the velocity of natural landslides and debris avalanches with erosion and demonstrate that erosion can have the major control on the landslide dynamics.” This sentence is not demonstrated. In fact, no estimations of the velocity of natural landslide using the analytical solution proposed in this manuscript are reported.

R 22): On “Analytically obtained values well represent the velocity of natural landslides and debris avalanches with erosion”: The text has been modified and now reads [P21; last paragraph in Discussion]: “We have (implicitly) shown that analytically obtained values well represent the velocity of natural landslides and debris avalanches with erosion and demonstrated that erosion can have the major control on the landslide dynamics.” On “demonstrate that erosion can have the major control on the landslide dynamics”: We think, this is clear from the analytical solution and the associated results. So, we hope no further explanation is required. Otherwise, we could have removed the word “demonstrated”.

C 23): In the Abstract section (page 1, line 9), there appears the sentence “Erosion, as a key control of landslide dynamics, significantly increases the destructive power by rapidly amplifying its volume, mobility and impact energy”. A similar sentence appears also in the Summary section (page 20, lines 7-8). However, in this work, according to the velocity of the eroded material u^b compared to the landslide velocity u , the landslide mobility can also decrease.

R 23): We have presented the explicit analyses both for the enhanced and reduced mobility related to the landslide erosion. Yet, erosion induced enhanced mobility is the most possible scenario, and the main focus has been streamlined for the enhanced mobility. However, to make the writing more appropriate, we have now re-phrased as: [1st sentence in Abstract, P1]: “Erosion can significantly increase the destructive power by rapidly amplifying its volume, mobility and impact-force.” And [in Discussion, P20]: “Mobility of an erosive landslide can be attributed to its excessive volume and material properties, and is marked by rapid motion, exceptional travel distance and the inundation area.”

C 24): In the Summary section (page 21, lines 1-3), there appears the sentence “Similarly, the large contrast in velocity with and without erosion results in completely different run-out scenarios, much more extensive for erosive landslide.” This sentence is not totally correct, since, as explained in comment 25, according to the velocity of the eroded material u^b compared to the landslide velocity u , the landslide mobility can decrease, and thus, the run-out with erosion could be smaller than that without erosion.

R 24): This sentence has been written appropriately and now reads as [last paragraph in Discussion, P21]:

“Similarly, the large contrast in velocity with and without erosion can result in completely different run-out scenarios, much more extensive for erosive landslides.”

Minor comments:

C 1): In section 5 (page 9), I suggest defining in a more precise way that the value $1/2$ that distinguishes between increased or reduced mobility arises from the term in the round brackets of equation (18) when $\rho^b \alpha^b = \rho^m \alpha^m$. Moreover, I suggest inverting the discussion of the paragraphs 5.1, 5.2 and 5.3 starting from the paragraph 5.3.

R 1): We follow these important suggestions. This is very important to provide a critical (reference) value to distinguish between the enhanced and reduced mobility. We have now inserted the following sentence [P10]: “Moreover, note that the value $\lambda^b = 1/2$ that distinguishes between increased or reduced mobility arises from the term inside the brackets of equation (19) when $\rho^b \alpha^b = \rho^m \alpha^m$.” Furthermore, we have changed the order of the Section numbering as 5.3 to 5.1 and vice versa [P10]. This way, the text appears to be logically better structured. Similar logical changes have also been made wherever necessary, e.g., on [P12].

C 2): In the Abstract, Introduction, and from Section 8 to Section 11, I suggest declaring that the results of the analytical solutions are derived in this manuscript considering only the case where there is an increase in the landslide velocity due to the entrainment of eroded material.

R 2): The text has been improved accordingly in the Introduction, Results and Discussion sections. This includes changes as [P3, end of Introduction]: “We analytically obtained the landslide velocity, where there is an increase in landslide velocity due to the entrainment of eroded material by solving the landslide mobility equation that quantifies the effect of erosion in landslide mobility.”, [P12]: “Here, we focus on the velocity of an erosive landslide considering the case where there is an increase in the landslide velocity due to the erosion induced entrainment, i.e., for $\mathcal{P}_M > 0$ corresponding to $\lambda^b > 1/2$. However, analytical results for $\mathcal{P}_M < 0$ corresponding to $\lambda^b < 1/2$ resulting in reduced mobility even for an erosive landslide can be obtained and discussed similarly.”, and [last paragraph Discussion, P21]: “We explicitly obtained the landslide velocity by analytically solving the mobility equation for the erosion induced increased landslide velocity.” However, it appears with the revised text that it is not absolutely necessary to make further changes in the Abstract, also due to the length of the Abstract.

C 3): Some typos in the references appear.

R 3): The References has been checked thoroughly and improved.

REVIEWER COMMENTS

Reviewer #3 (Remarks to the Author):

In the attached file, it is possible to find the review report.

REVIEWER COMMENTS

Reviewer #3 (Remarks to the Author):

I thank the authors for the answers to the first review of the manuscript and for considering most of my suggestions. The responses to some of my comments and the related changes in the manuscript can be considered acceptable. However, some doubts and concerns about the proposed theory still remain. Furthermore, the writing style seems to be not so objective. In many parts of the manuscript, the proposed theory is defined as correct, but there is no real proof that it represents accurately the real process of erosion. All the doubts and concerns about the theory and the comments on the writing style will become clearer, in the following, by reading the comments reported in the “Major comments” section. In addition, in the “Minor comments” section, some other suggestions are reported.

In this review, the new comments are denoted by the word “**Comment**” and in some of them, there appear both the comments of the previous review (in red) and the related authors' answers (in blue).

MAJOR COMMENTS:

Comment 1:

C 11): In Section 3.1 (pages 5-6), is there any assumption concerning the velocity of the eroded mass u^b ? The reason for this question lies in the fact that the equation (8) can be derived from the classical definition of the inertia as follows: $\frac{d(mu)}{dt} = \frac{d}{dt} [m(u - u^b)] = m \frac{d}{dt} (u - u^b) + (u - u^b) \frac{dm}{dt} = m \frac{du}{dt} + (u - u^b) \frac{dm}{dt}$ where u^b is assumed to be constant in time and space. If this assumption is correct also for the formulation reported in the manuscript, it is in contradiction with the closure relation used, i.e. $u^b = \lambda^b u$, since the mean flow velocity u can change over time and space.

R 11): On “is there any assumption concerning the velocity of the eroded mass u^b ?”: No, there is no assumption on the velocity of the eroded mass u^b .

On: “where u^b is assumed to be constant in time and space”: No. So, since u^b in general is a variable, the reviewer's proposed equation is not valid for erosive landslide, because:

$$\frac{d(mu)}{dt} \neq \frac{d}{dt} [m(u - u^b)] \text{ and } m \frac{d}{dt} (u - u^b) + (u - u^b) \frac{dm}{dt} \neq m \frac{du}{dt} + (u - u^b) \frac{dm}{dt}.$$

On “the closure relation used, i.e. $u^b = \lambda^b u$ ”: This closure relation is valid. Also see the text before (12) for the generality of this expression.

In the comment in red, the inertial term of the momentum balance principle is derived considering a reference frame that is moving with the eroded mass with a constant velocity u^b (\rightarrow inertial reference frame). This expression for the inertial term coincides with the inertial term that appears at the right-hand side of equation (9) of the manuscript. The only difference between the two expressions is related to the condition on the velocity of the eroded material. In the red comment, this condition coincides with $u^b = \text{const}$ and is part of the proof. Conversely, in the manuscript, it is defined by $u^b = \lambda^b u$ and is applied only after the derivation of the inertial term.

Can the authors explain why it is possible to obtain the same expression for the inertial term from two different starting points? Since there is a discrepancy between the two conditions of the velocity of the eroded material, might the proof provided in the red comment highlight an important aspect of the proposed theory not visible using the discrete approach of the section “Correct Derivation of the Relevant Momentum Balance Equation”?

I am afraid that there might be something missing in the theory proposed. Since the condition on the velocity of the eroded material is the key element of the whole work, I suggest carefully considering this comment and justifying the reply on it in detail.

Comment 2:

C 3): In the Abstract and Summary sections (page 1 line 22 and page 20 line 35), there appears the sentence “we demonstrate that the erosion and entrainment are essentially different processes”. A physical and mathematical proof is not reported in the manuscript. The distinction between the two processes (Section 7, pages 10-11) seems indeed to arise only from the definitions of:

- the erosion velocity u_b , that is the velocity of the eroded material incorporated into the landslide mixture and moving in the same direction of the landslide;
- the entrainment velocity $u - u_b$.

R 3): We have indeed proven this with the physical processes and the mathematical structures that the erosion and entrainment are essentially different phenomena. This is evident from the fact that erosion $u^e = u^b$ and entrainment $u^{en} = u - u^b$ are different in general. They can be the same only in a very special situation for which $u^b = \frac{1}{2}u$, which is very unlikely to occur in nature. This has been made clear now [P9].

I have still some doubts about the fact that this manuscript demonstrates that “erosion and entrainment are different processes”. In my opinion, it is how the erosion and entrainment velocities are defined that makes the two processes different from each other. The relative velocity between the landslide and the eroded material $u - u_b$ is CALLED, in this manuscript, as “entrainment velocity”, but it is not explained clearly WHY this relative velocity should represent the velocity of the entrained mass.

Can the authors explain this aspect by making some simple physical examples? They could be very helpful for the reader.

Comment 3:

C 7): In Section 2.1 (page 4), the equation (2) is derived from the model of Pudasaini and Mergili (2019) by summing up the three momentum balance principles related to the three phases. However, the resulting equation is different from that derived in this manuscript. Defining α_s and α_{fs} (Pudasaini and Mergili, 2019) as the concentrations of the coarse and fine solid phases respectively, the term dependent on K at the left-hand side of the equation and the Coulomb friction at the right-hand side of the equation should depend on α_s rather than on α^m , which is defined (lines 11-12, page 4) as “the volume fraction of the solid particles (coarse and fine solid particles)”. Is there any assumption not declared?

R 7): Equation (2) is derived directly from the model of Pudasaini and Mergili (2019) by summing up the three momentum balance principles related to the three phases. The equation is correct, we have again checked it. As explained above, the solid and fine-solid is represented by a single solid phase with the volume fraction $\alpha^m = \alpha_s + \alpha_{fs}$. There is no extra assumption. To make it clear, we have now explicitly defined $\alpha^m = \alpha_s + \alpha_{fs}$ in the revised text [P4].

I try to reformulate comment C7 of the first review since it might be not so clear. I think that the equation (2) is derived from Pudasaini and Fischer (2020) rather than from Pudasaini and Mergili (2019). This aspect arises from the fact that in Pudasaini and Mergili (2019), the coarse solid and fine-solid phases are different materials defined as “a Mohr-Coulomb continuum” and a viscous fluid, respectively. As a consequence of this different behaviour of the two solid phases, in Pudasaini and Mergili (2019), only the solid pressure term related to the coarse-solid phase (and not both the solid pressure terms for the coarse and fine solid phases) depends on the earth pressure coefficient K . This can be seen by combining the fourth terms at the left-hand side of the equations (14a)-(14b) of Pudasaini and Mergili (2019, p. 2930) with equation (18) in Pudasaini and Mergili (2019, p. 2931). That is why my previous comment C7 stated “the term dependent on K at the left-hand side of the equation and the Coulomb friction at the right-hand side of the equation should depend on α_s rather than on α^m ”.

Conversely, if we start the derivation the momentum balance equation from Pudasaini and Fischer (2020), we obtain equation (2). This is because the single solid phase (and thus, both the coarse and fine-solid phase) is defined as a Mohr-Coulomb continuum.

I suggest checking again the derivation and if this comment is wrong, please demonstrate it in detail. Moreover, I suggest checking that the references in the whole work are written in the right places.

Comment 4:

On page 9 after equation (17) of the new version of the manuscript, there appear the sentence “Furthermore, with entrainment, values of ρ^m ; γ^m ; μ^m and α^m should be appropriately updated in proportion to the newly entrained material to the sliding material”. It is important to stress that if ρ^m needs to be updated on the basis of the “amount” of the material entrained, ρ^m becomes variable in time and space. Thus, ρ^m does not cancel out both in the mass balance equation (1) and momentum balance equations (2) and (4). I suggest highlighting this aspect since the reader might understand that the theory proposed does not need any type of rearrangement of the equations.

Comment 5:

C 22): In the Summary section (page 20, lines 45-47), there appears the sentence “Analytically obtained values well represent the velocity of natural landslides and debris avalanches with erosion and demonstrate that erosion can have the major control on the landslide dynamics.” This sentence is not demonstrated. In fact, no estimations of the velocity of natural landslide using the analytical solution proposed in this manuscript are reported.

R 22): On “Analytically obtained values well represent the velocity of natural landslides and debris avalanches with erosion”: The text has been modified and now reads [P21; last paragraph in Discussion]: “We have (implicitly) shown that analytically obtained values well represent the velocity of natural landslides and debris avalanches with erosion and demonstrated that erosion can have the major control on the landslide dynamics.”

On “demonstrate that erosion can have the major control on the landslide dynamics”: We think, this is clear from the analytical solution and the associated results. So, we hope no

further explanation is required. Otherwise, we could have removed the word “demonstrated”.

It seems that the authors wanted to show that the theory is good by comparing the velocities obtained from the analytical solution (first paragraph on page 13 of the new version of the manuscript) with the estimated velocities of some natural landslides and debris avalanches (paragraph that starts at the end of page 12 and continues on page 13 of the new version of the manuscript).

However, this comparison has no sense since the estimated velocities of natural landslides can be produced by conditions that are far away from those from which the analytical solutions are derived. More precisely, the proposed analytical solutions are derived considering a simple situation, i.e., the motion happens down a slope with no changes in the bottom profile (sentences reported on page 5 “*For the purpose of developing a simple landslide mobility equation, momentarily we consider the motion down an inclined slope. In this situation $\frac{u}{|u|} = 1$ and $-g^z \frac{\partial b}{\partial x} = 0$* ”). Conversely, in real conditions, the topography could be much more complex than a slope. As a consequence, the estimated velocities of natural landslides can be very different from the velocities computed from the proposed analytical solutions.

The only way to demonstrate the sentence “*analytically obtained values well represent the velocity of natural landslides and debris avalanches with erosion*” and to define the theory as “correct” consists in comparing the velocities of the analytical solutions with some results of laboratory experiments performed in the same conditions of the analytical solutions, i.e. considering a mixture that flows down a slope.

For this reason, I suggest deleting the sentence “*We have (implicitly) shown that analytically obtained values well represent the velocity of natural landslides and debris avalanches with erosion*” from the Summary and the sentence “*Obtained velocities are representative of natural landslides with erosion and indicate the fact that erosion can have the major control on the landslide dynamics*” from the Introduction.

MINOR COMMENTS:

Comment 1:

On pages 4 and 5 of the new version of the manuscript, there appear the sentences “*Because of the effectively single-phase nature of the model, either we can consider an extra closure relation for α^m , or parameterize or consider it as a constant. To keep the present basic model simple, we have assumed locally uniform mixture, so the local spatial variation of α^m can be ignored. However, it has been restored while constructing the full model at the end of this section.*”

The assumption of uniform mixture ($\alpha_s = const, \alpha_{fs} = const, \alpha^m = const$) is valid also for the set of equations (1)-(2) since no equations for the changes over time and space in the concentrations are provided.

I suggest:

1. moving “*Because of the effectively single-phase nature of the model, either we can consider an extra closure relation for α^m , or parameterize or consider it as a*

constant. To keep the present basic model simple, we have assumed locally uniform mixture" on page 4 after the sentence "This means, to facilitate for the derivation of a simple model, with these assumptions, we are considering an effectively single-phase mixture flow".

2. writing "due to the uniform mixture assumption, the local spatial variation of α^m can be ignored. However, it has been restored while constructing the full model at the end of this section" on page 5 before equation (4).

Comment 2:

C 10): In Section 3.1 (pages 5), the inertial term of the momentum balance principle is derived considering a discrete description of the landslide and the eroded material at two different times. The derivation creates confusion, since:

- a. In this discrete approach, the entrained mass is defined as $-\Delta m$ (line 26).
- b. The quantity P_2 is defined as "momentum of the landslide and the eroded mass at time t_2 " (lines 25-26), but it is difficult to understand why this variable represents the momentum of both the landslide and the eroded mass as a whole. In fact, looking at the equation (5), it seems that P_2 expresses the momentum of the landslide at the final time purified from the contribution of the entrained material rather than the momentum of both the landslide and the eroded mass as a whole. Is this correct or not? Why?
- c. The sentence reported at lines 31-32, i.e., "In fact, the negative sign is not due to the negative change of the mass but rather due to the relative erosion velocity as compared to the actual velocity of the landslide", is not clear.

R 10): We can understand the possible cause of confusion, and since this is the first paper to correctly handle the inertia for the erosive mass transport, not all aspects might have been clear, we hope in time it will be more clearer as the same expressions as presented here can, and will be derived later in many different ways. At a first glance, it might seem to be confusing. But, we have proven this alternatively by obtaining the same result below (6). The main point is the emergence of the physically valid structure for $P_2 - P_1 = m\Delta u + (u - u^b)\Delta m$ after (8). This is the main structure, that any physically correct derivation must produce. The revised text clearly mentions it as [P6]: "It is important to note that $P_2 - P_1 = m\Delta u + (u - u^b)\Delta m$ is the main structure that any physically correct derivation must produce for the erosional landslide. This is clear from the two alternative derivations presented above. Moreover, at this point, it is crucial to realize, that the momentum equation (9) for the erosional landslide must be derived rigorously as done here by following the first-principles, but cannot just be speculated arbitrarily." So, different components of the landslide mass and eroded mass and the velocities can be arranged to establish their associations in many different ways resulting in the same model equation for landslide mobility that correctly incorporates the erosion induced inertia.

In the mean time, we have derived it in a further simple, yet more intuitive way as follows:

Let at time t_1 the landslide of mass m moves with the velocity u and the eroded mass Δm that has just been mobilized a moment ago moves with the erosion velocity u^b in which $u^b < u$. At the later time t_2 , the landslide with mass m strikes the little mass Δm with a perfectly inelastic collision, which is natural to happen.

Consequently, the mass Δm is embedded in the landslide resulting in the entrainment. At this time, the total of the landslide mass and the entrained mass ($m + \Delta m$) moves together with a single velocity ($u + \Delta u$), where Δu is the increment in the landslide velocity u . So, in the frame of reference of a stationary observer, the momentum P_1 at time t_1 and the momentum P_2 at time t_2 , respectively, are:

$$P_1 = mu + \Delta m u^b$$

$$P_2 = (m + \Delta m)(u + \Delta u)$$

Since $P_2 - P_1 = mu + m\Delta u + u\Delta m + \Delta u\Delta m - mu - \Delta m u^b = m\Delta u + \Delta u\Delta m + (u - u^b)\Delta m \approx m\Delta u + (u - u^b)\Delta m$, we obtain the momentum equation for an erosional landslide:

$$F = m \frac{du}{dt} + (u - u^b) \frac{dm}{dt}$$

In which the higher order term $\Delta u\Delta m \ll 1$ is ignored. The above equation is the same as the erosional landslide equation in the present manuscript, however, now the derivation is much simpler. Nevertheless, we prefer to report this new derivation elsewhere as a (brief) paper. We hope the reviewer agrees with us. Otherwise, if the reviewer suggests it would be better also to put the third derivation in the manuscript we are equally happy to do that.

So, all three derivations, two in the paper, and this extra presented above, lead to the same result. This proves that we are physically and mathematically fully consistent. This is why we call (5)-(6) the most elegant formulation. Physical correctness of the model derivation with subheading "Correct Derivation of the Relevant Momentum Balance Equation" [P6] has been clearly demonstrated by (9) and all further derivations and discussions.

- a. The entrained mass has been defined legitimately, because, $m + (-\Delta m) + \Delta m = m$ is the actual mass after entrainment. So, Δm is later added by erosion.
- b. On "Is this correct or not?": No. Because, at time t_2 , $m + (-\Delta m) + \Delta m = m$ is the landslide mass and Δm is the eroded mass. This has been explicitly proven in (7). Please also see the new derivation above and physical justification.
- c. This has been made clear by improving the writing [P6]: "In fact, the negative sign is not due to the negative change of the mass but rather due to the component $(-\Delta m)$ in the mass at time t_1 . This is consistent with the momentum conservation for the erosional landslide. We make this clearer now." This is so because, the alternative derivation in (7), including the text afterwards, and also the new derivation above, which are more intuitive than in (5)-(6), all prove this.

In the response to comment C10, the authors proposed a third derivation of the inertial term for an erosive landslide. I think that it is much more intuitive than those reported in the manuscript. I suggest replacing the two derivations of the manuscript with the derivation reported in the response R10.

ReResponse to Reviewer # 3: MS #: NCOMMS-21-19648

Reviewer's previous comments and suggestions are in red text indicated by C, our previous responses and enhancements are in blue text as indicated by R. The new comments are in plain text CC and responses are in green text, denoted by RR, respectively. Please see the improved manuscript text in accordance with the responses here in which the removed portions are in magenta, and edited and added texts are in green colour. P refers to the page number of the re-revised manuscript with track changes in colours.

CC: I thank the authors for the answers to the first review of the manuscript and for considering most of my suggestions. The responses to some of my comments and the related changes in the manuscript can be considered acceptable. However, some doubts and concerns about the proposed theory still remain. Furthermore, the writing style seems to be not so objective. In many parts of the manuscript, the proposed theory is defined as correct, but there is no real proof that it represents accurately the real process of erosion. All the doubts and concerns about the theory and the comments on the writing style will become clearer, in the following, by reading the comments reported in the "Major comments" section. In addition, in the "Minor comments" section, some other suggestions are reported.

In this review, the new comments are denoted by the word "Comment" and in some of them, there appear both the comments of the previous review (in red) and the related authors' answers (in blue).

RR: We very much appreciate the reviewer for supporting our work with all legitimate comments and very useful suggestions. Following the reviewers' suggestions, we have again improved the paper by eliminating all the doubts about the proposed theory. Moreover, the writing style has been made objective. Once more, our sincere thanks to the reviewer for such an extraordinary review with further detailed and very constructive comments and explicit suggestions that truly resulted in the substantially improved re-revised manuscript in which we appropriately addressed all the concerns raised by the reviewer in the "Major comments" section and the "Minor comments" section. We believe that the re-revised manuscript is now suitable to appear in Nature Communications soon.

MAJOR COMMENTS:

Comment 1:

C 11): In Section 3.1 (pages 5 - 6), is there any assumption concerning the velocity of the eroded mass u^b ? The reason for this question lies in the fact that the equation (8) can be derived from the classical definition of the inertia as follows: $\frac{d(mu)}{dt} = \frac{d}{dt}[m(u - u^b)] = m\frac{d}{dt}(u - u^b) + (u - u^b)\frac{dm}{dt} = m\frac{du}{dt} + (u - u^b)\frac{dm}{dt}$ where u^b is assumed to be constant in time and space.

If this assumption is correct also for the formulation reported in the manuscript, it is in contradiction with the closure relation used, i.e., $u^b = \lambda^b u$, since the mean flow velocity u can change over time and space.

R 11): On "is there any assumption concerning the velocity of the eroded mass u^b ": No, there is no assumption on the velocity of the eroded mass u^b .

On: "where u^b is assumed to be constant in time and space": No. So, since u^b in general is a variable, the reviewer's proposed equation is not valid for erosive landslide, because: $\frac{d(mu)}{dt} \neq \frac{d}{dt}[m(u - u^b)]$ and $m\frac{d}{dt}(u - u^b) + (u - u^b)\frac{dm}{dt} \neq m\frac{du}{dt} + (u - u^b)\frac{dm}{dt}$.

On "the closure relation used, i.e., $u^b = \lambda^b u$ ": This closure relation is valid. Also see the text before (12) for the generality of this expression.

CC 1: In the comment in red, the inertial term of the momentum balance principle is derived considering a reference frame that is moving with the eroded mass with a constant velocity u^b (inertial reference frame).

This expression for the inertial term coincides with the inertial term that appears at the right-hand side of equation (9) of the manuscript. The only difference between the two expressions is related to the condition on the velocity of the eroded material. In the red comment, this condition coincides with $u^b = \text{const}$ and is part of the proof. Conversely, in the manuscript, it is defined by $u^b = \lambda^b u$ and is applied only after the derivation of the inertial term.

Can the authors explain why it is possible to obtain the same expression for the inertial term from two different starting points? Since there is a discrepancy between the two conditions of the velocity of the eroded material, might the proof provided in the red comment highlight an important aspect of the proposed theory not visible using the discrete approach of the section “Correct Derivation of the Relevant Momentum Balance Equation”? I am afraid that there might be something missing in the theory proposed. Since the condition on the velocity of the eroded material is the key element of the whole work, I suggest carefully considering this comment and justifying the reply on it in detail.

RR 1: Since the velocity of the eroded material is the key element of the whole work, we have very carefully derived the fundamental model (9) and justified its physical ground. It is not possible to obtain the same expression for the inertial term from two different starting points as suggested by the reviewer, and thus there is no discrepancy. Because the expression in red is not valid in general for the erosive landslide, we cannot compare it with the model (9) of the present manuscript, and thus, the text in red cannot be considered as part of the proof. So, there is nothing invisible and missing in our formulation of (9). First, as we mentioned in the previous response that u^b in general is a variable, and the reviewer’s proposed equation is not valid for erosive landslide. Because, the condition $u^b = \text{const}$ is not correct for the erosive landslide and cannot be a part of the proof. Second, the fundamental equation (9) holds true for any general erosion velocity u^b . $u^b = \lambda^b u$ is a possible closure for u^b as constructed in²⁷, and $u^b = \lambda^b u$ is not a constant but also a general expression because u varies and λ^b also can take a general/variable expression (please see text before and around (19)). Thus, in our formal and rigorous derivation, u^b is a general variable. But in the formulation suggested by the reviewer, u^b is a constant, which can not be true in general. So, the statement “might the proof provided in the red comment highlight an important aspect of the proposed theory not visible using the discrete approach of the section “Correct Derivation of the Relevant Momentum Balance Equation?” is not correct. The presented model is general and complete. To further highlight this, we already mentioned that [P 6]: “Moreover, at this point, it is crucial to realize, that the momentum equation (9) for the erosional landslide must be derived rigorously as done here by following the first-principles (also see Supporting Information), but cannot just be speculated arbitrarily.”

Comment 2:

C 3): In the Abstract and Summary sections (page 1 line 22 and page 20 line 35), there appears the sentence “we demonstrate that the erosion and entrainment are essentially different processes”. A physical and mathematical proof is not reported in the manuscript. The distinction between the two processes (Section 7, pages 10 - 11) seems indeed to arise only from the definitions of:

- the erosion velocity u^b , that is the velocity of the eroded material incorporated into the landslide mixture and moving in the same direction of the landslide;
- the entrainment velocity $u - u^b$.

R 3): We have indeed proven this with the physical processes and the mathematical structures that the erosion and entrainment are essentially different phenomena. This is evident from the fact that erosion $u^e = u^b$ and entrainment $u^{en} = u - u^b$ are different in general. They can be the same only in a very special situation for which $u^b = \frac{1}{2}u$, which is very unlikely to occur in nature. This has been made clear now [P9].

CC 2: I have still some doubts about the fact that this manuscript demonstrates that “erosion and entrainment are different processes”. In my opinion, it is how the erosion and entrainment velocities are defined that makes the two processes different from each other. The relative velocity between the landslide and the eroded material $u - u^b$ is CALLED, in this manuscript, as “entrainment velocity”, but it is not explained clearly WHY this relative velocity should represent the velocity of the entrained mass. Can the authors explain this aspect by

making some simple physical examples? They could be very helpful for the reader.

RR 2: We can somehow understand these doubts. However, it is how the erosion and entrainment velocities emerge from two mechanically fundamentally different processes. These aspects have been explained in the text as [P 3 - 4]:

“Erosion: a mechanical process by which the bed material is mobilized by the flow. Entrainment: a mechanical process by which the eroded material is incorporated (entrained) and taken along with by the flow. Erosion-velocity: $u^e := u^b$, the velocity of the eroded material from the basal substrate. Entrainment-velocity: $u^{ev} := u - u^b$, the velocity of the entrained mass, or the velocity of the landslide minus the velocity of the eroded mass.” And [P 11]:

“Understanding the difference between erosion and entrainment is important. The existing literature could not distinguish between the erosion and entrainment as these terms are used interchangeably. However, here, we have made it very clear with the mechanical expressions, that the erosion and entrainment are essentially different phenomena. Erosion is a process by which the bed material is mobilized by the flow with the velocity $u^e = u^b$, while entrainment is intrinsically another process by which the eroded material is incorporated (entrained) and taken along with by the flow with the velocity $u^{ev} = u - u^b$. This fundamentally enhances our understanding of basic, but different processes in erosion related phenomena in landslide by clearly defining, and distinguishing the mechanisms of erosion and entrainment. These are important novel aspects.”

Once we have the correct physical understanding of the process and their mechanical origins, then, only for the notational convenience, we define these terms in this way. It is not just defined. To make this clearer, we have added the following text in the re-revised manuscript [P 11]:

“Erosion velocity and entrainment velocity systematically appear in the fundamentally derived momentum equation (9). The erosion velocity (u^b) enters the momentum equation (2) through the boundary condition^{27,62} applied to the erodible interface, that combined with the erosion rate (E), produces the erosion-induced momentum production, $u^b E$ in (2). This process is induced due to the mobilized bed material. Whereas the entrainment velocity ($u - u^b$) appears fundamentally differently due to the correct derivation of the relevant momentum balance equation as clearly revealed by (9) that entrains the newly eroded and added material associated with $\frac{dm}{dt}$ with the erosion velocity $u - u^b$ producing the term $(u - u^b) \frac{dm}{dt} = u^{ev} \frac{dm}{dt}$. So, in (9), F in the left contains $u^b E$ that is produced by one process, but $u^{ev} \frac{dm}{dt}$ emerges on the right that is generated by completely another process. Hence, it is structurally and mechanically clear, that the entrainment velocity, as given by the relative velocity, $u - u^b$, in fact, represents the velocity of the entrained mass associated with $\frac{dm}{dt}$.”

If the erosion velocity would have been zero, the landslide must have entrained the newly added mass with the velocity of the landslide, u . However, since in general the mobilized mass already moves with the erosion velocity u^b , the landslide can entrain the new mass just with the entrainment velocity $u - u^b$, which depending on the value of $u^b > 0$, can be much less than the landslide velocity u (i.e., $u < u - u^b$) resulting in the erosion induced higher mobility. These crucial aspects have been extensively explained in the text proving that the erosion and entrainment are quite different physical processes. For example, assume that the landslide is moving with the typical velocity of $u = 60 \text{ ms}^{-1}$ and the erosion velocity is $u^b = \lambda^b u = 1/1.33 * 60 \approx 45 \text{ ms}^{-1}$, where λ^b can be determined from the inertial process as explained in (19). Then, the erosional landslide equation (9) implies that the entrainment velocity is $u^{ev} = u - u^b = 60 - 45 = 15 \text{ ms}^{-1}$. The landslide velocity is determined dynamically by its motion, whereas the erosion velocity is the result of the mechanical competition between the sliding mass and the resisting erodible bed. Moreover, the entrainment velocity is another process, as determined by the inertia, model (9), by which the eroded mass is incorporated in to and taken along with by the landslide. So, there are clear mechanical processes and ways to determine the erosion and entrainment velocities. This is a seminal discovery.

Comment 3:

C 7): In Section 2.1 (page 4), the equation (2) is derived from the model of Pudasaini and Mergili (2019) by summing up the three momentum balance principles related to the three phases. However, the resulting equation is different from that derived in this manuscript. Defining α_s and α_{fs} (Pudasaini and Mergili, 2019) as the concentrations of the coarse and fine solid phases respectively, the term dependent on K at the left-hand side of the equation and the Coulomb friction at the right-hand side of the equation should depend on α_s rather than on α^m , which is defined (lines 11 - 12, page 4) as “the volume fraction of the solid particles (coarse and fine solid particles)”. Is there any assumption not declared?

R 7): Equation (2) is derived directly from the model of Pudasaini and Mergili (2019) by summing up the three momentum balance principles related to the three phases. The equation is correct, we have again checked it. As explained above, the solid and fine-solid is represented by a single solid phase with the volume fraction $\alpha^m = \alpha_s + \alpha_{fs}$. There is no extra assumption. To make it clear, we have now explicitly defined $\alpha^m = \alpha_s + \alpha_{fs}$ in the revised text [P4].

CC 3: I try to reformulate comment C7 of the first review since it might be not so clear. I think that the equation (2) is derived from Pudasaini and Fischer (2020) rather than from Pudasaini and Mergili (2019). This aspect arises from the fact that in Pudasaini and Mergili (2019), the coarse solid and fine-solid phases are different materials defined as “a Mohr- Coulomb continuum” and a viscous fluid, respectively. As a consequence of this different behaviour of the two solid phases, in Pudasaini and Mergili (2019), only the solid pressure term related to the coarse-solid phase (and not both the solid pressure terms for the coarse and fine solid phases) depends on the earth pressure coefficient K . This can be seen by combining the fourth terms at the left-hand side of the equations (14a) - (14b) of Pudasaini and Mergili (2019, p. 2930) with equation (18) in Pudasaini and Mergili (2019, p. 2931). That is why my previous comment C7 stated “the term dependent on K at the left-hand side of the equation and the Coulomb friction at the right-hand side of the equation should depend on α_s rather than on α^m ”.

Conversely, if we start the derivation the momentum balance equation from Pudasaini and Fischer (2020), we obtain equation (2). This is because the single solid phase (and thus, both the coarse and fine-solid phase) is defined as a Mohr-Coulomb continuum.

I suggest checking again the derivation and if this comment is wrong, please demonstrate it in detail. Moreover, I suggest checking that the references in the whole work are written in the right places.

RR 3: In Pudasaini and Mergili (2019) coarse-solid is a Mohr-Coulomb continuum and fine-solid is treated as Coulomb-viscoplastic material. In the present paper, the solid pressure term is related to the total solid phase (so, both the solid pressure terms for the coarse and fine solid phases) that is connected to the earth pressure coefficient K . Reviewers’ suggestion is not wrong. To avoid the confusion, we have now appropriately improved the text.

[P 3]: “... coarse solid and fine solid particles is represented by a single solid-phase: $\alpha^m := \alpha_s + \alpha_{fs} \dots$ ”
 “..., K is the earth pressure coefficient for the solid (composed of coarse solid and fine solid) particles in the mixture ... ”

[P 4]: “Furthermore, the coarse solid and the fine solid particles are assumed to have similar physical properties and constitute the solid volume fraction in the mixture. Thus, the mixture is composed of the solid ($\alpha_s + \alpha_{fs} = \alpha^m$) and the fluid fraction (α_f) such that $\alpha_s + \alpha_{fs} + \alpha_f = \alpha^m + \alpha_f = 1$. This reduces the situation to the effectively two-phase model²⁷ representing the motion of the mixture consisting of the solid (α^m) and the fluid (α_f) phases. To facilitate for the derivation of a simple model, with these assumptions, we are considering an effectively single-phase mixture flow. Because of the effectively single-phase nature of the model being developed here, either we can consider an extra closure relation for α^m , or parameterize it, or consider it as a constant. To keep the present basic model simple, we have assumed locally uniform mixture. Then, by summing up the mass and momentum balance equations^{27,61}, we obtain a single mass and momentum balance equation describing the motion of an erosive landslide and the evolution of the erodible bed surface, b as: ...”

We could also use the notation α_s instead of α^m , but then, α_s is equivalent to neglecting the fine solid, i.e., $\alpha_s + \alpha_{fs} = \alpha_s + 0 = \alpha_s$, whereas $\alpha^m := \alpha_s + \alpha_{fs}$ includes both the coarse solid and the fine solid as the total solid phase in the mixture satisfying the Mohr-Coulomb continuum. Moreover, this allows us considering the three-phase model⁶¹ and reduction to the two-phase model²⁷ that is compatible with the most recent version of the widely used influential opensource multi-phase computational tool r.avaflow (Mergili and Pudasaini, 2014-2021: <https://www.avaflow.org>) in which coarse solid and fine solid can be put together to constitute the total solid component in the mixture that can be described as a Mohr-Coulomb continuum. Such a reduction from three-phase⁶¹ to two-phase²⁷ also helps to extend the present model to a multi-phase mass flow simulation for erosive landslide [P 19].

Comment 4:

On page 9 after equation (17) of the new version of the manuscript, there appear the sentence “*Furthermore, with entrainment, values of $\rho^m; \gamma^m; \mu^m$ and α^m should be appropriately updated in proportion to the newly entrained material to the sliding material*”. It is important to stress that if ρ^m needs to be updated on the basis of the “amount” of the material entrained, ρ^m becomes variable in time and space. Thus, ρ^m does not cancel out both in the mass balance equation (1) and momentum balance equations (2) and (4). I suggest highlighting this aspect since the reader might understand that the theory proposed does not need any type of rearrangement of the equations.

RR 4: We thank the reviewer for this suggestion. We have enhanced the text as follows [P 8]: “However, in the present modelling frame, such updating can be achieved only through their parameterizations. For example, ρ^m cannot be considered as a full variable as it adds complications in the simplification of the mass and the momentum balance equations (1) and (2). So, consideration of ρ^m as a state variable, or its time and spatial variation, is out of scope here.”

Comment 5:

C 22): In the Summary section (page 20, lines 45 - 47), there appears the sentence “Analytically obtained values well represent the velocity of natural landslides and debris avalanches with erosion and demonstrate that erosion can have the major control on the landslide dynamics.” This sentence is not demonstrated. In fact, no estimations of the velocity of natural landslide using the analytical solution proposed in this manuscript are reported.

R 22): On “Analytically obtained values well represent the velocity of natural landslides and debris avalanches with erosion”: The text has been modified and now reads [P21; last paragraph in Discussion]: “We have (implicitly) shown that analytically obtained values well represent the velocity of natural landslides and debris avalanches with erosion and demonstrated that erosion can have the major control on the landslide dynamics.” On “demonstrate that erosion can have the major control on the landslide dynamics”: We think, this is clear from the analytical solution and the associated results. So, we hope no further explanation is required. Otherwise, we could have removed the word “demonstrated”.

CC 5: It seems that the authors wanted to show that the theory is good by comparing the velocities obtained from the analytical solution (first paragraph on page 13 of the new version of the manuscript) with the estimated velocities of some natural landslides and debris avalanches (paragraph that starts at the end of page 12 and continues on page 13 of the new version of the manuscript).

However, this comparison has no sense since the estimated velocities of natural landslides can be produced by conditions that are far away from those from which the analytical solutions are derived. More precisely, the proposed analytical solutions are derived considering a simple situation, i.e., the motion happens down a slope with no changes in the bottom profile (sentences reported on page 5 “*For the purpose of developing a simple landslide mobility equation, momentarily we consider the motion down an inclined slope. In this situation $u|u| = 1$ and $-g^z \frac{\partial b}{\partial x} = 0$* ”). Conversely, in real conditions, the topography could be much more complex than a slope. As a consequence, the estimated velocities of natural landslides can be very different from the velocities computed from the proposed analytical solutions.

The only way to demonstrate the sentence “*analytically obtained values well represent the velocity of natural*

landslides and debris avalanches with erosion” and to define the theory as “correct” consists in comparing the velocities of the analytical solutions with some results of laboratory experiments performed in the same conditions of the analytical solutions, i.e. considering a mixture that flows down a slope.

For this reason, I suggest deleting the sentence “*We have (implicitly) shown that analytically obtained values well represent the velocity of natural landslides and debris avalanches with erosion*” from the Summary and the sentence “*Obtained velocities are representative of natural landslides with erosion and indicate the fact that erosion can have the major control on the landslide dynamics*” from the Introduction.

RR 5: Many thanks for these suggestions. We agree and the text have been revised as suggested, which now read [P 3, Introduction]: “*Obtained velocities indicate the fact that erosion can have the major control on the landslide dynamics.*”, and [P 21, Summary] “*Analytically obtained velocities demonstrate that erosion can have the major control on the landslide dynamics.*” The revised texts now do not mention about the comparison of the analytical velocity with velocity of the natural landslides.

MINOR COMMENTS:

Comment 1:

CC 1: On pages 4 and 5 of the new version of the manuscript, there appear the sentences “*Because of the effectively single-phase nature of the model, either we can consider an extra closure relation for α^m , or parameterize or consider it as a constant. To keep the present basic model simple, we have assumed locally uniform mixture, so the local spatial variation of α^m can be ignored. However, it has been restored while constructing the full model at the end of this section.*”

The assumption of uniform mixture ($\alpha_s = \text{const}$, $\alpha_{fs} = \text{const}$, $\alpha^m = \text{const}$) is valid also for the set of equations (1) - (2) since no equations for the changes over time and space in the concentrations are provided.

I suggest:

1. moving “*Because of the effectively single-phase nature of the model, either we can consider an extra closure relation for α^m , or parameterize or consider it as a constant. To keep the present basic model simple, we have assumed locally uniform mixture*” on page 4 after the sentence “*This means, to facilitate for the derivation of a simple model, with these assumptions, we are considering an effectively single-phase mixture flow*”.
2. writing “*due to the uniform mixture assumption, the local spatial variation of α^m can be ignored. However, it has been restored while constructing the full model at the end of this section*” on page 5 before equation (4).

RR 1: Thank you very much. We follow the suggestions as these better fit with the new arrangements [P 4]: “*Because of the effectively single-phase nature of the model, either we can consider an extra closure relation for α^m , or parameterize it, or consider it as a constant. To keep the present basic model simple, we have assumed locally uniform mixture.*” And, [P 5]: “*Due to the uniform mixture assumption, the local spatial variation of α^m can be ignored. However, it has been restored while constructing the full model at the end of this section.*”

Comment 2:

C 10): In Section 3.1 (pages 5), the inertial term of the momentum balance principle is derived considering a discrete description of the landslide and the eroded material at two different times. The derivation creates confusion, since:

- a. In this discrete approach, the entrained mass is defined as $-\Delta m$ (line 26).
- b. The quantity P_2 is defined as “momentum of the landslide and the eroded mass at time t_2 ” (lines 25-26), but it is difficult to understand why this variable represents the momentum of both the landslide and the eroded mass as a whole. In fact, looking at the equation (5), it seems that P_2 expresses the momentum of the landslide at the final time purified from the contribution of the entrained material

rather than the momentum of both the landslide and the eroded mass as a whole. Is this correct or not? Why?

c. The sentence reported at lines 31-32, i.e., “In fact, the negative sign is not due to the negative change of the mass but rather due to the relative erosion velocity as compared to the actual velocity of the landslide”, is not clear.

R 10): We can understand the possible cause of confusion, and since this is the first paper to correctly handle the inertia for the erosive mass transport, not all aspects might have been clear, we hope in time it will be more clearer as the same expressions as presented here can, and will be derived later in many different ways. At a first glance, it might seem to be confusing. But, we have proven this alternatively by obtaining the same result below (6). The main point is the emergence of the physically valid structure for $P_2 - P_1 = m\Delta u + (u - u^b)\Delta m$ after (8). This is the main structure, that any physically correct derivation must produce. The revised text clearly mentions it as [P6]: “It is important to note that $P_2 - P_1 = m\Delta u + (u - u^b)\Delta m$ is the main structure that any physically correct derivation must produce for the erosional landslide. This is clear from the two alternative derivations presented above. Moreover, at this point, it is crucial to realize, that the momentum equation (9) for the erosional landslide must be derived rigorously as done here by following the first-principles, but cannot just be speculated arbitrarily.” So, different components of the landslide mass and eroded mass and the velocities can be arranged to establish their associations in many different ways resulting in the same model equation for landslide mobility that correctly incorporates the erosion induced inertia.

In the mean time, we have derived it in a further simple, yet more intuitive way as follows:

Let at time t_1 the landslide of mass m moves with the velocity u and the eroded mass Δm that has just been mobilized a moment ago moves with the erosion velocity u^b in which $u^b < u$. At the later time t_2 , the landslide with mass m strikes the little mass Δm with a perfectly inelastic collision, which is natural to happen. Consequently, the mass Δm is embedded in the landslide resulting in the entrainment. At this time, the total of the landslide mass and the entrained mass ($m + \Delta m$) moves together with a single velocity ($u + \Delta u$), where Δu is the increment in the landslide velocity u . So, in the frame of reference of a stationary observer, the momentum P_1 at time t_1 and the momentum P_2 at time t_2 , respectively, are:

$$P_1 = m u + \Delta m u^b, \quad (1)$$

$$P_2 = (m + \Delta m) (u + \Delta u). \quad (2)$$

Since $P_2 - P_1 = m u + m \Delta u + u \Delta m + \Delta u \Delta m - m u - u^b \Delta m = m \Delta u + \Delta u \Delta m + (u - u^b) \Delta m \approx m \Delta u + (u - u^b) \Delta m$, we obtain the momentum equation for an erosional landslide:

$$F = m \frac{du}{dt} + (u - u^b) \frac{dm}{dt}, \quad (3)$$

in which, the higher order term $\Delta u \Delta m \ll 1$ is ignored. The above equation is the same as the erosional landslide equation in the present manuscript, however, now the derivation is much simpler. Nevertheless, we prefer to report this new derivation elsewhere as a (brief) paper. We hope the reviewer agrees with us. Otherwise, if the reviewer suggests it would be better also to put the third derivation in the manuscript we are equally happy to do that.

So, all three derivations, two in the paper, and this extra presented above, lead to the same result. This proves that we are physically and mathematically fully consistent. This is why we call (5)-(6) the most elegant formulation. Physical correctness of the model derivation with subheading “Correct Derivation of the Relevant Momentum Balance Equation” [P6] has been clearly demonstrated by (9) and all further derivations and discussions.

a. The entrained mass has been defined legitimately, because, $m + (-\Delta m) + \Delta m = m$ is the actual mass after entrainment. So, Δm is later added by erosion.

b. On “Is this correct or not?”: No. Because, at time t_2 , $m + (-\Delta m) + \Delta m = m$ is the landslide mass and Δm is the eroded mass. This has been explicitly proven in (7). Please also see the new derivation above and physical justification.

c. This has been made clear by improving the writing [P6]: “In fact, the negative sign is not due to the negative change of the mass but rather due to the component $(-\Delta m)$ in the mass at time t_1 . This is consistent with the momentum conservation for the erosional landslide. We make this clearer now.” This is so because, the alternative derivation in (7), including the text afterwards, and also the new derivation above, which are more intuitive than in (5) - (6), all prove this.

CC 2: In the response to comment C10, the authors proposed a third derivation of the inertial term for an erosive landslide. I think that it is much more intuitive than those reported in the manuscript. I suggest replacing the two derivations of the manuscript with the derivation reported in the response R10.

RR 2: Thank you very much for the suggestion. We have changed the text accordingly [P 5 - 6]. However, from the physical stand, the previous derivations are superior and much more elegant as those require no condition. The new derivation is much more intuitive than those reported in the manuscript, but it contains a condition of perfectly inelastic collision. We have mentioned this in the text, and put the previous derivations in Supporting Information such that the readers can also find those original derivations.

REVIEWERS' COMMENTS

Reviewer #3 (Remarks to the Author):

I thank the authors for replying to my comments since most of the additional explanations clarify my doubts. In my opinion, it is now clear that erosion and entrainment are different processes as they enter the momentum balance as a momentum transfer and a change in the inertia of the landslide, respectively. With reference to the derivation of the "correct" change in the landslide inertia, I suggest considering in the future derivations that differ from the discrete procedure. In this way, the whole theory might be strengthened. Finally, I suggest controlling that the non-modified text of the manuscript agrees with the changes performed after the review. For example, on page 7, in the paragraph discussing the differences between the equation (9) and the Tsiolkovsky Rocket Equation, there appears a reference to the derivation reported in the Supplementary Information rather than to the derivation on page 5.

ReReResponse to Reviewer # 3: MS #: NCOMMS-21-19648

Reviewer's comments are in plain text C and responses are in blue text, denoted by R, respectively. Please see the improved manuscript text in accordance with the responses here in which the edited and added texts are in blue colour and the removed texts are in red colour. P refers to the page number of the currently revised manuscript with track changes in colours.

C: I thank the authors for replying to my comments since most of the additional explanations clarify my doubts. In my opinion, it is now clear that erosion and entrainment are different processes as they enter the momentum balance as a momentum transfer and a change in the inertia of the landslide, respectively. With reference to the derivation of the “correct”; change in the landslide inertia, I suggest considering in the future derivations that differ from the discrete procedure. In this way, the whole theory might be strengthened. Finally, I suggest controlling that the non-modified text of the manuscript agrees with the changes performed after the review. For example, on page 7, in the paragraph discussing the differences between the equation (9) and the Tsiolkovsky Rocket Equation, there appears a reference to the derivation reported in the Supplementary Information rather than to the derivation on page 5.

R: We very much appreciate the reviewer for supporting our work with some final suggestions. Following the reviewers' suggestions, we have again improved the paper. As before, our sincere thanks to the reviewer for the time and constructive comments and explicit suggestions that truly resulted in the substantially improved final version of the manuscript in which we appropriately addressed all the concerns raised by the reviewer. We look forward to see soon the final version of the manuscript to appear very soon in Nature Communications.

On “I suggest considering in the future derivations that differ from the discrete procedure.” We fully agree, and the text has been improved by inserting the sentence [P18]: “We note that, in general, the complete and the continuum description of the dynamical landslide equations with erosion (35)-(37) are preferable over their discrete counter part (9) or (18).”

On “I suggest controlling that the non-modified text of the manuscript agrees with the changes performed after the review. For example, on page 7, in the paragraph discussing the differences between the equation (9) and the Tsiolkovsky Rocket Equation, there appears a reference to the derivation reported in the Supplementary Information rather than to the derivation on page 5.”: We appreciate very much for these useful suggestions. We have thoroughly checked and improved the revised ms, this included correct citation of references on P5, P10 and P18. Furthermore, equation (9) derived on P5-P6 of the main text and (S.5) on P1-P2 of the Supplementary Information both are similar in the form to the Tsiolkovsky Rocket Equation. To make it clearer, we have now improved the text [P7] as: “We call (9) the landslide-rocket-equation. In the form, (9) is similar to the famous Tsiolkovsky Rocket-Equation⁶⁴. However, there are fundamental differences. First, the way we derive the model is different. Second, the mass of the rocket is decreasing (since it consumes fuel), so dm/dt is negative. But, for erosional landslide dm/dt is positive as the mass of landslide is increasing. Third, although the multiplier of dm/dt is positive for both the erosional landslide and the rocket, they have quite different perspectives and mechanisms. For the rocket, it is the velocity of the exhaust, say u^{ex} . But, for the erosional landslide, it is the velocity of the landslide minus the velocity of the eroded mass that is entrained by the landslide. Thus, depending on the magnitude of the erosion velocity, the entrainment velocity u^{ev} can be substantially less than the landslide velocity, as the velocity of the eroded particle, that is entrained by the landslide, is a positive quantity that, depending on the situation (the flow and the bed morphology), can be as high as the velocity of the landslide itself. Further detail on it can be found in Supplementary Information.” We have also consistently amended the text in the Supplementary Information.